

SciPost Phys. Lect. Notes 92 (2025)

# Scattering from an external field in quantum chromodynamics at high energies: From foundations to interdisciplinary connections

Athanasia-Konstantina Angelopoulou[1], Anh Dung Le[2,3] and Stéphane Munier[4*]

**1** School of Mathematics and Hamilton Mathematics Institute,
Trinity College, Dublin 2, Ireland
**2** Department of Physics, University of Jyväskylä, P.O. Box 35,
FI-40014 University of Jyväskylä, Finland
**3** Helsinki Institute of Physics, P.O. Box 64, 00014 University of Helsinki, Finland
**4** CPHT, CNRS, École polytechnique, Institut Polytechnique de Paris, 91120 Palaiseau, France

⋆ stephane.munier@polytechnique.edu

## Abstract

We review the factorization of the *S*-matrix elements in the context of particle scattering off an external field, which can serve as a model for the field of a large nucleus. The factorization takes the form of a convolution of light cone wave functions describing the physical incoming and outgoing states in terms of bare partons, and products of Wilson lines. The latter represent the interaction between the bare partons and the external field. Specializing to elastic scattering amplitudes of onia at very high energies, we introduce the color dipole model, which formulates the calculation of the modulus-squared of the wave functions in quantum chromodynamics with the help of a branching random walk, and the scattering amplitudes as observables on this classical stochastic process. Methods developed for general branching processes produce analytical formulas for the asymptotics of such observables, and thus enable one to derive exact large-rapidity expressions for onium-nucleus cross sections, from which electron-nucleus cross sections may be inferred.

| | |
|---|---|
| Received | 2024-10-15 |
| Accepted | 2025-02-11 |
| Published | 2025-03-12 |

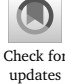

# 1   Introduction

Observables measured at high-energy colliders such as the Large Hadron Collider (LHC), the future Electron-Ion Collider (EIC) [1,2], or planned facilities such as the Large Hadron-electron Collider (LHeC) [3] or the Future Circular Collider (FCC) [4] have fostered a lot of interest in the high-energy regime of quantum chromodynamics (QCD) [5]. The latter is sometimes also called the "Regge regime". This refers to kinematics in which the center-of-mass energy $\sqrt{s}$ of the reactions is very large, much larger than all other scales involved, such as the masses of the produced particles, the transverse momenta of jets, or the virtuality $Q$ of the photon that mediates the electron-hadron interaction in deep-inelastic scattering.

The high-energy regime was recognized a long time ago to be particularly interesting from a theoretical viewpoint, since multiple rescatterings between the interacting objects, and possibly parton recombination in the evolution of their states, may become instrumental effects [6–8]. The standard paradigm assumes that such non-linearities are negligible. This proved to be a very good approximation at lower energies, but it turns out that under such assumptions, the fundamental property of unitarity of scattering amplitudes gets violated at higher energies, already at short distances. This is a sign that the linear formalism is incomplete. The effects mentioned, captured by non-linear terms in the dynamical equations governing scattering amplitudes, make it possible to remedy this locally.[1]

To date, several formalisms based on QCD have been developed to address the calculation of scattering amplitudes in the high-energy regime (see e.g. original papers such as [12–21], and the textbook [5] for an overview and more references). The one we shall focus on in these lectures, based on "old-fashioned" perturbation theory in a light cone gauge and frame, is particularly close to the intuition of the parton model. In this model, fast hadrons are thought of as fluctuating sets of pointlike quarks and gluons (named "partons"), looking as free to an observer. A scattering amplitude involving a large atomic nucleus and a particle interacting as a "dilute hadron", such as a proton, or an electron which necessarily interacts through a virtual photon converting to a quark-antiquark dipole, may be factorized as the convolution

---

[1]A purely perturbative picture can obviously not be complete at all distance scales: the Froissart bound [9] on the total hadronic cross sections for example, a direct consequence of unitarity of the $S$-matrix, requires a mass gap which can only be understood non-perturbatively; see the controversy in Refs. [10,11].

of light-cone wave functions and products of Wilson lines. The former describe the partonic content of the initial and final asymptotic states of the dilute hadron, and the latter encode the interactions of the partons with the nucleus. Of particular interest is the forward elastic scattering amplitude, from which the main observable measured at colliders, namely the total cross section, can be derived through the optical theorem.

Light cone perturbation theory (LCPT) is used to compute the probability amplitudes of the partonic fluctuations. In the limit of a large number of colors, these fluctuations may be organized in sets of color dipoles building up in a stochastic process formulated as an evolution in rapidity of states through a binary branching process [12, 22]. Dipole evolution turns out to belong to a wide class of classical stochastic processes, the simplest representant of which might be the branching Brownian motion (BBM) [23]. The latter is under active study in mathematics [24], in different areas of physics [25], but also in biology [26], chemistry [27], and finds applications in computer science [28] and economics [29].

The main goal of these lectures is to explain the connections between QCD in the regime of very high energies and the BBM. A large part of it is dedicated to the introduction of the color dipole model, starting from the basics of the particular formulation of quantum field theory on which it is based. We do not present an extensive review of the light cone quantization of QCD, for there are already several of them (see e.g. the classical reviews [30, 31], the more recent textbook [5] for the aspects of LCPT that are needed for our purpose, and research papers such as Ref. [32] for a synthesis of the formalism and to get a flavor of advanced calculations based on it). However, an effort was made to keep this review self-contained and elementary enough. For example, we expose a quite complete derivation of the factorization of amplitudes in the high-energy regime, which is usually overlooked or dealt with very quickly in modern reviews or texts. Our goal is to provide all tools to someone who would like to understand the derivation of one of the most emblematic equations of the field of high-energy QCD, namely the Balitsky-Kovchegov (BK) equation [13, 16], but in a slightly different way to the one used in textbooks: our presentation of the subject is in fact geared towards explaining how high-energy QCD cross sections may be understood in terms of classical stochastic processes.

In a first step, we most generally formulate scattering amplitudes of quantum particles off an external field, that may model a large nucleus, in the limit of very high energies (Sec. 2). In a second step, we specialize to forward elastic onium scattering, first in a purely classical approximation, then including the lowest-order quantum correction in a perturbative expansion of the state of the onium (Sec. 3). We eventually generalize to higher orders (Sec. 4) by establishing the color dipole model, and present the all-order calculation in the form of the BK equation. We argue that the latter belongs to the universality class of equations for observables in branching random walks. We then naturally digress into the topic of branching random walks in a broader context (Sec. 5), to present known properties that can be used to help understanding QCD evolution equations, before applying them to high-energy scattering (Sec. 6). In the concluding section 7, we summarize and briefly mention some other applications of classical branching processes not covered in these lectures. Appendix A gathers some useful formulas and calculation techniques used in the body of the text. Appendix B contains the details of the calculation of the light cone wave function of a virtual photon in a quark-antiquark state.

## 2 High-energy scattering off an external classical field

In this section, we shall review the most general formulation of scattering in quantum field theory (Sec. 2.1), before turning to the high-energy limit of the scattering of a quantum particle off a classical vector field (Sec. 2.2), whose amplitude may be elegantly factorized. We then

specialize to quantum chromodynamics (Sec. 2.3) and, in this context, we derive an explicit formula for $S$-matrix elements in the form of a convolution of light cone wave functions, and products of Wilson lines. We provide all the tools for the evaluation of the former.

Section 2.1 involves general quantum field theory [33–35]. Sections 2.2 and 2.3 heavily rely on the classical original papers [36–38] (see also [39]), which were also reviewed recently (with a slightly different focus and less details) in the lecture notes [40] and in the textbook [35] (Sec. 7.6.6).

## 2.1 Exact $S$-matrix

We shall formulate scattering probability amplitudes, first for the case of quantum particles scattering among themselves, and then for quantum particles scattering off an external classical field.

### 2.1.1 Scattering of quantum particles

Consider two systems described by the state vectors $|i\rangle$ and $|f\rangle$ in the Heisenberg picture of quantum mechanics. The first vector represents the initial state of a scattering process, while the second corresponds to the final state. Accordingly, we will refer to them as the "in" and "out" states, respectively.

Recall that in the Heisenberg picture, the states $|\psi\rangle_H$ are independent of time $\tau$, while the operators $O_H(\tau)$ contain all the time dependence. This contrasts with the Schrödinger picture, where the states $|\psi(\tau)\rangle_S$ carry the full time dependence, and the operators $O_S$ are time-independent, except for any explicit time dependence they may have. Given the Schrödinger Hamiltonian $H$ encoding the dynamics, the states and operators in the two pictures are related by the following unitary transformation:

$$|\psi\rangle_H = e^{iH\tau} |\psi(\tau)\rangle_S \,, \qquad O_H(\tau) = e^{iH\tau} O_S(\tau) e^{-iH\tau} \,. \tag{1}$$

The "in" and "out" states are such that if they were measured at asymptotically large negative and positive times $\tau$ respectively, they would "look like" states of sets of non-interacting particles. The transition amplitude between the "in" and the "out" states is just the overlap of the vectors $|i\rangle$ and $|f\rangle$:

$$S_{fi} = \langle f | i \rangle \,. \tag{2}$$

This is a matrix element of the scattering operator, the so-called "$S$-matrix". For a theory such as QED or QCD, the $S$-matrix elements may, for example, be expressed in the form of an integral over field configurations, weighted by a phase factor, the phase being the action. Lorentz-covariant Feynman rules can then be derived, from which any amplitude may in principle be computed systematically in perturbation theory (see e.g. Refs. [33,34]). However, in certain cases, such as the one addressed in this review, it is often more suitable to adopt a tailored formalism. In this work, we will introduce and employ (light-cone) time-ordered perturbation theory.

Assume that the Hamiltonian $H$ may be written as the sum $H_0 + H_1$, where $H_1$ contains all interaction terms. $H_0$, called the "free Hamiltonian", evolves the particles in the theoretical limit in which the interactions are switched off. It will prove convenient to expand the "in" and "out" state vectors as superpositions of the known eigenstates of the free Hamiltonian. For our purpose, this is best done in a time-dependent framework. The most suitable picture in which to formulate our problem is the "interaction", or "Dirac", representation of quantum mechanics. The states $|\psi(\tau)\rangle_I$ [respectively the operators $O_I(\tau)$] in the interaction picture are related to the states $|\psi(\tau)\rangle_S$ [respectively the operators $O_S(\tau)$] in the Schrödinger picture through the unitary transformation

$$|\psi(\tau)\rangle_I = e^{iH_0\tau} |\psi(\tau)\rangle_S \,, \qquad O_I(\tau) = e^{iH_0\tau} O_S(\tau) e^{-iH_0\tau} \tag{3}$$

(We displayed a $\tau$-dependence in $O_S$ to keep this equation general, but in most cases, the operators of interest are time-independent in the Schrödinger picture). Note that the Dirac picture would match the Heisenberg picture if the full Hamiltonian were $H_0$, i.e. in the free theory [compare Eqs. (1) and (3)].

We will obtain the Heisenberg states to compute the $S$-matrix elements (2) as interaction states evaluated at zero time, namely $|\Psi(0)\rangle_I$.

Let us write the evolution equations in the interaction picture. The Schrödinger equation

$$i\frac{d}{d\tau}|\psi(\tau)\rangle_S = H|\psi(\tau)\rangle_S \tag{4}$$

reads, when taken over to the interaction picture,

$$i\frac{d}{d\tau}|\psi(\tau)\rangle_I = H_{1I}(\tau)|\psi(\tau)\rangle_I\,, \qquad \text{with} \qquad H_{1I}(\tau) = e^{iH_0\tau}H_1 e^{-iH_0\tau}\,. \tag{5}$$

The operators (such as $H_1$) which have no explicit time dependence in the Schrödinger picture evolve instead as

$$i\frac{d}{d\tau}O_I(\tau) = [O_I(\tau), H_0]\,, \tag{6}$$

when expressed in the interaction picture, which just follows from taking the derivative of the second equation in (3).

Equation (5) explicitly shows that the time evolution of the Dirac-picture states is entirely driven by the interactions. While the solution of Eq. (4) reads

$$|\psi(\tau)\rangle_S = e^{-iH(\tau-\tau_0)}|\psi(\tau_0)\rangle_S\,, \tag{7}$$

equation (5) can be solved iteratively as

$$|\psi(\tau)\rangle_I = U_I(\tau, \tau_0)|\psi(\tau_0)\rangle_I\,, \tag{8}$$

where

$$U_I(\tau, \tau_0) = \mathbb{I} + (-i)\int_{\tau_0}^{\tau}d\tau_1 H_{1I}(\tau_1) + (-i)^2\int_{\tau_0}^{\tau}d\tau_2\int_{\tau_0}^{\tau_2}d\tau_1 H_{1I}(\tau_2)H_{1I}(\tau_1) + \cdots, \tag{9}$$

may formally be written in a resummed form as a "time-ordered exponential"[2]

$$U_I(\tau, \tau_0) \equiv T\exp\left(-i\int_{\tau_0}^{\tau}d\tau' H_{1I}(\tau')\right). \tag{10}$$

The operator $U_I(\tau, \tau_0)$ obeys the same Schrödinger equation (5) as $|\psi(\tau)\rangle_I$,

$$i\frac{d}{d\tau}U_I(\tau, \tau_0) = H_{1I}(\tau)U_I(\tau, \tau_0), \tag{11}$$

endowed with the boundary condition $U_I(\tau, \tau) = \mathbb{I}$ that has to be satisfied for all $\tau$. Using this fact together with the hermiticity of $H_{1I}(\tau)$, it is easy to check that $\partial_\tau[U_I^\dagger(\tau, \tau_0)U_I(\tau, \tau_0)] = 0$, and thus, as a consequence of the initial condition, $U_I^\dagger(\tau, \tau_0)U_I(\tau, \tau_0) = \mathbb{I}$. This means that $U_I(\tau, \tau_0)$ is an isometric operator. Now, taking the derivative of Eq. (9) with respect to $\tau_0$, we see that $U_I(\tau, \tau_0)$ also obeys a "backward" Schrödinger equation

$$-i\frac{d}{d\tau_0}U_I(\tau, \tau_0) = U_I(\tau, \tau_0)H_{1I}(\tau_0), \tag{12}$$

---

[2]Equation (10) would reduce to an ordinary exponential if $H_{1I}$ evaluated at different times commuted.

from which the identity $U_I(\tau, \tau_0)U_I^\dagger(\tau, \tau_0) = \mathbb{I}$ follows. Altogether, this shows that $U_I(\tau, \tau_0)$ is a unitary operator.[3]

We eventually wish to send $\tau_0$ to $-\infty$. But this limit is singular. There are various ways of defining properly an asymptotic evolution operator. For the purposes of this review, we shall stick to the so-called "adiabatic prescription",[4] which consists in modifying the Hamiltonian in the following way:

$$H \to H(\tau) = H_0 + e^{-\epsilon|\tau|}H_1 \,. \tag{13}$$

The idea is that it matches smoothly that of a free theory at asymptotically large times. The evolution operator is then amended as follows:

$$U_I(\tau, \tau_0) \to U_I^\epsilon(\tau, \tau_0) \equiv T \exp\left(-i \int_{\tau_0}^\tau d\tau' \, e^{-\epsilon|\tau'|} H_{1I}(\tau')\right). \tag{14}$$

It is easy to see that this operator is also unitary: the proof follows the same steps as that of the unitarity of $U_I$. The proof of the isometry property goes over to the limiting operator obtained by letting $\tau_0 \to -\infty$.

Let us write explicitly the states obtained by evolution from time $\tau = -\infty$ of a non-interacting state $|\mathbb{i}_\bullet\rangle$, eigenstate of $H_0$, as a superposition of eigenstates of $H_0$. (The expansion of the final state would go along the same lines). This follows from the iterative solution of the interaction-picture Schrödinger equation, in the form of the series (9). Decomposing the identity operator as a sum of projectors $\{|\mathbb{n}_i\rangle \langle \mathbb{n}_i|\}$ on the eigenspaces of $H_0$, the expansion of the "in" state can be organized as follows:

$$U_I^\epsilon(0, -\infty)|\mathbb{i}_\bullet\rangle = |\mathbb{i}_\bullet\rangle + (-i) \int_{-\infty}^0 d\tau_1 \, e^{\epsilon\tau_1} \sum_{|\mathbb{n}_0\rangle} |\mathbb{n}_0\rangle \langle \mathbb{n}_0| H_{1I}(\tau_1)|\mathbb{i}_\bullet\rangle \tag{15}$$

$$+ (-i)^2 \int_{-\infty}^0 d\tau_2 \int_{-\infty}^{\tau_2} d\tau_1 \, e^{\epsilon(\tau_2+\tau_1)} \sum_{|\mathbb{n}_0\rangle, |\mathbb{n}_1\rangle} |\mathbb{n}_0\rangle \langle \mathbb{n}_0| H_{1I}(\tau_2)|\mathbb{n}_1\rangle \langle \mathbb{n}_1| H_{1I}(\tau_1)|\mathbb{i}_\bullet\rangle + \cdots .$$

The sums over the Fock states $|\mathbb{n}_i\rangle$ include both discrete sums over the particle content and their discrete quantum numbers, as well as integrals over continuous quantum numbers, using the appropriate measure. Combinatorial factors are accounted for as well. The explicit form of the completeness identities, which determine these factors, will be discussed in detail as needed.

Substituting $H_{1I}(\tau)$ by its definition given in Eq. (5) and introducing the energies $E_{\mathbb{n}_i}$ of the states $|\mathbb{n}_i\rangle$, we can factorize the integrations over the times $\{\tau_i\}$ from the matrix elements of the time-independent interaction Hamiltonian $H_1$:

$$U_I^\epsilon(0, -\infty)|\mathbb{i}_\bullet\rangle = |\mathbb{i}_\bullet\rangle + \sum_{|\mathbb{n}_0\rangle} |\mathbb{n}_0\rangle \left[ (-i) \int_{-\infty}^0 d\tau_1 e^{i(E_{\mathbb{n}_0}-E_{\mathbb{i}_\bullet})\tau_1+\epsilon\tau_1} \langle \mathbb{n}_0| H_1 |\mathbb{i}_\bullet\rangle \right. \tag{16}$$

$$\left. + (-i)^2 \int_{-\infty}^0 d\tau_2 \int_{-\infty}^{\tau_2} d\tau_1 \sum_{|\mathbb{n}_1\rangle} e^{i(E_{\mathbb{n}_0}-E_{\mathbb{n}_1})\tau_2+\epsilon\tau_2} e^{i(E_{\mathbb{n}_1}-E_{\mathbb{i}_\bullet})\tau_1+\epsilon\tau_1} \langle \mathbb{n}_0| H_1 |\mathbb{n}_1\rangle \langle \mathbb{n}_1| H_1 |\mathbb{i}_\bullet\rangle + \cdots \right].$$

---

[3]The unitarity of $U_I$ is much more straightforward to see starting from the interaction picture representation of the Schrödinger evolution operator, namely $U_I(\tau, \tau_0) = e^{iH_0\tau}e^{-iH(\tau-\tau_0)}e^{-iH_0\tau_0}$. But it is less easy to take over to time-dependent potentials.

[4]Note that our formulation of the scattering problem differs from the one used in this context in most of the classical works, see e.g. Ref. [41], leading to a subtly different (though equivalent) Fock expansion (17), (18). (See however e.g. Refs. [22, 42] for a practical application of the formulation presented here.) This difference does however not show up in the tree-level calculations we will need to perform.

The time integrations can be performed, yielding

$$
\begin{aligned}
U_I^\epsilon(0,-\infty)\,|\mathbb{i}_\bullet\rangle = |\mathbb{i}_\bullet\rangle + \sum_{|\mathbb{n}_0\rangle} |\mathbb{n}_0\rangle \Bigg( & \frac{1}{E_{\mathbb{i}_\bullet} - E_{\mathbb{n}_0} + i\epsilon}\, \langle \mathbb{n}_0| H_1 |\mathbb{i}_\bullet\rangle \\
& + \frac{1}{E_{\mathbb{i}_\bullet} - E_{\mathbb{n}_0} + 2i\epsilon} \sum_{|\mathbb{n}_1\rangle} \frac{\langle \mathbb{n}_0|H_1|\mathbb{n}_1\rangle \langle \mathbb{n}_1|H_1|\mathbb{i}_\bullet\rangle}{E_{\mathbb{i}_\bullet} - E_{\mathbb{n}_1} + i\epsilon} \\
& + \frac{1}{E_{\mathbb{i}_\bullet} - E_{\mathbb{n}_0} + 3i\epsilon} \sum_{|\mathbb{n}_1\rangle,|\mathbb{n}_2\rangle} \frac{\langle \mathbb{n}_0|H_1|\mathbb{n}_2\rangle \langle \mathbb{n}_2|H_1|\mathbb{n}_1\rangle \langle \mathbb{n}_1|H_1|\mathbb{i}_\bullet\rangle}{(E_{\mathbb{i}_\bullet} - E_{\mathbb{n}_2} + 2i\epsilon)(E_{\mathbb{i}_\bullet} - E_{\mathbb{n}_1} + i\epsilon)} + \cdots \Bigg).
\end{aligned}
\tag{17}
$$

Since some energy denominators in the sums over the states are proportional to powers of $1/\epsilon$, the limit $\epsilon \to 0$ of this state vector is a priori singular. It turns out that within the adiabatic prescription, in the absence of infrared divergences or once they have been regularized, the singularity consists in an overall phase factor that oscillates faster and faster as the limit is approached.[5] Thus we could use the expression of $U_I^\epsilon(0,-\infty)\,|\mathbb{i}_\bullet\rangle$ given by Eq. (17) as the "in" state, without having to worry about the singularities in the $\epsilon \to 0$ limit: they would anyway drop out from transition rates. Alternatively, we may define regular wave functions dividing out the singular phase factor $e^{i\alpha(\epsilon)}$ from the coefficients of the different Fock states in the expansion (17), and subsequently letting $\epsilon$ go to 0. The expansion of the "in" state then takes the form

$$
|\mathbb{i}\rangle = \lim_{\epsilon \to 0}\Big[ e^{-i\alpha(\epsilon)} U_I^\epsilon(0,-\infty)\,|\mathbb{i}_\bullet\rangle \Big] \equiv \sqrt{Z}\,|\mathbb{i}_\bullet\rangle + \sum_{|\mathbb{n}_0\rangle \neq |\mathbb{i}_\bullet\rangle} |\mathbb{n}_0\rangle\, \psi_{\mathbb{n}_0,\mathbb{i}}.
\tag{18}
$$

The real number $Z$ is the probability to find the interaction state $|\mathbb{i}\rangle$ in the initial partonic realization $|\mathbb{i}_\bullet\rangle$. If $|\mathbb{i}_\bullet\rangle$ is a one-particle state, $Z$ coincides with the renormalization constant. It may be evaluated directly from the modulus squared of the coefficient of $|\mathbb{i}_\bullet\rangle$ in Eq. (17). But for our purpose, it will prove convenient to compute it using its relation to the other transition probabilities, which can be derived as follows.

One first forms a convenient superposition of Eq. (18) convoluting it with a wave packet $\phi_\bullet$. Thanks to the isometry property of the operator $U_I^\epsilon(0,-\infty)$, the "in" state obtained in this way in the left-hand side, that we may denote by $|\mathbb{i}(\phi_\bullet)\rangle$, has the same norm as the superposition $|\phi_\bullet\rangle$ of asymptotic states $|\mathbb{i}_\bullet\rangle$ in the first term in the right-hand side. One chooses the wave packet $\phi_\bullet$ such that $\langle \phi_\bullet | \phi_\bullet\rangle = 1$, and one eventually shrinks it to a monochromatic state, a limit that we shall formally denote by "$\phi_\bullet \to \mathbb{i}_\bullet$". One arrives at

$$
1 = Z + \sum_{|\mathbb{n}_0\rangle \neq |\mathbb{i}_\bullet\rangle} \lim_{\phi_\bullet \to \mathbb{i}_\bullet} \left| \psi_{\mathbb{n}_0,\mathbb{i}(\phi_\bullet)} \right|^2,
\tag{19}
$$

in self-explanatory notations. This equation just formalizes the fact that the probability of finding the scattering state $|\mathbb{i}\rangle$ in any Fock realization must be unity.

The "out" state $|\mathbb{f}\rangle$ can be treated similarly: it is derived from the regularized evolution operator $U_I^\epsilon(0,+\infty)$ acting on an eigenstate $|\mathbb{f}_\bullet\rangle$ of $H_0$. Its expansion in terms of Fock states will take a similar form to Eq. (17). Once one has worked out the expansion of the "in" and "out" state vectors, the scattering amplitude is simply given by their inner product (2).

---

[5]This was shown most generally in Ref. [43]. The main point of the latter reference was to prove that the limiting state,

$$
\lim_{\epsilon \to 0} \frac{U_I^\epsilon(0,-\infty)\,|\mathbb{i}_\bullet\rangle}{\langle \mathbb{i}_\bullet | U_I^\epsilon(0,-\infty)|\mathbb{i}_\bullet\rangle},
$$

is an eigenstate of $H$, if the limit exists (Gell-Mann and Low's theorem).

Note that the terms of this expansion involve an increasing number of matrix elements of the interaction part of the Hamiltonian between on-shell states, as well as energy denominators. Peculiarities of time-ordered perturbation theory include that intermediate states are all on shell but energy is not conserved, at variance with a covariant formalism.

In the case of the scattering of a particle (hadron, electron, photon) off a large nucleus, which will be the class of physical processes of interest to us, it is convenient to go to the restframe of the nucleus, and view the latter as the source of a classical field. The relevant problem to address is then the scattering of physical particles (subject to QED and QCD interactions) off an external field. Let us formulate most generally this problem, along the lines of Refs. [37, 38].

### 2.1.2 Scattering of quantum particles off an external classical field

We keep denoting by $H$ the time-independent Hamiltonian ruling the dynamics of the particles in the absence of the external field, and write the full Hamiltonian as $H + V$, where $V$ contains all interaction terms with the external field. We introduce another interaction picture, similar to the one worked out in the previous section, in which the states are related to the ones in the Schrödinger picture through

$$|\psi(\tau)\rangle_{I'} = e^{iH\tau}|\psi(\tau)\rangle_S. \tag{20}$$

The Schrödinger equation reads, in this representation,

$$i\frac{d}{d\tau}|\psi(\tau)\rangle_{I'} = V_{I'}(\tau)|\psi(\tau)\rangle_{I'}, \qquad \text{where} \qquad V_{I'}(\tau) = e^{iH\tau}Ve^{-iH\tau}, \tag{21}$$

and is solved by an evolution operator similar to $U_I$ defined in Eq. (10):

$$|\psi(\tau)\rangle_{I'} = U_{I'}(\tau, \tau_0)|\psi(\tau_0)\rangle_{I'}, \qquad \text{where} \qquad U_{I'}(\tau, \tau_0) \equiv T\exp\left(-i\int_{\tau_0}^{\tau} d\tau' V_{I'}(\tau')\right). \tag{22}$$

Hence in this picture, the states evolve under the action of the external potential $V$.

The initial and final states, in which the system is found at the asymptotic times $\tau = -\infty$ and $\tau = +\infty$ respectively, are eventually thought of as wave packets localized at spatial infinity. We shall assume that the external field is present only in a limited region of space around the origin of our reference frame. Then $V_{I'}$ does not perturb the asymptotic states of the incoming and outgoing particles. Thus in this context, scattering amplitudes of quantum particles off the external field are extracted from the matrix elements of the evolution operator $U_{I'}$ in a basis of eigenstates of the Hamiltonian $H$, and taking the infinite-time limits does not pose a problem:

$$S_{\mathbb{fi}} = \lim_{\tau \to +\infty} \lim_{\tau_0 \to -\infty} \langle\mathbb{f}| U_{I'}(\tau, \tau_0) |\mathbb{i}\rangle, \tag{23}$$

where $|\mathbb{i}\rangle$ and $|\mathbb{f}\rangle$ are the "in" and "out" states discussed in the previous section. Expanding the latter on the Fock states, we arrive at the following formula for the $S$-matrix element $S_{\mathbb{fi}}$ defined in Eq. (23):

$$S_{\mathbb{fi}} = \sum_{|\mathbb{m}_0\rangle,|\mathbb{n}_0\rangle} \psi^*_{\mathbb{m}_0,\mathbb{f}} \langle\mathbb{m}_0| U_{I'}(+\infty, -\infty) |\mathbb{n}_0\rangle \psi_{\mathbb{n}_0,\mathbb{i}}, \tag{24}$$

in terms of the wave functions (18) and of the matrix elements of the interaction picture evolution operators defined in Eq. (22).

## 2.2 Evaluation of the $S$-matrix in the high-energy limit

We are going to show that in the case of scattering at very high energies, that is if the states $|\mathbb{i}\rangle$ and $|\mathbb{f}\rangle$ are that of highly boosted particles in the rest frame of the source of the external field, then the interaction with the latter, encoded in the operator $U_{I'}(+\infty,-\infty)$, becomes simple, and in particular independent of the rapidity.

To this aim, we give to $|\mathbb{i}\rangle$ and $|\mathbb{f}\rangle$ the further boost of rapidity $Y$ along the direction $(Oz)$ of the incoming beam, thus increasing the center-of-mass energy of the process, while remaining in the rest frame of the source of the external field. This is realized through the substitutions

$$|\mathbb{i}\rangle \to e^{-iYK_3}|\mathbb{i}\rangle\,, \qquad \text{and} \qquad |\mathbb{f}\rangle \to e^{-iYK_3}|\mathbb{f}\rangle\,, \tag{25}$$

of the initial and final states in Eq. (23). Here, $K_3$ is the generator of boosts along the $(Oz)$ direction in a unitary representation of the Poincaré group on the Hilbert space of the states of the incoming particles. We then write $S_{\mathbb{f}\mathbb{i}}$ as

$$S_{\mathbb{f}\mathbb{i}} = \langle\mathbb{f}|\,\mathsf{F}(Y)\,|\mathbb{i}\rangle\,, \qquad \text{where} \qquad \mathsf{F}(Y) \equiv e^{+iYK_3}\,T\exp\left(-i\int d\tau\,V_{I'}(\tau)\right)e^{-iYK_3}\,. \tag{26}$$

The operator $\mathsf{F}$ is just the evolution operator due to the interaction with the external field, boosted by $-Y$. Time ordering being unaffected by boosts, we may rewrite $\mathsf{F}$ as

$$\mathsf{F}(Y) \equiv T\exp\left(-i\int d\tau\,e^{+iYK_3}V_{I'}(\tau)e^{-iYK_3}\right)\,. \tag{27}$$

To go further, we need to specify the interaction operator $V_{I'}(\tau)$, which describes the coupling to the external potential $\mathcal{A}$. Most generally, with the only a priori assumption that the latter is a vector potential,

$$V_{I'}(\tau) = g\int d^3\vec{x}\,J_a^\mu(\tau,\vec{x})\mathcal{A}_\mu^a(\tau,\vec{x})\,, \tag{28}$$

where $g$ is the coupling constant. $J$ is a vector operator, which will be obtained from the quantization of the Noether current deduced from the invariance under the gauge symmetry group of interest in this review; see Sec. 2.3. In this section, we shall keep it still general.

**Light cone coordinates.** For ultrarelativistic problems, in which the energy of the particles almost coincides with their momentum, it is convenient to use light cone coordinates, defined as follows [36, 44]. To the Lorentz components $v^\mu$ of a generic 4-vector $v$ we associate

$$v^+ = \frac{v^0+v^3}{\sqrt{2}}\,, \qquad v^- = \frac{v^0-v^3}{\sqrt{2}}\,, \qquad v^\perp = \begin{pmatrix} v^1 \\ v^2 \end{pmatrix}\,. \tag{29}$$

The Minkowskian inner product of two vectors $v$ and $w$, $v\cdot w = v^0w^0-v^1w^1-v^2w^2-v^3w^3$ is written in components in these non-Lorentz coordinates, as

$$v\cdot w = v^+w^- + v^-w^+ - v^\perp\cdot w^\perp\,, \qquad \text{where} \qquad v^\perp\cdot w^\perp = v^1w^1+v^2w^2\,. \tag{30}$$

The components of the metric tensor in the basis $(+,-,\perp)$ can be read off this formula:

$$\begin{pmatrix} 0 & 1 & 0 & 0 \\ 1 & 0 & 0 & 0 \\ 0 & 0 & -1 & 0 \\ 0 & 0 & 0 & -1 \end{pmatrix}\,. \tag{31}$$

Note, in particular, that $v^{\pm} = v_{\mp}$. The fact that there is no nontrivial numerical factor in this identity is an advantage of the particular definition (29).

For a particle moving close to the speed of light along the positive $(Oz)$-axis, it is natural to interpret the "+" component of its position 4-vector $x$ as a time coordinate, that we will call the light cone time. Then, the conjugate variable in the Minkowskian product $x \cdot p$ is the "−" component of the energy-momentum four-vector $p$, and therefore, we naturally identify it to the light cone energy. In the case of a real particle of mass $m$, the mass-shell condition leads to the following expression for the energy:

$$p^- = \frac{p^{\perp 2} + m^2}{2p^+} \,. \tag{32}$$

This form is reminiscent of the expression of the non-relativistic kinetic energy $E = \vec{p}^2/2m$ of a particle of 3-momentum $\vec{p}$ and mass $m$.

In the following, momentum 3-vectors $\vec{p}$ will be understood to have components $(p^+, p^\perp)$. Position 3-vectors $\vec{x}$ will have components $(x^-, x^\perp)$.

**Eikonal scattering.** We are now ready to express the effect of the inverse boost along the $(Oz)$-axis with rapidity $Y$ on $V_{I'}$, that appears in Eq. (27). The external potential $\mathcal{A}$ remains unaffected by this transformation. Physically, this arises because in this context, the boost is not a change of reference frame but an active transformation intended to increase the rapidity of the projectile relative to the source of the classical field by $Y$. From a technical perspective, the operator $K_3$ acts on the Hilbert space associated with the quantum states of the incoming particles, not on the function $\mathcal{A}$. Equation (27), supplemented by Eq. (28), becomes

$$\mathsf{F}(Y) \equiv T \exp\left[-ig \int dx^+ d^3\vec{x} \left(e^{+iYK_3} J_a^\mu(x^+, \vec{x}) e^{-iYK_3}\right) \mathcal{A}_\mu^a(x^+, \vec{x})\right], \tag{33}$$

where we have set $\tau$ to the light cone time $x^+$.

Since the current is a vector operator, by definition, it transforms as

$$J_a^\mu(x) \mapsto e^{+iYK_3} J_a^\mu(x) e^{-iYK_3} = \Lambda^\mu{}_\nu(Y) J_a^\nu[\Lambda^{-1}(Y)x], \tag{34}$$

where $\Lambda(Y)$ is the SO(1,3) matrix representing the active boost of rapidity $Y$ along the direction $(Oz)$ (see e.g. Ref. [45]). From the explicit Lorentz-frame expression of the boost as a function of the rapidity, we derive the transformation rules of the light cone coordinates of the generic vector $v$:

$$v^+ \mapsto e^{+Y} v^+, \qquad v^- \mapsto e^{-Y} v^-, \qquad v^\perp \mapsto v^\perp. \tag{35}$$

In particular, this boost communicates to an initial particle of mass $M$ that was (almost) at rest before the boost, the momentum $p_\bullet^\mu$ such that

$$p_\bullet^+ = \frac{M}{\sqrt{2}} e^{+Y}, \qquad p_\bullet^- = \frac{M}{\sqrt{2}} e^{-Y}, \qquad p_\bullet^\perp = 0^\perp. \tag{36}$$

Inserting the transformations (35) into Eq. (34), we find that under the boost, the components of the current transform as

$$
\begin{aligned}
J_a^+(x) &\mapsto e^{+Y} J_a^+(e^{-Y} x^+, e^{+Y} x^-, x^\perp), \\
J_a^-(x) &\mapsto e^{-Y} J_a^-(e^{-Y} x^+, e^{+Y} x^-, x^\perp), \\
J_a^\perp(x) &\mapsto J_a^\perp(e^{-Y} x^+, e^{+Y} x^-, x^\perp).
\end{aligned} \tag{37}
$$

Hence, the term $J_a^+ \mathcal{A}_a^-$ parametrically dominates the Minkowskian product $J_a^\mu \mathcal{A}_\mu^a$, being exponentially enhanced at large rapidities. Therefore, the exponent in Eq. (33) becomes

$$g \int dx^+ d^3 \vec{x} \left( e^{+iYK_3} J_a^\mu(x^+, \vec{x}) e^{-iYK_3} \right) \mathcal{A}_\mu^a(x^+, \vec{x})$$

$$\simeq g \int dx^+ dx^- d^2 x^\perp e^Y J_a^+(e^{-Y}x^+, e^Y x^-, x^\perp) \mathcal{A}_a^-(x^+, x^-, x^\perp). \tag{38}$$

Next, we perform the change of integration variable $x^- \mapsto e^{-Y}x^-$. The term in the right-hand side becomes

$$g \int dx^+ dx^- d^2 x^\perp J_a^+(e^{-Y}x^+, x^-, x^\perp) \mathcal{A}_a^-(x^+, e^{-Y}x^-, x^\perp). \tag{39}$$

In the limit of $Y \to +\infty$, we see that $J_a^+$ becomes localized in $x^+$ at $x^+ = 0$, and $\mathcal{A}_a^-$ in $x^-$ at $x^- = 0$, provided the integrand $J_a^+ \mathcal{A}_a^-$ possesses enough mathematical regularities. At the very least, it must be integrable, as well as its derivatives. In practice, we may assume that the potential is non-zero only in a finite region of space. Defining

$$\rho_a(x^\perp) = \int dx^- J_a^+(0, x^-, x^\perp), \qquad \text{and} \qquad \chi_a(x^\perp) = \int dx^+ \mathcal{A}_a^-(x^+, 0, x^\perp), \tag{40}$$

the $S$-matrix element reads

$$S_{\mathbb{fi}} = \langle \mathbb{f} | \mathsf{F} | \mathbb{i} \rangle, \qquad \text{with} \qquad \mathsf{F} = T \, e^{-ig \int d^2 x^\perp \rho_a(x^\perp) \chi_a(x^\perp)}. \tag{41}$$

As stated previously, the $\mathsf{F}$ operator is independent of the boost rapidity, up to terms which are exponentially suppressed with $Y$, as $Y \to +\infty$. We may then relate the physical states to partonic states as in the previous section, to get

$$S_{\mathbb{fi}} = \sum_{|\mathbb{m}_0\rangle, |\mathbb{n}_0\rangle} \psi^*_{\mathbb{m}_0, \mathbb{f}} \langle \mathbb{m}_0 | \mathsf{F} | \mathbb{n}_0 \rangle \psi_{\mathbb{n}_0, \mathbb{i}}. \tag{42}$$

This is the same formula as (24), except that the matrix element describing the scattering of the Fock states off the external field most generally has been replaced by its high-energy limit, defined in Eq. (41). This step is called the *eikonal approximation*.

Looking at the form of the matrix elements of $\mathsf{F}$, we see that it consists of a series of interactions between the incoming on-shell partons and the external field, through a particularly simple coupling of the "+" component of the current of the partons and the "−" component of the vector potential $\mathcal{A}$ of the field. In these interactions, the current of the partons is evaluated at $x^+ = 0$, and thus, the field is "seen" as perfectly localized in light-cone time. This is due to the infinite Lorentz contraction in the frame of the partons. This fact will have several important consequences. One of them is that there is no change in the parton content while crossing the region where the external field is non-zero. We will investigate the other consequences below, after having completely specified the current $J$.

## 2.3 High-energy scattering in non-Abelian gauge theories

We now specialize to non-Abelian gauge theories in order to be able to arrive at explicit expressions for $S_{\mathbb{fi}}$. We assume the Lagrangian, and canonically quantize the theory at equal light cone times. Then, we compute the color Noether current appearing in the eikonal factor $\mathsf{F}$ defined in Eq. (41). Finally, we present the matrix elements of the interaction Hamiltonian that will be needed for our calculations of the wave functions through Eqs. (18),(17).

We shall develop the formalism for a non-Abelian gauge theory, since our main goal is QCD, but it may of course be readily transposed to quantum electrodynamics (QED) through a few straightforward substitutions.

We emphasize once again that our goal is not to provide a fully rigorous construction of field theory quantization on the light cone in all its intricacies. Instead, we will offer a simplified exposition, aimed at readers familiar with quantization in a Lorentz frame. We will focus on providing the practical tools necessary to compute the relevant objects which appear in formula (42), namely matrix elements of the eikonal operator F and light cone wave functions. We refer readers who wish to explore the topic further to the original paper in Ref. [37] or to a classical review such as Ref. [31].

### 2.3.1 Light cone quantization

**Lagrangian.** We consider a theory endowed with the gauge symmetry group SU($N$). Its Lie algebra $\mathfrak{su}(N)$ is the set of Hermitian traceless matrices of dimension $N \times N$. Let us denote by $\{t^a\}$ (with $a = 1, \cdots, N^2 - 1$) a basis of this Lie algebra, and introduce the structure constants $f^{abc}$ in such a way that the Lie brackets of the basis vectors read

$$[t^a, t^b] = i f^{abc} t^c. \tag{43}$$

We normalize these matrices such that $\text{Tr}\left(t^a t^b\right) = \frac{1}{2}\delta^{ab}$. We limit ourselves to one generic fermion/antifermion of mass $m$. Since it can come in $N$ different colors, it is described by $N$ Dirac fields $\Psi_i(x)$, $i \in \{1, \cdots, N\}$. The gauge fields are represented by a $\mathfrak{su}(N)$-valued vector field: we shall denote its components on the basis $\hat{e}_\mu$ of the Minkowski space and $t^a$ of $\mathfrak{su}(N)$ as $A_a^\mu(x)$.

In the case of QCD, $N = 3$, and $t^a$ are (up to a normalization) the Gell-Mann matrices. One may also take over the formalism developed here to QED: it is enough to formally set $N \to 1$, $f^{abc} \to 0$, $t^a \to 1$, and replace $g$ by the electric charge of the fermion in all formulas. Even for general $N$, we shall call the fermions "quarks", and the bosons "gluons".

A comment on the notations is in order. With the exception of Appendix A.3, we will not attribute any meaning to the position (up or down) of the indices that label the components of tensors in the different representations of the color group: we will put them wherever it makes the formulas easier to read. In particular, the distinction between a representation and its conjugate will be understood from the context. We will apply the same to the indices that label the components of the Dirac spinors. When a given index is repeated twice, a summation is understood, unless otherwise stated.

The Lagrangian density can be decomposed in two terms. The Yang-Mills (YM) part describes the dynamics of the gauge field, and the Dirac (D) part rules the fermions and their coupling to the former:

$$\mathcal{L} = \mathcal{L}_{\text{YM}} + \mathcal{L}_{\text{D}}, \quad \text{where} \quad \mathcal{L}_{\text{YM}} = -\frac{1}{4} F_{\mu\nu}^a F_a^{\mu\nu}, \quad \text{and} \quad \mathcal{L}_{\text{D}} = \bar{\Psi}_i \left( i\gamma^\mu D_{\mu,ij} - m\delta_{ij} \right) \Psi_j. \tag{44}$$

The field-strength tensor reads, in components on the basis $\{t^a\}$ of the Lie algebra,

$$F_{\mu\nu}^a = \partial_\mu A_\nu^a - \partial_\nu A_\mu^a + g f^{abc} A_\mu^b A_\nu^c, \tag{45}$$

and the covariant derivative reads

$$D_{\mu,ij} = \delta_{ij}\partial_\mu - ig\,(t^a)_{ij} A_\mu^a. \tag{46}$$

This Lagrangian is invariant under the SU($N$) gauge transformation, that maps the fields as

$$\Psi(x) \mapsto \Omega(x)\Psi(x),$$
$$A_\mu(x) \mapsto \Omega(x)A_\mu(x)\Omega(x)^{-1} - \frac{i}{g}\partial_\mu\Omega(x)\Omega(x)^{-1}, \tag{47}$$
$$F_{\mu\nu}(x) \mapsto \Omega(x)F_{\mu\nu}(x)\Omega(x)^{-1},$$

where $A_\mu = A_\mu^a t^a$, $F_{\mu\nu} = F_{\mu\nu}^a t^a$, while $\Omega(x) = e^{-i\theta^a(x)t^a}$ is a generic SU($N$) matrix parametrized by the $N^2 - 1$ real-valued fields $\theta^a(x)$.

Before proceeding, let us clarify the terminology we will use in what follows. Consider the set of matrices $\{\Omega = e^{-i\theta^a t^a}, \theta^a \in \mathbb{R}\}$ under which the $\Psi$'s transform. This set defines an $N$-dimensional representation of SU($N$), which we shall call the *fundamental representation*.[6] Its infinitesimal generators are the matrices $T_F^a \equiv t^a$. The complex conjugate matrices $\{\Omega^*\}$ define another $N$-dimensional representation (inequivalent to the latter when $N \geq 3$), referred to as *anti-fundamental representation*, with generators $T_{\bar{F}}^a \equiv -(t^a)^T$. Finally, the transformation law of $F_{\mu\nu}$ defines the *adjoint representation*. This representation is generated by $(N^2-1) \times (N^2-1)$-dimensional matrices $T_A^a$, whose matrix element $(b, c)$ is given by $(T_A^a)^{bc} = -if^{abc}$, where $f^{abc}$ are the structure constants of SU($N$).

In the following, we will use a light cone gauge condition

$$\eta \cdot A = 0, \tag{48}$$

where $\eta$ is a light-like four-vector ($\eta^2 = 0$), that we choose as

$$\eta^+ = 0, \qquad \eta^- = 1, \qquad \eta^\perp = 0^\perp. \tag{49}$$

This implies $A^+ = 0$. This gauge will help to simplify the calculations in the light cone coordinate system used here.

It will prove useful to introduce the operators

$$\Lambda_\pm \equiv \frac{\gamma^0 \gamma^\pm}{\sqrt{2}} = \frac{\gamma^\mp \gamma^\pm}{2}, \tag{50}$$

which are Hermitian orthogonal projectors onto two-dimensional supplementary subspaces of the Dirac spinor representation space. They possess the following properties:

$$\Lambda_\pm^\dagger = \Lambda_\pm, \qquad \Lambda_\pm^2 = \Lambda_\pm, \qquad \Lambda_\pm\Lambda_\mp = 0, \quad \text{and} \quad \Lambda_+ + \Lambda_- = \mathbb{I}_{4\times4}. \tag{51}$$

We use $\Lambda_\pm$ to decompose the spinors as

$$\Psi = \Psi_+ + \Psi_-, \qquad \text{with} \qquad \Psi_+ \equiv \Lambda_+\Psi, \quad \text{and} \quad \Psi_- \equiv \Lambda_-\Psi. \tag{52}$$

**Mode expansion of the dynamical fields.** In view of the canonical quantization, we determine the fields conjugate to $\Psi$ and $A$. The conjugates to the two-component fields $\Psi_{i+}$ and $A_c^\perp$ read

$$\Pi_{i,\alpha}^{(\Psi_+)} = \frac{\partial\mathcal{L}}{\partial(\partial_+\Psi_{i+,\alpha})} = i\sqrt{2}\Psi_{i+,\alpha}^\dagger, \qquad \text{and} \qquad \Pi_c^{(A^\perp)\perp l} = \frac{\partial\mathcal{L}}{\partial(\partial_+A_c^{\perp l})} = \partial^+A_c^{\perp l}, \tag{53}$$

where $\alpha$ labels the spinorial components and $l = 1, 2$ the components of the transverse part of four-vectors. As for the fields $\Psi_{i-}$ and $A_c^- = A_+^c$, there is no term in the Lagrangian involving

---

[6]This representation whose operators are the SU($N$) matrices themselves is sometimes called "defining" or "standard" (see e.g. Ref. [46]).

their light cone time derivative, meaning that they are non-dynamical. They do not have non-zero conjugates. Their equations of motion are constraint equations, which can be used to express them in terms of the independent fields $\Psi_{i+}$ and $A_c^\perp$. The relevant Euler-Lagrange equations read

$$\partial^+ \frac{\partial \mathcal{L}}{\partial \partial^+ A_+^a} + \partial^\perp \frac{\partial \mathcal{L}}{\partial \partial^\perp A_+^a} = \frac{\partial \mathcal{L}}{\partial A_+^a}, \qquad \text{and} \qquad \frac{\partial \mathcal{L}}{\partial \Psi_-^\dagger} = 0. \tag{54}$$

Their formal solutions read, respectively,

$$A_+^a = \frac{1}{\partial_-} \partial^\perp \cdot A_a^\perp - g f^{abc} \frac{1}{\partial_-^2} \partial_- A_b^\perp \cdot A_c^\perp + \sqrt{2} g \frac{1}{\partial_-^2} \Psi_+^\dagger t^a \Psi_+,$$

$$\Psi_- = -\frac{i}{\sqrt{2}} \frac{1}{\partial_-} \gamma^0 \left( -i\gamma^j D_j + m \right) \Psi_+. \tag{55}$$

The "inverse derivative" operator $1/\partial_- = 1/\partial^+$ represents integration over the spatial variable $x^- = x_+$. We define its action on a sufficiently well-behaved function $f$ as follows:

$$\left[ \frac{1}{\partial_-} f \right](x^+, x^-, x^\perp) = \frac{1}{2} \int_{-\infty}^{+\infty} d\xi \, \mathrm{sgn}(x^- - \xi) f(x^+, \xi, x^\perp) \equiv F(x^+, x^-, x^\perp), \tag{56}$$

where the boundary conditions are chosen as

$$\lim_{x^- \to -\infty} F(x^+, x^-, x^\perp) = - \lim_{x^- \to +\infty} F(x^+, x^-, x^\perp), \tag{57}$$

as discussed in Refs. [37, 38]. In the same way, the operator $1/\partial_-^2$, representing a double integration over $x^-$, acts as follows:

$$\left[ \frac{1}{\partial_-^2} f \right](x^+, x^-, x^\perp) = \frac{1}{2} \int_{-\infty}^{+\infty} d\xi \, |x^- - \xi| f(x^+, \xi, x^\perp). \tag{58}$$

The contraction "$-\gamma^j D_j$" may also be written "$+\gamma^\perp \cdot D^\perp$", but this last notation might be misleading in the context in which we will use this formula. Equations (55) relate explicitly the dependent fields to the independent ones at any light cone time.

Let us expand the dynamical, independent, fields in modes at some given light cone time $\tau = x^+$. We write the "+" component of the fermion field as

$$\Psi_{i+}(x^+, \vec{x}) = 2^{1/4} \int \frac{d^2 p^\perp}{(2\pi)^3} \int_0^{+\infty} \frac{dp^+}{2p^+} \sum_{\sigma = \pm 1/2} \left[ \sqrt{p^+} w_{(\sigma)} b_i^{(\sigma)}(x^+, \vec{p}) e^{ip^\perp \cdot x^\perp - ip^+ x^-} \right.$$

$$\left. + \sqrt{p^+} w_{(-\sigma)} d_i^{(\sigma)\dagger}(x^+, \vec{p}) e^{-ip^\perp \cdot x^\perp + ip^+ x^-} \right], \tag{59}$$

where $w_{(1/2)}$ and $w_{(-1/2)}$ are Dirac spinors that form a complete basis of the image space of the projector $\Lambda_+$. They are chosen to satisfy the completeness and orthogonality relations

$$\sum_{\sigma = \pm 1/2} w_{(\sigma)} w_{(\sigma)}^\dagger = \Lambda_+, \qquad w_{(\sigma)}^\dagger w_{(\sigma')} = \delta_{\sigma \sigma'}. \tag{60}$$

We also ask that they be eigenstates of the spin matrix corresponding to a measurement along the $z$-axis with eigenvalues $\sigma = \pm \frac{1}{2}$, namely

$$\frac{i}{2} \gamma^1 \gamma^2 w_{(\sigma)} = \sigma w_{(\sigma)}. \tag{61}$$

See Appendix A.2 for a more detailed discussion of the choice of this basis.

As for the two-dimensional transverse gauge potential, we decompose it as

$$
A_c^\perp(x^+, \vec{x}) = \int \frac{d^2 k^\perp}{(2\pi)^3} \int_0^{+\infty} \frac{dk^+}{2k^+} \sum_{\lambda=\pm 1} \left[ \varepsilon_{(\lambda)}^\perp a_c^{(\lambda)}(x^+, \vec{k}) e^{ik^\perp \cdot x^\perp - ik^+ x^-} \right.
$$
$$
\left. + \varepsilon_{(\lambda)}^{\perp *} a_c^{(\lambda)\dagger}(x^+, \vec{k}) e^{-ik^\perp \cdot x^\perp + ik^+ x^-} \right], \tag{62}
$$

where the two-dimensional polarization vectors $\varepsilon_{(\lambda)}^\perp = (\varepsilon_{(\lambda)}^1, \varepsilon_{(\lambda)}^2)$ (with $\lambda = \pm 1$) are eigenstates of the spin-projection operator on the $z$-axis, with respective eigenvalues $\lambda$, that obey the following completeness and orthogonality relations:

$$
\sum_{\lambda=\pm 1} \varepsilon_{(\lambda)}^{\perp l} \varepsilon_{(\lambda)}^{\perp l' *} = \delta^{ll'}, \qquad \varepsilon_{(\lambda)}^\perp \cdot \varepsilon_{(\lambda')}^{\perp *} = \delta_{\lambda \lambda'}. \tag{63}
$$

**Quantization.** Let us introduce the notations

$$
a_c^{(\lambda)}(\vec{k}) \equiv a_c^{(\lambda)}(x^+ = 0, \vec{k}), \qquad b_i^{(\lambda)}(\vec{p}) \equiv b_i^{(\lambda)}(x^+ = 0, \vec{p}), \qquad d_i^{(\lambda)}(\vec{p}) \equiv d_i^{(\lambda)}(x^+ = 0, \vec{p}), \tag{64}
$$

for the coefficients in the mode expansions (59),(62) of the fields evaluated at zero light cone time. We quantize canonically the theory promoting the latter to creation/annihilation operators of single-particle excitations. We impose commutation (resp. anticommutation) relations between the bosonic (resp. fermionic) creation and annihilation operators. The non-zero commutators of the boson creation/annihilation operators are set to[7]

$$
\left[ a_c^{(\lambda)}(\vec{p}), a_{c'}^{(\lambda')\dagger}(\vec{p}') \right] = (2\pi)^3 2p^+ \delta^3(\vec{p} - \vec{p}') \delta^{\lambda\lambda'} \delta_{cc'}. \tag{65}
$$

The non-zero anticommutators of the fermionic ones are chosen as

$$
\left\{ b_i^{(\sigma)}(\vec{p}), b_{i'}^{(\sigma')\dagger}(\vec{p}') \right\} = \left\{ d_i^{(\sigma)}(\vec{p}), d_{i'}^{(\sigma')\dagger}(\vec{p}') \right\} = (2\pi)^3 2p^+ \delta^3(\vec{p} - \vec{p}') \delta^{\sigma\sigma'} \delta_{ii'}. \tag{66}
$$

The fermionic operators are postulated to commute with the bosonic operators.

We shall now work out the light cone time dependence of the ladder operators. In the interaction picture, the latter evolve as prescribed in Eq. (6), which involves the free Hamiltonian $H_0$.

The Hamiltonian is obtained from the Lagrangian (44) through a Legendre transformation:

$$
H = \int d^3 \vec{x} \left( \sum_c \Pi_c^{(A^\perp)\perp} \cdot \partial_+ A_c^\perp + \sum_i \Pi_{i,\alpha}^{(\Psi_+)} \partial_+ \Psi_{+i,\alpha} - \mathcal{L} \right), \tag{67}
$$

where $\mathcal{L}$ needs to be expressed in terms of the dynamical degrees of freedom with the help of the constraint equations [Eq. (44), with $A^-$ and $\Psi_-$ being replaced by (55)]. At this stage, we only need the free part of $H$. Considering the Lagrangian with the coupling constant $g$ set to 0, we obtain

$$
H_0 = \int d^3 \vec{x} \left( \sum_c A_c^{\perp l} \frac{(i\partial^\perp)^2}{2} A_c^{\perp l} + \sum_i \sqrt{2} \Psi_{i+}^\dagger \frac{(i\partial^\perp)^2 + m^2}{2i\partial^+} \Psi_{i+} \right), \tag{68}
$$

---

[7]Note that we have picked relativistic-invariant normalizations: $2p^+ \delta(p^+ - p'^+)$ is indeed obviously invariant under boosts along the $z$ direction.

and in terms of the ladder operators,

$$H_0 = \int \frac{d^2k^\perp}{(2\pi)^3} \int_0^{+\infty} \frac{dk^+}{2k^+} \sum_{\lambda,c} \frac{k^{\perp 2}}{2k^+} \frac{1}{2} \left[ a_c^{(\lambda)}(x^+,\vec{k}) a_c^{(\lambda)\dagger}(x^+,\vec{k}) + a_c^{(\lambda)\dagger}(x^+,\vec{k}) a_c^{(\lambda)}(x^+,\vec{k}) \right] \tag{69}$$

$$+ \int \frac{d^2p^\perp}{(2\pi)^3} \int_0^{+\infty} \frac{dp^+}{2p^+} \sum_{\sigma,i} \frac{p^{\perp 2}+m^2}{2p^+} \left[ b_i^{(\sigma)\dagger}(x^+,\vec{p}) b_i^{(\sigma)}(x^+,\vec{p}) - d_i^{(\sigma)}(x^+,\vec{p}) d_i^{(\sigma)\dagger}(x^+,\vec{p}) \right].$$

The commutators of $H_0$ and of the $a$, $b$, $d$ operators is readily evaluated, using multiply the (anti-)commutation relations (65),(66):

$$[a_c^{(\lambda)}(x^+,\vec{k}), H_0] = k^- a_c^{(\lambda)}(x^+,\vec{k}),$$
$$[b_i^{(\sigma)}(x^+,\vec{p}), H_0] = p^- b_i^{(\sigma)}(x^+,\vec{p}), \tag{70}$$
$$[d_i^{(\sigma)}(x^+,\vec{p}), H_0] = p^- d_i^{(\sigma)}(x^+,\vec{p}),$$

where $k^- = k^{\perp 2}/(2k^+)$ and $p^- = (p^{\perp 2}+m^2)/(2p^+)$. By integrating Eq. (6), one obtains

$$a_c^{(\lambda)}(x^+,\vec{k}) = e^{-ix^+k^-} a_c^{(\lambda)}(\vec{k}),$$
$$b_i^{(\sigma)}(x^+,\vec{p}) = e^{-ix^+p^-} b_i^{(\sigma)}(\vec{p}), \tag{71}$$
$$d_i^{(\sigma)}(x^+,\vec{p}) = e^{-ix^+p^-} d_i^{(\sigma)}(\vec{p}).$$

The time-dependence of the creation operators is of course obtained by taking the Hermitian conjugates of these relations.

As a check of the normalization of the fields, we verify by direct computation that the equal-light-cone-time commutation and anti-commutation relations for the dynamical fields and their conjugates have the canonical form:

$$\left[ A_c^{\perp l}(x^+,\vec{x}), \Pi_{c'}^{(A^\perp)\perp l'}(x^+,\vec{x}') \right] = \frac{i}{2} \delta^{ll'} \delta^3(\vec{x}-\vec{x}') \delta_{cc'},$$
$$\left\{ \Psi_{i+,\alpha}(x^+,\vec{x}), \Pi_{i',\alpha'}^{(\Psi_+)}(x^+,\vec{x}') \right\} = i (\Lambda_+)_{\alpha\alpha'} \delta^3(\vec{x}-\vec{x}') \delta_{ii'}. \tag{72}$$

**One-particle states.** The one-particle on-shell quark, antiquark, gluon states of definite momentum, helicity and color are created from the vacuum as

$$|q;\vec{p},\sigma,i\rangle = b_i^{(\sigma)\dagger}(\vec{p})|0\rangle, \qquad |\bar{q};\vec{p},\sigma,i\rangle = d_i^{(\sigma)\dagger}(\vec{p})|0\rangle, \qquad |g;\vec{k},\lambda,c\rangle = a_c^{(\lambda)\dagger}(\vec{k})|0\rangle. \tag{73}$$

We will drop some of the labels when there is no ambiguity, in order to simplify the notations. We note that our choices of normalization obviously imply the following orthogonality relations for, say, single quark states:

$$\langle q;\vec{p},\sigma,i | q;\vec{p}',\sigma',i' \rangle = (2\pi)^3 2p^+ \delta^3(\vec{p}-\vec{p}') \delta_{\sigma\sigma'} \delta_{ii'}. \tag{74}$$

Let us also write explicitly the completeness relation in the single-quark Hilbert space:

$$\sum_{\sigma;i} \int \frac{d^2p^\perp}{(2\pi)^3} \int_0^{+\infty} \frac{dp^+}{2p^+} |q;\vec{p},\sigma,i\rangle \langle q;\vec{p},\sigma,i| = \mathbb{I}. \tag{75}$$

The same applies to antiquarks and gluons, with the appropriate substitution of the discrete quantum numbers. This also extends to multi-particle Hilbert spaces by taking the corresponding tensor products, which we will introduce later, at the point where they are needed.

### 2.3.2 Eikonal scattering operator

The Noether color current associated to the invariance under global SU($N$) transformations can be derived from the Lagrangian as

$$J_a^\mu = \frac{\partial \mathcal{L}}{\partial(\partial_\mu \Psi_{i,\alpha})} \frac{\delta \Psi_{i,\alpha}}{\delta \theta^a} + \frac{\partial \mathcal{L}}{\partial(\partial_\mu A_\nu^b)} \frac{\delta A_\nu^b}{\delta \theta^a}. \tag{76}$$

The partial derivatives of the Lagrangian read

$$\frac{\partial \mathcal{L}}{\partial(\partial_\mu \Psi_{i,\alpha})} = i \bar{\Psi}_{i,\beta} \gamma_{\beta\alpha}^\mu, \qquad \frac{\partial \mathcal{L}}{\partial(\partial_\mu A_\nu^b)} = -F_b^{\mu\nu}. \tag{77}$$

The variations of the fields in an infinitesimal gauge transformation are read off Eq. (47):

$$\frac{\delta \Psi_{i,\alpha}}{\delta \theta^a} = -i t^a \Psi_{i,\alpha}, \qquad \frac{\delta A_\nu^b}{\delta \theta^a} = f^{bac} A_\nu^c. \tag{78}$$

Hence, the explicit expression of the current in a generic Lorentz frame is

$$J_a^\mu = \bar{\Psi}_i \gamma^\mu (t^a)_{ij} \Psi_j + f^{abc} F_b^{\mu\nu} A_\nu^c. \tag{79}$$

Let's digress on the conservation of this current. It is instructive to check directly that $\partial_\mu J_a^\mu = 0$. The Dirac equation implies that the fermionic current $J_a^{(\Psi)\mu} \equiv \bar{\Psi}\gamma^\mu t^a \Psi$, that would be conserved in QED, is covariantly conserved in a non-Abelian gauge theory:

$$\left(D_\mu^A\right)^{ab} J_b^{(\Psi)\mu} = 0, \qquad \text{with} \qquad D_\mu^A \equiv \partial_\mu - i g T_A^c A_\mu^c, \quad \text{and} \quad \left(T_A^c\right)^{ab} \equiv -i f^{cab}. \tag{80}$$

On the other hand, the equations of motion of the gauge field read

$$\left(D_\mu^A\right)^{ab} F_b^{\mu\nu} = -g J_a^{(\Psi)\nu}. \tag{81}$$

Expressing $J^{(\Psi)}$ with the help of $J$ and $F$, and reshuffling the terms of this equation, one gets

$$\partial_\mu F_a^{\mu\nu} = -g J_a^\nu. \tag{82}$$

The conservation of $J_a^\nu$ then simply follows from the antisymmetry of the tensor $F^{\mu\nu}$.

The + component of this current is the only relevant one in the eikonal approximation [see Eq. (40)]. It involves only the dynamical $\Psi_+$ field, since the factor $\gamma^0 \gamma^+ = \sqrt{2}\Lambda_+$ appears in the fermionic part of the current, and the projectors $\Lambda_\pm$ satisfy $\Lambda_\pm^\dagger = \Lambda_\pm$. The gluonic current turns out to also have a simple form in the $A^+ = 0$ light cone gauge. All in all,

$$J_a^+ = \sqrt{2}\,\Psi_{i+}^\dagger (t^a)_{ij} \Psi_{j+} - f^{abc} \partial^+ A_b^\perp \cdot A_c^\perp. \tag{83}$$

Replacing the fields by their mode expansions (62),(59), we arrive at an expression for $J_a^+$ in terms of creation and annihilation operators, and eventually, after integration over the $x^-$ variable as prescribed in Eq. (40), at an expression for the operator $\rho_a$:

$$\rho_a(x^\perp) = \frac{1}{2\pi} \int \frac{d^2 p^\perp}{(2\pi)^2} \frac{d^2 p'^\perp}{(2\pi)^2} \int_0^{+\infty} \frac{dp^+}{2p^+} e^{-i(p^\perp - p'^\perp)\cdot x^\perp}$$

$$\times \left\{ \sum_{\sigma;ij} (t^a)_{ij} \left[ b_i^{(\sigma)\dagger}(p^\perp, p^+) b_j^{(\sigma)}(p'^\perp, p^+) - d_j^{(\sigma)\dagger}(p^\perp, p^+) d_i^{(\sigma)}(p'^\perp, p^+) \right] \right.$$

$$\left. + \sum_{\lambda;bc} (-i f^{abc}) a_b^{(\lambda)\dagger}(p^\perp, p^+) a_c^{(\lambda)}(p'^\perp, p^+) \right\}. \tag{84}$$

Although the ladder operators were not a priori normal-ordered in the current (83), we have arrived at a linear combination of the normal-ordered operators $b^\dagger b$, $d^\dagger d$, $a^\dagger a$ only. Technically, this is due to the fact that, on the one hand, the terms involving the anticommutators of the $b$, $b^\dagger$ and $d$, $d^\dagger$ operators that show up a priori in the calculation eventually cancel with each other. On the other hand, the commutator of the bosonic operators $a_c$, $a_b^\dagger$ gives a null contribution when contracted with the structure constants $f^{abc}$, due to the symmetry of the former and the antisymmetry of the latter under the exchange of the color indices $b$ and $c$.[8]

We see that the operator $\rho_a(x^\perp)$ annihilates quarks, antiquarks and gluons before recreating them *with the same "+" momentum and helicity*. The color instead undergoes a transformation generated by the basis vectors $T_R^a$ of the corresponding representations $R$ of the Lie algebra $\mathfrak{su}(N)$ of the color group:

$$\begin{cases} (T_{R=F}^a)_{ij} = (t^a)_{ij}\,, & \text{for a quark (fundamental representation),} \\ (T_{R=\bar{F}}^a)_{ij} = -(t^a)_{ji}\,, & \text{for an anti-quark (anti-fundamental representation),} \\ (T_{R=A}^a)^{bc} = -if^{abc}\,, & \text{for a gluon (adjoint representation).} \end{cases} \tag{85}$$

As of the transverse momentum, it is apparent in Eq. (84) that it gets shifted by a vector conjugate to $x^\perp$. This fact encourages us to localize the incoming particles in transverse position space.

**Mixed representation. Wilson lines.** A fermion state of definite transverse position amounts to the following superposition of states of definite momenta:

$$|q; x^\perp, p^+, \sigma, i\rangle = \int \frac{d^2 p^\perp}{(2\pi)^2} e^{-ix^\perp \cdot p^\perp} |q; p^\perp, p^+, \sigma, i\rangle\,. \tag{86}$$

This is the so-called "mixed representation". From Eq. (75), we deduce the completeness relation

$$\sum_{\sigma,i} \int d^2 x^\perp \int_0^{+\infty} \frac{dp^+}{2p^+(2\pi)} \sum_{\sigma,i} |q; x^\perp, p^+, \sigma, i\rangle \langle q; x^\perp, p^+, \sigma, i| = \mathbb{I}\,, \tag{87}$$

and the orthogonality relation

$$\langle q; x^\perp, p^+, \sigma, i | q; x'^\perp, p'^+, \sigma', i'\rangle = \delta^2(x^\perp - x'^\perp)(2\pi)2p^+ \delta(p^+ - p'^+)\delta_{\sigma\sigma'}\delta_{ii'}\,. \tag{88}$$

The operator $\rho_a$ acts on such states as

$$\rho_a(x'^\perp)|q; x^\perp, p^+, \sigma, i\rangle = \delta^2(x'^\perp - x^\perp) \sum_{i'} |q; x^\perp, p^+, \sigma, i'\rangle (t^a)_{i'i}\,. \tag{89}$$

From its expression (41), we see that the F operator applied to a quark state localized in transverse position space transforms it into the following superposition:

$$\mathsf{F}|q; x^\perp, p^+, \sigma, i\rangle = \sum_{i'} |q; x^\perp, p^+, \sigma, i'\rangle \left[W^F(x^\perp)\right]_{i'i}\,, \tag{90}$$

where $W^F(x^\perp)$ reads

$$W^F(x^\perp) \equiv T\, e^{-igt^a \chi_a(x^\perp)}\,. \tag{91}$$

The function $\chi_a(x^\perp)$ [see Eq. (40) for its definition] is an integral of the external potential, evaluated at coordinates $(x^+, x^- = 0, x^\perp)$, over the light cone time $x^+$. Hence the time ordering $T$ is actually a path ordering $P$ along the line parametrized by $x^+ \in (-\infty, +\infty)$ of constant $x^- = 0$ and $x^\perp$. $W^F$ is called a "Wilson line" (in the fundamental representation). It

---

[8]We thank F. Gelis for having brought this fact to our attention.

turns out to be a matrix in the SU($N$) group. This is not completely obvious: we show it in Appendix A.1.

If the operator $\rho_a(x'^\perp)$ acted on a single-antiquark state or on a single-gluon state, then Eq. (89) would still hold, up to the substitution of the infinitesimal generator $t^a = T_F^a$ by $T_{\bar{F}}^a$ and $T_A^a$ respectively. The Wilson line (91) would be replaced by the following one:

$$W^R(x^\perp) \equiv P\,e^{-ig T_R^a \chi_a(x^\perp)}\,, \tag{92}$$

where the $T_R^a$'s are defined in Eq. (85). We note that the matrix element of the fundamental and anti-fundamental Wilson lines are related by

$$\left[W^{\bar{F}}(x^\perp)\right]_{i'i} = \left[W^F(x^\perp)\right]_{ii'}^\dagger\,, \tag{93}$$

see again Appendix A.1 for a proof.

**Generalization to multi-particle states.** We shall now address multi-particle states. Let us consider an initial system made of $n$ quarks. From the definition (41) of F and the expression (84) of $\rho$ , F$|0\rangle = |0\rangle$, and we may write, for a generic multi-quark state,

$$\mathsf{F}(\underbrace{b^\dagger \cdots b^\dagger}_{n}|0\rangle) = (\mathsf{F}b^\dagger\mathsf{F}^{-1})\cdots(\mathsf{F}b^\dagger\mathsf{F}^{-1})|0\rangle\,, \tag{94}$$

where we have understood the labels of the creation operators. The action of F on a single ladder operator can be computed from the expression of F as a series of convolutions of $\rho$ factors, and using repeatedly the commutator $[\rho, b^\dagger]$ that can be evaluated with the help of the expression (84) of $\rho_a$ as a function of the ladder operators, and the canonical commutation relations (66):

$$[\rho_a(x^\perp), b_i^{(\sigma)\dagger}(p^\perp, p^+)] = \sum_{i'=1}^N \int \frac{d^2 p'^\perp}{(2\pi)^2} b_{i'}^{(\sigma)\dagger}(p'^\perp, p^+) e^{-ix^\perp \cdot (p'^\perp - p^\perp)}(t^a)_{i'i}\,. \tag{95}$$

We find

$$\mathsf{F}\,b_i^{(\sigma)\dagger}(p^\perp, p^+)\,\mathsf{F}^{-1} = \sum_{i'=1}^N \int \frac{d^2 p'^\perp}{(2\pi)^2} b_{i'}^{(\sigma)\dagger}(p'^\perp, p^+) \int d^2 x^\perp e^{-ix^\perp \cdot (p'^\perp - p^\perp)}\left[W^F(x^\perp)\right]_{i'i}\,. \tag{96}$$

Consequently, going back to Eq. (94),

$$\mathsf{F}\left|\{q; p_k^\perp, p_k^+, \sigma_k, i_k\}\right\rangle = \int \prod_{k=1}^n \frac{d^2 p_k'^\perp}{(2\pi)^2} \int \prod_{k=1}^n d^2 x_k^\perp \exp\left[-ix_k^\perp \cdot (p_k'^\perp - p_k^\perp)\right]$$

$$\times \bigotimes_{k=1}^n \left(\sum_{i_k'=1}^N \left|q; p_k'^\perp, p_k^+, \sigma_k, i_k'\right\rangle\left[W^F(x_k^\perp)\right]_{i_k' i_k}\right)\,. \tag{97}$$

Localizing the states in transverse position, we find a generalization of Eq. (90) for the action of F on a multi-quark state:

$$\mathsf{F}\left|\{q; x_k^\perp, p_k^+, \sigma_k, i_k\}\right\rangle = \sum_{\{i_k'\}} \left|\{q; x_k^\perp, p_k^+, \sigma_k, i_k'\}\right\rangle \prod_k \left[W^F(x_k^\perp)\right]_{i_k' i_k}\,, \tag{98}$$

where the following notation has been introduced,

$$\left|\{q; p_k^\perp, p_k^+, \sigma_k, i_k\}\right\rangle \equiv \bigotimes_{k=1}^n \left|q; p_k^\perp, p_k^+, \sigma_k, i_k\right\rangle$$

$$= b_{i_1}^{(\sigma_1)\dagger}(p_1^\perp, p_1^+)\cdots b_{i_k}^{(\sigma_k)\dagger}(p_k^\perp, p_k^+)\cdots b_{i_n}^{(\sigma_n)\dagger}(p_n^\perp, p_n^+)|0\rangle\,, \tag{99}$$

to denote the multi-quark Fock states.

This formula is readily extended to states of any numbers of particles of different type: it is enough to substitute the discrete quantum numbers, and the representation of the Wilson line. Let us introduce notations. For a type $f = \bar{q}, q, g$ of particle, we denote generically by $h$ its helicity (it may represent $\sigma = \pm\frac{1}{2}$ or $\lambda = \pm 1$ in the case of a quark or antiquark, or gluon respectively), and by $c$ its color (which may represent $i \in \{1, \cdots, N\}$ or $a \in \{1, \cdots, N^2 - 1\}$. A multi-particle state is denoted by

$$\left|\{f_k; p_k^\perp, p_k^+, h_k, i_k\}\right\rangle \equiv \bigotimes_k \left[\left(\mathbb{c}_{f_k}\right)_{c_k}^{(h_k)\dagger}(p_k^\perp, p_k^+)|0\rangle\right], \tag{100}$$

where the generic operator $\mathbb{c}_f$ stands for $d, b, a$ when $f = \bar{q}, q, g$ respectively. The state denoted by $\left\langle\{f_k; p_k^\perp, p_k^+, h_k, c_k\}\right|$ is the adjoint of $\left|\{f_k; p_k^\perp, p_k^+, h_k, c_k\}\right\rangle$.[9] The action of $\mathsf{F}$ on the states (100) is then an obvious generalization of Eq. (98):

$$\mathsf{F}\left|\{q; x_k^\perp, p_k^+, h_k, c_k\}\right\rangle = \sum_{\{c_k'\}}\left|\{f_k; x_k^\perp, p_k^+, h_k, c_k'\}\right\rangle \prod_k \left[W^{R_k}(x_k^\perp)\right]_{c_k'c_k}. \tag{101}$$

**S-matrix elements.** We specialize Eq. (42), replacing $|\mathbb{m}_0\rangle$ and $|\mathbb{m}_0\rangle$ by multi-parton states $\left|\{f_k; x_k^\perp, p_k^+, h_k, c_k\}\right\rangle$ and $\left|\{f_k'; x_k'^\perp, p_k'^+, h_k', c_k'\}\right\rangle$ respectively.

We first need to extend the completeness and orthogonality relations (87),(88) to these multi-particle states. Let us start with two-quark states in momentum space. We write

$$\begin{aligned}
|\vec{p}_1, \sigma_1, i_1; \vec{p}_2, \sigma_2, i_2\rangle &= b_{i_1}^{(\sigma_1)\dagger}(\vec{p}_1)\, b_{i_2}^{(\sigma_2)\dagger}(\vec{p}_2)|0\rangle, \\
\left|\vec{p}_1', \sigma_1', i_1'; \vec{p}_2', \sigma_2', i_2'\right\rangle &= b_{i_1'}^{(\sigma_1')\dagger}(\vec{p}_1')\, b_{i_2'}^{(\sigma_2')\dagger}(\vec{p}_2')|0\rangle.
\end{aligned} \tag{102}$$

The inner product of these two states can be expressed in terms of the inner product of one-particle states:

$$\left\langle\vec{p}_1, \sigma_1, i_1; \vec{p}_2, \sigma_2, i_2\middle|\vec{p}_1', \sigma_1', i_1'; \vec{p}_2', \sigma_2', i_2'\right\rangle = \sum_{s\in\mathfrak{S}_2}(-1)^{\sigma(s)}\prod_{k=1}^{2}\left\langle\vec{p}_{s(k)}, \sigma_{s(k)}, i_{s(k)}\middle|\vec{p}_k', \sigma_k', i_k'\right\rangle, \tag{103}$$

where $\mathfrak{S}_2$ is the group of permutations of the indices $(1,2)$, and $\sigma(s)$ is the parity of the permutation $s$. (The sign is technically a consequence of the fact that the creation/annihilation operators of fermions obey anticommutation relations.) It is straightforward to transpose this identity to mixed space by means of an appropriate Fourier transformation. We can immediately write

$$\begin{aligned}
&\left\langle x_1^\perp, p_1^+, \sigma_1, i_1; x_2^\perp, p_2^+, \sigma_2, i_2\middle|x_1'^\perp, p_1'^+, \sigma_1', i_1'; x_2'^\perp, p_2'^+, \sigma_2', i_2'\right\rangle \\
&= \sum_{s\in\mathfrak{S}_2}(-1)^{\sigma(s)}\prod_{k=1}^{2}\left[\delta^2(x_{s(k)}^\perp - x_k'^\perp)(2\pi)2p_{s(k)}^+\delta(p_{s(k)}^+ - p_k'^+)\delta_{h_{s(k)}h_k'}\delta_{c_{s(k)}c_k'}\right].
\end{aligned} \tag{104}$$

The completeness relation now reads

$$\begin{aligned}
&\sum_{\sigma_1,i_1;\sigma_2,i_2}\int d^2x_1^\perp d^2x_2^\perp \int_0^{+\infty}\frac{dp_1^+}{2p_1^+(2\pi)}\frac{dp_2^+}{2p_2^+(2\pi)} \\
&\times \frac{1}{2}\left|x_1^\perp, p_1^+, \sigma_1, i_1; x_2^\perp, p_2^+, \sigma_2, i_2\right\rangle\left\langle x_1^\perp, p_1^+, \sigma_1, i_1; x_2^\perp, p_2^+, \sigma_2, i_2\right| = \mathbb{I}, \tag{105}
\end{aligned}$$

---

[9]Beware that the order in which the creation operators are applied to the vacuum is important in the case in which there are fermionic operators!

where we made use of the antisymmetry of the states under the exchange of two fermions.

Let us denote by $n_{\bar{q}}, n_q, n_g$ the number of antiquarks, quarks and gluons respectively in the state $\left|\{f_k; x_k^\perp, p_k^+, h_k, c_k\}\right\rangle$. For the purpose of writing the orthogonality relation, it is convenient to split the particle content of the state into three groups corresponding to the three types of particles:

$$\left|\{f_k; x_k^\perp, p_k^+, h_k, c_k\}\right\rangle \rightarrow \left|\{\bar{q}; x_k^\perp, p_k^+, \sigma_k, i_k\}, \{q; x_k^\perp, p_k^+, \sigma_k, i_k\}, \{g; x_k^\perp, p_k^+, \lambda_k, a_k\}\right\rangle. \quad (106)$$

Two states with different particle content are of course orthogonal. Considering now two states of the same set of particles, the orthogonality relation (104) generalizes as

$$\left\langle\{f_k; x_k^\perp, p_k^+, h_k, c_k\}\middle|\{f_k'; x_k'^\perp, p_k'^+, h_k', c_k'\}\right\rangle \quad (107)$$

$$= \sum_{s\in\mathfrak{S}(n_{\bar{q}}, n_q, n_g)} (-1)^{\sigma(s|_{\bar{q},q})} \prod_{k=1}^{n_{\bar{q}}+n_q+n_g} \left[\delta^2(x_{s(k)}^\perp - x_k'^\perp)(2\pi)2p_{s(k)}^+\delta(p_{s(k)}^+ - p_k'^+)\delta_{h_{s(k)}h_k'}\delta_{c_{s(k)}c_k'}\right],$$

where $\mathfrak{S}(n_{\bar{q}}, n_q, n_g)$ denotes the set of independent permutations of the $n_{\bar{q}}$ antiquark, $n_q$ quark, $n_g$ gluon indices among themselves, without mixing between the three groups of indices, and $s|_{\bar{q},q}$ is an element of this set restricted to the fermions. The number of terms in the previous equation is clearly $n_{\bar{q}}!n_q!n_g!$.

As a consequence of the orthogonality formula (107), the correct normalization of the completeness relation is

$$\sum_{\{h_k, c_k\}} \int \prod_{k=1}^{n_{\bar{q}}+n_q+n_g} d^2x_k^\perp \int_0^{+\infty} \prod_{k=1}^{n_{\bar{q}}+n_q+n_g} \frac{dp_k^+}{2p_k^+(2\pi)}$$

$$\times \frac{1}{n_{\bar{q}}!n_q!n_g!} \left|\{f_k; x_k^\perp, p_k^+, h_k, c_k\}\right\rangle\left\langle\{f_k; x_k^\perp, p_k^+, h_k, c_k\}\right| = \mathbb{I}. \quad (108)$$

We are now ready to write (42) for partonic states. Replacing $|\mathtt{m}\rangle, |\mathtt{n}\rangle$ by multi-parton states and reading the sum over the latter off the completeness relation (108), we get

$$S_{\mathtt{fi}} = \sum_{\substack{n_{\bar{q}}, n_q, n_g \\ n_{\bar{q}}', n_q', n_g'}} \int \prod_{k=1}^{n_{\bar{q}}+n_q+n_g} d^2x_k^\perp \prod_{k=1}^{n_{\bar{q}}'+n_q'+n_g'} d^2x_k'^\perp \int_0^{+\infty} \prod_{k=1}^{n_{\bar{q}}+n_q+n_g} \frac{dp_k^+}{2p_k^+(2\pi)} \prod_{k=1}^{n_{\bar{q}}'+n_q'+n_g'} \frac{dp_k'^+}{2p_k'^+(2\pi)}$$

$$\times \sum_{\{h_k', h_k; c_k', c_k\}} \widetilde{\psi}_{\mathtt{f}}^*(\{f_k'; x_k'^\perp, p_k'^+, h_k', c_k'\}) \frac{1}{n_{\bar{q}}'!n_q'!n_g'!} \left\langle\{f_k'; x_k'^\perp, p_k'^+, h_k', c_k'\}\middle|\mathsf{F}\middle|\{f_k; x_k^\perp, p_k^+, h_k, c_k\}\right\rangle$$

$$\times \frac{1}{n_{\bar{q}}!n_q!n_g!} \widetilde{\psi}_{\mathtt{i}}(\{f_k; x_k^\perp, p_k^+, h_k, c_k\}). \quad (109)$$

The first summation is over all possible particles in the interaction state of the projectile. The wave functions $\widetilde{\psi}$ are expressed in the mixed representation: they are related to the wave functions $\psi$ in the momentum representation through

$$\widetilde{\psi}(\{f_k; x_k^\perp, p_k^+, h_k, c_k\}) \equiv \int \left(\prod_{k=1}^n \frac{d^2p_k^\perp}{(2\pi)^2} e^{ix_k^\perp\cdot p_k^\perp}\right) \psi(\{f_k; p_k^\perp, p_k^+, h_k, c_k\}). \quad (110)$$

Replacing by Eq. (101) and using the orthogonality relation (107), the $S$-matrix element $S_{\mathtt{fi}}$ reads

$$S_{\mathtt{fi}} = \sum_{n_{\bar{q}}, n_q, n_g} \frac{1}{n_{\bar{q}}!n_q!n_g!} \int \prod_{k=1}^{n_{\bar{q}}+n_q+n_g} d^2x_k^\perp \int_0^{+\infty} \prod_{k=1}^{n_{\bar{q}}+n_q+n_g} \frac{dp_k^+}{2p_k^+(2\pi)}$$

$$\times \sum_{\{h_k; c_k', c_k\}} \widetilde{\psi}_{\mathtt{f}}^*(\{f_k; x_k^\perp, p_k^+, h_k, c_k'\}) \left(\prod_{k=1}^n \left[W^{R_k}(x_k^\perp)\right]_{c_k'c_k}\right) \widetilde{\psi}_{\mathtt{i}}(\{f_k; x_k^\perp, p_k^+, h_k, c_k\}). \quad (111)$$

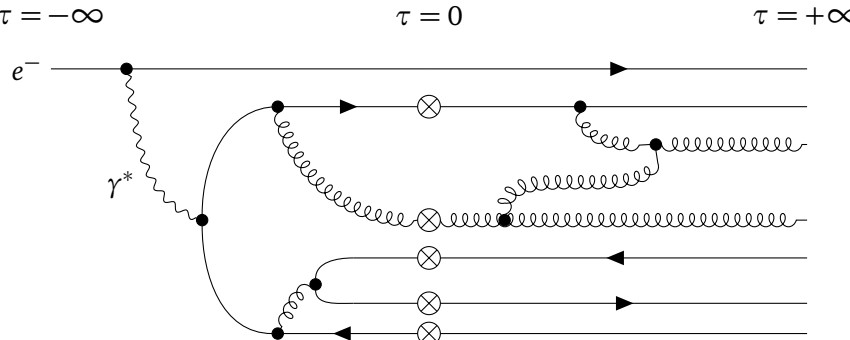

Figure 1: Illustration of Eqs. (42),(111) for an initial electron that fluctuates into two quarks, two antiquarks and one gluon, before it interacts with an external field at (light cone) time $\tau = x^+ = 0$. The crossed dots represent the couplings of the partonic color currents to the external field: technically, they stand for insertions of Wilson lines (92), in the appropriate representation of the color group. After the interaction, evolution to the final state takes place.

This remarkable formula, which is exact in the limit of infinite rapidity $Y$ of the incoming particle, is one of the main equations of these lectures. It expresses the $S$-matrix element for the transition from an initial state $|\mathbb{i}\rangle$ to a final state $|\mathbb{f}\rangle$, under the action of an external potential, as the sum over all possible partonic states of convolutions of the probability amplitudes to find the initial and final systems in that same partonic state, except for the colors of the partons. The latter pick a non-Abelian "eikonal phase", i.e. they undergo "rotations" in color, encoded in products of Wilson lines. These Wilson lines may be thought of as evolution operators in color space along the trajectories of the partons.

The finite-rapidity corrections consist of terms suppressed exponentially with the rapidity, namely of order $e^{-Y}$. Such terms turn out to be negligible. Indeed, one usually considers that the high-energy regime is reached for rapidities above say $\sim 5$, in which case finite-rapidity corrections are less than 1% in order of magnitude. By contrast, we will see in Secs. 3 and 4 that in the perturbative expansion of the wave functions, each power of the coupling constant comes with a power of the rapidity. This implies that the quantum corrections are potentially much larger than the terms we neglect assuming the factorization (111). Hence the latter remains very good, even for large but not strictly infinite rapidities.

The formalism introduced in this section is illustrated in Fig. 1. The initial asymptotic state $|\mathbb{i}_\bullet\rangle$ is that of an electron, and the final state $|\mathbb{f}_\bullet\rangle$ is that of a set composed of an electron, two quarks, two antiquarks and two gluons.

### 2.3.3 Interaction Hamiltonian

In order to be able to compute the partonic wave functions, we now need the matrix elements of the QCD interaction Hamiltonian $H_1$ [see Eq. (17)]. We get the full Hamiltonian $H = H_0 + H_1$ from the Lagrangian density $\mathcal{L}$ through the Legendre transformation (67).

Working out a useful expression for the Hamiltonian is rather cumbersome. The latter is best presented in terms of free fields, the dependent component of which solve the constraint equations with the coupling $g$ set to 0. We introduce

$$\begin{cases} \tilde{A}^\mu \equiv (0, \tilde{A}_+, A^\perp), & \text{with} \quad \tilde{A}_+ \equiv \frac{1}{\partial^+} \partial^\perp \cdot A^\perp, \\ \tilde{\Psi} \equiv \Psi_+ + \tilde{\Psi}_-, & \text{with} \quad \tilde{\Psi}_- \equiv \frac{1}{2i\partial^+} \gamma^+ \left( -i\gamma^j \partial_j + m \right) \Psi_+. \end{cases} \tag{112}$$

The mode expansions of these free fields are then obtained by inserting Eqs. (62),(59) into

Eq. (112). They read

$$\tilde{A}_c^\mu(x) = \int \frac{d^2k^\perp}{(2\pi)^3} \int_0^{+\infty} \frac{dk^+}{2k^+} \sum_{\lambda=\pm 1} \left[ \varepsilon_{(\lambda)}^\mu(k) a_c^{(\lambda)}(\vec{k}) e^{-ikx} + \varepsilon_{(\lambda)}^{*\mu}(k) a_c^{(\lambda)\dagger}(\vec{k}) e^{ikx} \right],$$

$$\tilde{\Psi}_i(x) = \int \frac{d^2p^\perp}{(2\pi)^3} \int_0^{+\infty} \frac{dp^+}{2p^+} \sum_{\sigma=\pm 1/2} \left[ u_{(\sigma)}(p) b_i^{(\sigma)}(\vec{p}) e^{-ipx} + v_{(\sigma)}(p) d_i^{(\sigma)\dagger}(\vec{p}) e^{ipx} \right],$$

(113)

where we have introduced the polarization four-vectors $\varepsilon_{(\lambda)}$ and the spinors $u$, $v$ associated to asymptotic single-particle states of given momentum, helicity and color:

$$\varepsilon_{(\lambda)}^\mu(k) = \left( 0, \frac{k^\perp \cdot \varepsilon_{(\lambda)}^\perp}{k^+}, \varepsilon_{(\lambda)}^\perp \right),$$

$$u_{(\sigma)}(p) = 2^{1/4} \sqrt{p^+} \left[ 1 + \frac{\gamma^0}{\sqrt{2} p^+} \left( \gamma^\perp \cdot p^\perp + m \right) \right] w_{(\sigma)},$$

(114)

$$v_{(\sigma)}(p) = 2^{1/4} \sqrt{p^+} \left[ 1 + \frac{\gamma^0}{\sqrt{2} p^+} \left( \gamma^\perp \cdot p^\perp - m \right) \right] w_{(-\sigma)}.$$

Note that the integration over $x^-$, represented by the operator $1/\partial^+ = 1/\partial_-$ in Eq. (112), actually required a regularization. This regularization may have involved substituting $k^+ \to k^+ \pm i0^+$, $p^+ \to p^+ \pm i0^+$ in the phase factors, with the choice of sign being consistent with the boundary conditions specified in the definition of $1/\partial^+$ [Eq. (56)]. We have not tracked the resulting small imaginary components that shift the poles at vanishing "+" components of momenta in Eq. (114), as these do not affect the outcome of the calculations we will perform.

We check that the expression of the gluon polarization four-vectors is consistent with the gauge conditions $\eta \cdot \varepsilon_{(\lambda)} = \varepsilon_{(\lambda)}^+ = 0$ and $k \cdot \varepsilon_{(\lambda)} = 0$. These equations, along with the previously mentioned boundary conditions on the fields, would determine all components unambiguously when the transverse components have been chosen.

The two polarization vectors obey the completeness and orthogonality relations

$$\sum_{\lambda=\pm 1} \varepsilon_{(\lambda)\mu}(k) \varepsilon_{(\lambda)\nu}^*(k) = -g_{\mu\nu} + \frac{k_\mu \eta_\nu + k_\nu \eta_\mu}{k \cdot \eta}, \qquad \varepsilon_{(\lambda)}^{*\mu}(k) \varepsilon_{(\lambda')\mu}(k) = -\delta_{\lambda\lambda'}$$

(115)

(The regularization of the pole at $k^+ = 0$ is again left implicit). The spinors $u_{(\sigma)}(p)$ and $v_{(\sigma)}(p)$ obey the Dirac equation for fermions and antifermions respectively, and one can check that their helicities viewed from a frame moving along the negative $z$-axis at almost the speed of light, the so-called "infinite-momentum frame", are $\sigma$ and $-\sigma$ respectively.[10] As a consequence of the properties of the $w$'s in Eq. (60), they obey the following completeness and orthogonality relations:

$$\sum_{\sigma\pm 1/2} u_{(\sigma)}(p) \bar{u}_{(\sigma)}(p) = \slashed{p} + m, \qquad \bar{u}_{(\sigma)}(p) u_{(\sigma')}(p) = 2m\delta_{\sigma\sigma'},$$

$$\sum_{\sigma=\pm 1/2} v_{(\sigma)}(p) \bar{v}_{(\sigma)}(p) = \slashed{p} - m, \qquad \bar{v}_{(\sigma)}(p) v_{(\sigma')}(p) = -2m\delta_{\sigma\sigma'}.$$

(116)

We are now in a position to write a practical expression for $H$. The free Hamiltonian $H_0$ eventually reads

$$H_0 = \int d^3\vec{x} \left( -\tilde{A}_\mu^a \frac{\left( i\partial^\perp \right)^2}{2} \tilde{A}_a^\mu + \bar{\tilde{\Psi}} \gamma^+ \frac{\left( i\partial^\perp \right)^2 + m^2}{2i\partial^+} \tilde{\Psi} \right).$$

(117)

---

[10]To check this statement, one just needs to work out the expression of the corresponding spin matrix, see e.g. Appendix B of Ref. [37], and verify that the four-spinors $u_{(\sigma)}$, $v_{(\sigma)}$ are eigenstates with respective eigenvalues $+\sigma$ and $-\sigma$.

$\vec{p}\,',\sigma',i' \longrightarrow \vec{p},\sigma,i$
$\quad\vec{k},\lambda,a$

$\vec{k}\,',\lambda',a' \quad\quad \vec{k}_1,\lambda_1,a_1$
$\quad\quad\quad\quad \vec{k}_2,\lambda_2,a_2$

$\vec{k}\,',\lambda',a' \quad\quad \vec{p}_1,\sigma_1,i_1$
$\quad\quad\quad\quad \vec{p}_0,\sigma_0,i_0$

(a) Gluon emission off a quark.

(b) Branching of a gluon into two gluons.

(c) Branching of a gauge boson into a quark-antiquark pair.

Figure 2: Feynman diagrams for particular matrix elements of the interaction Hamiltonian $H_1$. All momenta flow from left to right. The corresponding matrix elements are given in Eqs. (120), (121) and (122) respectively.

We have written it with the help of the full fields $\tilde{A}$ and $\tilde{\Psi}$, but only the transverse components of the former and the "+" projection of the latter actually contribute [see the expression obtained earlier, Eq. (68)]. Replacing $A^{\perp}$ and $\Psi_{+}$ by their mode expansions (59) and (62), one checks that the states of sets of free particles are indeed eigenstates of $H_0$, understood as a normal-ordered operator on the Hilbert space, with the correct eigenvalue, namely the sum of their light cone energies (32).

We split the remaining terms in $H$, that is the interaction part $H_1$ of the Hamiltonian, into two parts: $H_1 = H_1^{\text{causal}} + H_1^{\text{instant}}$. The "causal" interaction terms have a similar structure as that of the interaction terms of the initial Lagrangian:

$$H_1^{\text{causal}} = \int d^3\vec{x}\left[gf^{abc}\partial_\mu\tilde{A}_\nu^a\tilde{A}_b^\mu\tilde{A}_c^\nu \quad +\frac{g^2}{4}\left(f^{abc}\tilde{A}_\mu^b\tilde{A}_\nu^c\right)^2 - g\bar{\tilde{\Psi}}t^a\gamma^\mu\tilde{A}_\mu^a\tilde{\Psi}\right]. \tag{118}$$

They define the familiar 3-gluon, 4-gluon and quark-gluon vertices, respectively. The remaining terms can be cast in the following form:

$$H_1^{\text{instant}} = \int d^3\vec{x}\left[\frac{g^2}{2}f^{abc}\partial^+\tilde{A}_b\cdot\tilde{A}_c\frac{1}{(i\partial^+)^2}f^{ab'c'}\partial^+\tilde{A}_{b'}\cdot\tilde{A}_{c'}\right.$$

$$+g^2\bar{\tilde{\Psi}}t^a\gamma^+\tilde{\Psi}\frac{1}{(i\partial^+)^2}f^{abc}\partial^+\tilde{A}_b\cdot\tilde{A}_c$$

$$+\frac{g^2}{2}\bar{\tilde{\Psi}}t^a\gamma^\mu\tilde{A}_\mu^a\gamma^+\frac{1}{i\partial^+}\gamma^\nu\tilde{A}_\nu^b t^b\tilde{\Psi}$$

$$\left.+\frac{g^2}{2}\bar{\tilde{\Psi}}t^a\gamma^+\tilde{\Psi}\frac{1}{(i\partial^+)^2}\bar{\tilde{\Psi}}t^a\gamma^+\tilde{\Psi}\right]. \tag{119}$$

These terms represent effective four-parton vertices (the crossed-out particle is actually exchanged instantaneously), similar to the Coulomb term in quantum electrodynamics.

The main matrix elements of the interaction Hamiltonian we will need below, in order to derive the color dipole model, are $\langle qg|{:}H_1{:}|q\rangle$, $\langle \bar{q}g|{:}H_1{:}|\bar{q}\rangle$ and $\langle gg|{:}H_1{:}|g\rangle$. The first two read (see Fig. 2a)

$$\langle qg|{:}H_1{:}|q\rangle \equiv \left(\langle q;\vec{p},\sigma,i|\otimes\langle g;\vec{k},\lambda,a|\right)\int d^3\vec{x}\left(-g{:}\bar{\tilde{\Psi}}t^a\gamma^\mu\tilde{A}_\mu^a\tilde{\Psi}{:}\right)|q;\vec{p}\,',\sigma',i'\rangle$$

$$= -g(t^a)_{ii'}\left[\bar{u}_{(\sigma)}(p)\not{\epsilon}^*_{(\lambda)}(k)u_{(\sigma')}(p')\right](2\pi)^3\delta^3(\vec{k}+\vec{p}-\vec{p}\,'),$$

$$\langle \bar{q}g|{:}H_1{:}|\bar{q}\rangle \equiv \left(\langle \bar{q};\vec{p},\sigma,i|\otimes\langle g;\vec{k},\lambda,a|\right)\int d^3\vec{x}\left(-g{:}\bar{\tilde{\Psi}}t^a\gamma^\mu\tilde{A}_\mu^a\tilde{\Psi}{:}\right)|\bar{q};\vec{p}\,',\sigma',i'\rangle$$

$$= -g\left[-(t^a)_{i'i}\right]\left[\bar{v}_{(\sigma')}(p')\not{\epsilon}^*_{(\lambda)}(k)v_{(\sigma)}(p)\right](2\pi)^3\delta^3(\vec{k}+\vec{p}-\vec{p}\,'). \tag{120}$$

In going from the first to the second lines in each of these pairs of equations, we have just replaced the fields in the interaction operator by their mode expansions (113), normal ordered, and contracted left and right by the state vectors. Note that these matrix elements include a Dirac distribution enforcing three-momentum conservation.

A comment is in order on the difference between the emission off the quark and that off the antiquark. The Dirac matrix elements are actually identical in these expressions, thanks to an identity found in Tab. 1 (Appendix A.2). Hence $\langle qg|{:}H_1{:}|q\rangle$ and $\langle \bar{q}g|{:}H_1{:}|\bar{q}\rangle$ effectively differ only by the color generator: in going from the first one to the second one, $T_F$ is replaced by $T_{\bar{F}}$, see Eq. (85).

The transition matrix element $\langle gg|{:}H_1{:}|g\rangle$ from a gluon to two gluons reads instead (see Fig. 2b)

$$
\begin{aligned}
\langle gg|{:}H_1{:}|g\rangle &\equiv \left(\langle g;\vec{k}_1,\lambda_1,a_1|\otimes\langle g;\vec{k}_2,\lambda_2,a_2|\right)\int d^3\vec{x}\left(gf^{abc}{:}\partial_\mu\tilde{A}^a_\nu\tilde{A}^\mu_b\tilde{A}^\nu_c{:}\right)\left|g;\vec{k}',\lambda',a'\right\rangle \\
&= -g\left(-if^{a_2a_1a'}\right)\left\{\left[(k_1-k_2)\cdot\varepsilon_{(\lambda')}(k')\right]\left[\varepsilon^*_{(\lambda_1)}(k_1)\cdot\varepsilon^*_{(\lambda_2)}(k_2)\right]\right. \\
&\quad +\left[(k_2+k')\cdot\varepsilon^*_{(\lambda_1)}(k_1)\right]\left[\varepsilon^*_{(\lambda_2)}(k_2)\cdot\varepsilon_{(\lambda')}(k')\right] \\
&\quad \left.+\left[(-k'-k_1)\cdot\varepsilon^*_{(\lambda_2)}(k_2)\right]\left[\varepsilon_{(\lambda')}(k')\cdot\varepsilon^*_{(\lambda_1)}(k_1)\right]\right\}(2\pi)^3\delta^3(\vec{k}_1+\vec{k}_2-\vec{k}').
\end{aligned}
\tag{121}
$$

As for the applications of the dipole model to phenomenology (Sec. 6), we will also need the transition amplitude from a gauge boson to a quark-antiquark pair. It reads (see Fig. 2c)

$$
\begin{aligned}
\langle \bar{q}q|{:}H_1{:}|g\rangle &\equiv \left(\langle \bar{q};\vec{p}_0,\sigma_0,i_0|\otimes\langle q;\vec{p}_1,\sigma_1,i_1|\right)\int d^3\vec{x}\left(-g{:}\bar{\tilde{\Psi}}t^a\gamma^\mu\tilde{A}^a_\mu\tilde{\Psi}{:}\right)\left|g;\vec{k}',\lambda',a'\right\rangle \\
&= -g(t^{a'})_{i_1i_0}\left[\bar{u}_{(\sigma_1)}(p_1)\slashed{\varepsilon}_{(\lambda')}(k')v_{(\sigma_0)}(p_0)\right](2\pi)^3\delta^3(\vec{p}_0+\vec{p}_1-\vec{k}').
\end{aligned}
\tag{122}
$$

If the initial particle were a photon instead of a gluon, then the only changes would be to replace the color matrix by the identity, namely in components $\delta_{i_1i_0}$, and the strong coupling constant by the electric charge of the produced fermions.

All other matrix elements of $H_1$ are obtained in a similar way.

We are now ready to compute cross sections for the scattering of particles off an external field. We will mostly focus on onia.

## 3 Onium scattering up to the first quantum correction

The main goal of this section is to compute the $S$-matrix elements that encode the eikonal scattering of an onium off an external field, specializing to elastic scattering and forward kinematics. An onium is a toy hadron whose asymptotic state consists of a superposition of globally color-neutral quark-antiquark pairs. By definition of forward elastic scattering, the initial and final asymptotic states, $|i_\bullet\rangle$ and $|f_\bullet\rangle$, are strictly identical. In this section, we will limit ourselves to the leading quantum correction.

In order to arrive at an expression for such matrix element of $S$, we need to understand the physical state of quark-antiquark pairs, namely, to compute their light cone wave functions in perturbation theory. We shall first compute the wave functions of the dressed states of a single quark up to the lowest-lying non-trivial fluctuation (Sec. 3.1). We subsequently add an antiquark to form the onium. Once the wave functions of this object are understood, we insert them into Eq. (111) to get an expression for $S$ (Sec. 3.2). The latter turns out to be singular: how to remedy is discussed in Sec. 3.3. The final section 3.4 reviews the obtained result and motivates the calculation of higher perturbative orders, on which we will elaborate in Sec. 4.

**Notations.** In order to lighten the formalism, it will prove useful to introduce the following notations for the one-particle phase-space integrals, in the momentum and the mixed representations respectively:

$$\int_{\vec{k}} \equiv \int \frac{d^2 k^\perp}{(2\pi)^2} \int_0^{+\infty} \frac{dk^+}{(2\pi)2k^+} , \qquad \int_{(x^\perp, k^+)} \equiv \int d^2 x^\perp \int_0^{+\infty} \frac{dk^+}{(2\pi)2k^+} . \tag{123}$$

Restrictions on the range of integration over the "+" components will be discussed when the problem arises. We shall also introduce a complete set of generic states $\left|x^\perp, k^+\right\rangle$:

$$\int_{(x^\perp, k^+)} \left|x^\perp, k^+\right\rangle\left\langle x^\perp, k^+\right| = \mathbb{I}, \qquad \left\langle x'^\perp, k'^+ \middle| x^\perp, k^+\right\rangle = \delta^2(x'^\perp - x^\perp) 2k^+ (2\pi)\delta(k'^+ - k^+). \tag{124}$$

This set of vectors would span the Hilbert space of the single-particle states of a scalar field.

## 3.1 Fock state of a quark at lowest order and in the soft-gluon limit

Let us consider a bare quark state $|\mathbb{i}_\bullet\rangle = |q\rangle$ at $x^+ = -\infty$, with definite momentum $\vec{k}_{\bullet 1}$, helicity $\sigma_{\bullet 1}$ and color $i_{\bullet 1}$. Its interaction state $|\mathbb{i}(q)\rangle$ at light cone time $x^+ = 0$ consists of a superposition of a single quark, of a quark-gluon pair, and of higher-multiplicity states. We shall focus on the lowest non-trivial quantum fluctuation, namely the $|qg\rangle$ state.

### 3.1.1 Probability amplitude of a quark-gluon fluctuation

**Wave function in momentum representation.** The corresponding wave function $\langle qg|\mathbb{i}(q)\rangle$ reads

$$\psi_{qg,\mathbb{i}(q)}^{\sigma_1,i_1;\lambda_2,a_2}(\vec{k}_1; \vec{k}_2) = \langle qg|H_1|q\rangle + \mathcal{O}(g^2) \tag{125}$$

$$= \frac{\left(\left\langle q; k_1^\perp, k_1^+, \sigma_1, i_1\right| \otimes \left\langle g; k_2^\perp, k_2^+, \lambda_2, a_2\right|\right) H_1 \left|q; k_{\bullet 1}^\perp, k_{\bullet 1}^+, \sigma_{\bullet 1}, i_{\bullet 1}\right\rangle}{k_{\bullet 1}^- - k_1^- - k_2^-} + \mathcal{O}(g^2),$$

where the explicit term displayed is just the lowest-order term in the perturbative expansion (17). We will denote it by $\psi_{qg,\mathbb{i}(q)}^{(1)\sigma_1,i_1;\lambda_2,a_2}(\vec{k}_1; \vec{k}_2)$ in the following. The matrix element of the interaction Hamiltonian which appears in the numerator can be replaced by the one given in Eq. (120).[11] Thanks to the mass-shell relation (32), the denominator can be expressed in terms of the phase-space variables, namely the three-dimensional momenta:

$$k_{\bullet 1}^- - k_1^- - k_2^- = \frac{k_{\bullet 1}^{\perp 2} + m^2}{2k_{\bullet 1}^+} - \frac{k_1^{\perp 2} + m^2}{2k_1^+} - \frac{k_2^{\perp 2}}{2k_2^+} . \tag{126}$$

We shall now specialize to the Regge kinematics: the initial quark is very fast relative to the source of the external field, and the "+" component of the momenta $k_1^+, k_{\bullet 1}^+, k_2^+$ are all much larger than the transverse components $|k_1^\perp|, |k_{\bullet 1}^\perp|, |k_2^\perp|$. In turn, the latter are all assumed to be on the same order of magnitude. We shall call $Q$ the typical scale of these transverse momenta, in reference to the usual notation for the photon virtuality, which sets the magnitude of the transverse momenta in deep-inelastic scattering processes. The quark mass $m$ can safely be set to zero if one is considering light quarks; otherwise, one may assume that $m$ is at most on the order of $Q$. In both cases, it never shows up in the results below.

---

[11]The disconnected graph would a priori contribute to the matrix element $\langle qg|H_1|q\rangle$, but it vanishes due to color charge conservation (as well as for other reasons!). Only the normal-ordered part of the operator $H_1$ eventually needs to be taken into account in the calculation of this wave function.

**Eikonal approximation.** We know from general quantum field theory (see e.g. [33–35]) that the emission of soft massless bosons is enhanced due to the infrared singularity of gauge theories. At high energy, there is a large phase space for gluons carrying a very small momentum fraction of their emitter: the latter will dominate any observable integrated over the gluon momentum, such as total cross sections. Therefore, it is in order to simplify the expression of the wave function (125) by assuming that the emitted gluon is soft with respect to the quark, i.e. $k_2^+ \ll k_1^+ \simeq k_{\bullet 1}^+$, while the transverse momenta are smaller and all on the same order, consistently with the Regge kinematics.

The energy denominator in Eq. (126) reduces to one single term

$$k_{\bullet 1}^- - k_1^- - k_2^- \simeq -\frac{k_2^{\perp 2}}{2k_2^+}, \tag{127}$$

that up to the sign, corresponds to the light cone energy of the softest particle, namely the emitted gluon.

We turn to the numerator in Eq. (125). Substituting it with the explicit expression (120), we see that it contains the factor

$$\bar{u}_{(\sigma_1)}(k_1)\slashed{\varepsilon}^*_{(\lambda_2)}(k_2)u_{(\sigma_{\bullet 1})}(k_{\bullet 1}) = \bar{u}_{(\sigma_1)}(k_1)\gamma^+ u_{(\sigma_{\bullet 1})}(k_{\bullet 1})\frac{k_2^\perp \cdot \varepsilon^{\perp *}_{(\lambda_2)}}{k_2^+}$$
$$- \left[\bar{u}_{(\sigma_1)}(k_1)\gamma^\perp u_{(\sigma_{\bullet 1})}(k_{\bullet 1})\right] \cdot \varepsilon^{\perp *}_{(\lambda_2)}, \tag{128}$$

where we have replaced the components of the polarization four-vector $\varepsilon^\mu_{(\lambda_2)}$ by their values in the light cone gauge, see Eq. (114). Using the formulas of the Dirac matrix elements from Tab. 1 (Appendix A.2) and taking the soft-gluon limit, the contribution of the "+" component of the fermionic current reads

$$\bar{u}_{(\sigma_1)}(k_1)\gamma^+ u_{(\sigma_{\bullet 1})}(k_{\bullet 1})\frac{k_2^\perp \cdot \varepsilon^{\perp *}_{(\lambda_2)}}{k_2^+} \simeq 2k_2^\perp \cdot \varepsilon^{\perp *}_{(\lambda_2)}\frac{k_1^+}{k_2^+}\delta_{\sigma_1,\sigma_{\bullet 1}}. \tag{129}$$

The contribution of the transverse component reads (see again Tab. 1)

$$\left[\bar{u}_{(\sigma_1)}(k_1)\gamma^\perp u_{(\sigma_{\bullet 1})}(k_{\bullet 1})\right] \cdot \varepsilon^{\perp *}_{(\lambda_2)} \simeq 2k_1^\perp \cdot \varepsilon^{*\perp}_{(\lambda_2)}\delta_{\sigma_1\sigma_{\bullet 1}} - 2\sigma_1 m\frac{k_2^+}{k_1^+}\left(\varepsilon^{*\perp 1}_{(\lambda_2)} - 2i\sigma_1 \varepsilon^{*\perp 2}_{(\lambda_2)}\right)\delta_{\sigma_1,-\sigma_{\bullet 1}}. \tag{130}$$

Let us examine the size of the terms of Eq. (130) relative to that of the term of Eq. (129). Parametrically, the ratio of the first term of Eq. (130) to Eq. (129) is proportional to $k_2^+/k_1^+$, which is very small, while the ratio of the last term in Eq. (130) to Eq. (129) is proportional to $\left(k_2^+/k_1^+\right)^2\left(m/|k_2^\perp|\right)$, which is even smaller. Therefore, we may ignore the contribution of Eq. (130) in Eq. (128), and use (129) as an approximation for the latter. Hence the matrix element of the interaction Hamiltonian reads, in this limit,

$$\left(\langle q; k_1^\perp, k_1^+, \sigma_1, i_1| \otimes \langle g; k_2^\perp, k_2^+, \lambda_2, a_2|\right)H_1|q; k_{\bullet 1}^\perp, k_{\bullet 1}^+, \sigma_{\bullet 1}, i_{\bullet 1}\rangle$$
$$\simeq -g\delta_{\sigma_1,\sigma_{\bullet 1}}(t^{a_2})_{i_1 i_{\bullet 1}}k_2^\perp \cdot \varepsilon^{\perp *}_{(\lambda_2)}\frac{2k_1^+}{k_2^+}(2\pi)^3\delta^3(\vec{k}_2 + \vec{k}_1 - \vec{k}_{\bullet 1}). \tag{131}$$

We recover the fact that the "+" component of the current of a fast particle that couples to the soft gauge field dominates [see Eq. (37)]. The $k^+$ term in the momentum conservation factor may be dropped: what is left is the approximate conservation of the "+" momentum of the quark, and the exact conservation of the transverse momenta. The existence of the factor $\delta_{\sigma_1\sigma_{\bullet 1}}$ means that the helicity of the fast particle does not change, a fact that we have also identified earlier as characteristic of the eikonal approximation [see e.g. Eq. (84)].

Inserting Eqs. (131) and (127) into Eq. (125), we arrive at the following expression for the quark-gluon wave function at order $g$ in perturbation theory and in the eikonal approximation:

$$\psi_{qg,\mathfrak{i}(q)}^{(1)\sigma_1,i_1;\lambda_2,a_2}(\vec{k}_1;\vec{k}_2) \simeq 2g\,\delta_{\sigma_1\sigma_{\bullet 1}}\,(t^{a_2})_{i_1 i_{\bullet 1}}\frac{k_2^\perp \cdot \varepsilon_{(\lambda_2)}^{\perp *}}{(k_2^\perp)^2}\,2k_1^+(2\pi)^3\delta^2(k_2^\perp + k_1^\perp - k_{\bullet 1}^\perp)\delta(k_1^+ - k_{\bullet 1}^+). \quad (132)$$

**Mixed representation.** Since our ultimate goal will be to compute $S$ through Eq. (111), in which the wave functions appear as depending on the transverse positions and "+" momenta of the partons, we take over this wave function to transverse position space. We just need to perform Fourier transforms with respect to the momenta $k_1^\perp \to x_1^\perp$, $k_2^\perp \to x_2^\perp$ and $k_{\bullet 1}^\perp \to x_{\bullet 1}^\perp$, using the normalization conventions defined e.g. in Eq. (110):

$$\widetilde{\psi}_{qg,\mathfrak{i}(q)}^{\sigma_1,i_1;\lambda_2,a_2}(x_1^\perp,k_1^+;x_2^\perp,k_2^+) = \left(\langle q;x_1^\perp,k_1^+,\sigma_1,i_1| \otimes \langle g;x_2^\perp,k_2^+,\lambda_2,a_2|\right)\big|\mathfrak{i}(q;x_{\bullet 1}^\perp,k_{\bullet 1}^+,\sigma_{\bullet 1},i_{\bullet 1})\rangle$$

$$= \int \frac{d^2k_1^\perp}{(2\pi)^2}\frac{d^2k_{\bullet 1}^\perp}{(2\pi)^2}\frac{d^2k_2^\perp}{(2\pi)^2}e^{i(x_1^\perp\cdot k_1^\perp + x_2^\perp\cdot k_2^\perp - x_{\bullet 1}^\perp\cdot k_{\bullet 1}^\perp)}\psi_{qg,\mathfrak{i}(q)}^{\sigma_1,i_1;\lambda_2,a_1}(\vec{k}_1;\vec{k}_2). \quad (133)$$

Inserting the expression for $\psi_{qg,\mathfrak{i}(q)}^{(1)}$ obtained in Eq. (132), using the formula[12]

$$\int d^2k^\perp e^{ik^\perp\cdot x^\perp}\frac{k^\perp}{(k^\perp)^2} = 2i\pi\frac{x^\perp}{(x^\perp)^2}, \quad (134)$$

to perform the integration over the transverse momentum of the gluon, we get

$$\widetilde{\psi}_{qg,\mathfrak{i}(q)}^{(1)\sigma_1,i_1;\lambda_2,a_2}(x_1^\perp,k_1^+;x_2^\perp,k_2^+) \simeq \langle x_1^\perp,k_1^+|x_{\bullet 1}^\perp,k_{\bullet 1}^+\rangle\,\delta_{\sigma_1\sigma_{\bullet 1}}\,(t^{a_2})_{i_1 i_{\bullet 1}}\frac{ig}{\pi}\frac{x_{21}^\perp \cdot \varepsilon_{(\lambda_2)}^{*\perp}}{\left(x_{21}^\perp\right)^2}, \quad (135)$$

where we have introduced the notation $x_{ab}^\perp = x_a^\perp - x_b^\perp$, and used the set of states defined in Eq. (124) to make the formula look more compact.

Note that if we had an antiquark instead of a quark, then the wavefunction would almost be identical, except for the substitution of the color generator $T_F^{a_2} = t^{a_2} \to T_{\bar{F}}^{a_2} = -(t^{a_2})^T$ [see Eq. (120)], which implies in particular a crucial change of sign:

$$\widetilde{\psi}_{\bar{q}g,\mathfrak{i}(\bar{q})}^{(1)\sigma_1,i_1;\lambda_2,a_2}(x_1^\perp,k_1^+;x_2^\perp,k_2^+) \simeq \langle x_1^\perp,k_1^+|x_{\bullet 1}^\perp,k_{\bullet 1}^+\rangle\,\delta_{\sigma_1\sigma_{\bullet 1}}\left[-(t^{a_2})_{i_{\bullet 1}i_1}\right]\frac{ig}{\pi}\frac{x_{21}^\perp \cdot \varepsilon_{(\lambda_2)}^{*\perp}}{\left(x_{21}^\perp\right)^2}. \quad (136)$$

**Wave packet.** For later convenience and future reference, we now consider as an initial state a normalized superposition of single-quark states of fixed transverse positions and "+" momenta. We shall assign to all of these states the same fixed color index $i_{\bullet 1}$ and polarization $\sigma_{\bullet 1}$:

$$\left|\phi_{\bullet q}\right\rangle = \int_{(x_{\bullet 1}^\perp,k_{\bullet 1}^+)}\phi_{\bullet q}(x_{\bullet 1}^\perp,k_{\bullet 1}^+)\left|q;x_{\bullet 1}^\perp,k_{\bullet 1}^+,\sigma_{\bullet 1},i_{\bullet 1}\right\rangle, \qquad \text{with} \qquad \int_{(x_{\bullet 1}^\perp,k_{\bullet 1}^+)}\left|\phi_{\bullet q}(x_{\bullet 1}^\perp,k_{\bullet 1}^+)\right|^2 = 1. \quad (137)$$

When measuring a system prepared in such a state, the probability amplitude to observe a quark and a gluon at fixed transverse positions, "+" momenta, polarizations and colors is then simply

$$\widetilde{\psi}_{qg,\mathfrak{i}(\phi_q)}^{(1)\sigma_1,i_1;\lambda_2,a_2}(x_1^\perp,k_1^+;x_2^\perp,k_2^+) = \left(\langle q;x_1^\perp,k_1^+,\sigma_1,i_1| \otimes \langle g;x_2^\perp,k_2^+,\lambda_2,a_2|\right)\big|\mathfrak{i}(\phi_{\bullet q})\rangle\big|_{\mathcal{O}(g)}$$

$$= \int_{(x_{\bullet 1}^\perp,k_{\bullet 1}^+)}\phi_{\bullet q}(x_{\bullet 1}^\perp,k_{\bullet 1}^+)\,\widetilde{\psi}_{qg,\mathfrak{i}(q)}^{(1)\sigma_1,i_1;\lambda_2,a_2}(x_1^\perp,k_1^+;x_2^\perp,k_2^+)$$

$$\simeq \phi_{\bullet q}(x_1^\perp,k_1^+)\,\delta_{\sigma_1\sigma_{\bullet 1}}\,(t^{a_2})_{i_1 i_{\bullet 1}}\frac{ig}{\pi}\frac{x_{21}^\perp \cdot \varepsilon_{(\lambda_2)}^{*\perp}}{\left(x_{21}^\perp\right)^2}. \quad (138)$$

---

[12]Equation (134) can be proven e.g. using the Cauchy theorem with appropriate contours.

One may construct the wave packet with antiquark states: the expression of the wave function would be the same, again up to the replacement of the color factor.

Comparing Eq. (138) with Eq. (135), the only difference is that the probability amplitude to find the initial quark at position $x_1^\perp$ and with momentum $p^+$ has been substituted as follows:

$$\left\langle x_1^\perp, k_1^+ \middle| x_{\bullet 1}^\perp, k_{\bullet 1}^+ \right\rangle \to \phi_{\bullet q}(x_1^\perp, k_1^+). \tag{139}$$

Finally, let us comment on the common structure of Eqs. (135), (136) and (138). The first two factors in these equations encode the probability amplitude to find the quark or the antiquark at the specified transverse position $x_1^\perp$ with the "+" momentum $k_1^+$ and the helicity $\sigma_1$. The remaining terms in these equations account for the emission of the gluon. The dependence upon the color appears in the form of an infinitesimal SU($N$) generator $\left(T_R^{a_2}\right)_{i_1 i_{\bullet 1}}$, with $R = F$ or $\bar{F}$; see Eq. (85). Between the emission off the quark and off the antiquark, the generator gets transposed and takes a "−" sign. It will prove convenient to associate this sign to the remaining factor, which exhibits the dependence of the emission probability amplitude on the transverse space variables. We shall denote this factor by $E_{21}$ for the emission off the quark, and by $\bar{E}_{21}$ for the emission off the antiquark, with the following definition:

$$E_{l'l} \equiv \frac{ig}{\pi} \frac{x_{l'l}^\perp \cdot \varepsilon_{(\lambda_{l'})}^{*\perp}}{x_{l'l}^{\perp 2}}, \qquad \bar{E}_{l'l} \equiv -E_{l'l}. \tag{140}$$

With these rules, the gluon emission factor consists of the color factor $t_{i'i}^{a_2}$, where $i'$ (resp. $i$) is the fundamental color index on the leg of the quark-gluon or antiquark-gluon vertex from which the arrow comes out (resp. in), multiplied by the factor $E_{21}$ or $\bar{E}_{21}$.

### 3.1.2 Virtual corrections up to order $g^2$

When we compute the probability that the quark interacts in a given state, the wave function is squared, and thus contributes to order $g^2$. At the same order, we have virtual corrections to consider. Thus for consistency, we need to push the expansion of the one and two-particle Fock states to order $g^2$.

Let us write $|\mathbb{i}(q)\rangle$ as a superposition of partonic states consisting of one or two particles, weighted by the corresponding wave functions, keeping explicit all terms up to $\mathcal{O}(g^2)$:

$$|\mathbb{i}(q)\rangle = \left| q; \vec{k}_{\bullet 1}, \sigma_{\bullet 1}, i_{\bullet 1} \right\rangle + \sum_{\lambda_2, \sigma_1; a_2, i_1} \int_{\vec{k}_2, \vec{k}_1} \left| q; \vec{k}_1, \sigma_1, i_1 \right\rangle \otimes \left| g; \vec{k}_2, \lambda_2, a_2 \right\rangle \psi_{qg, \mathbb{i}(q)}^{(1)\sigma_1, i_1; \lambda_2, a_2}(\vec{k}_1, \vec{k}_2)$$

$$+ \sum_{\sigma_1, i_1} \int_{\vec{k}_1} \left| q; \vec{k}_1, \sigma_1, i_1 \right\rangle \psi_{q, \mathbb{i}(q)}^{(2)\sigma_1, i_1}(\vec{k}_1) + \cdots. \tag{141}$$

$\psi_{qg, \mathbb{i}(q)}^{(1)}$ is the wave function we have computed in Eq. (132). The function $\psi_{q, \mathbb{i}(q)}^{(2)}$ instead is meant to represent the lowest-order non-trivial contribution to the probability amplitude to find the initial quark still in a single-quark state, namely to $\sqrt{Z_q}$. We could extract it from the calculation of the matrix element $\langle \mathbb{i}_\bullet| U_I^\epsilon(0, -\infty) |\mathbb{i}_\bullet\rangle$. Applying naively Eq. (17), the latter would read

$$\frac{1}{k_{\bullet 1}^- - k_1^- + 2i\epsilon} \sum_{\sigma', \lambda_2; i', a_2} \int_{\vec{k}_2, \vec{k}'} \left\langle q; \vec{k}_1, \sigma_1, i_1 \middle| H_1 \left( \left| q; \vec{k}', \sigma', i' \right\rangle \otimes \left| g; \vec{k}_2, \lambda_2, a_2 \right\rangle \right) \right.$$

$$\times \frac{1}{k_{\bullet 1}^- - k'^- - k_2^- + i\epsilon} \left( \left\langle q; \vec{k}', \sigma', i' \middle| \otimes \left\langle g; \vec{k}_2, \lambda_2, a_2 \middle| \right) H_1 \left| q; \vec{k}_{\bullet 1}, \sigma_{\bullet 1}, i_{\bullet 1} \right\rangle \right.$$

$$+ \frac{1}{k_{\bullet 1}^- - k_1^- + i\epsilon} \left\langle q; \vec{k}_1, \sigma_1, i_1 \middle| H_1 \middle| q; \vec{k}_{\bullet 1}, \sigma_{\bullet 1}, i_{\bullet 1} \right\rangle. \tag{142}$$

$$\vec{k}_{\bullet 1}, \sigma_{\bullet 1}, i_{\bullet 1} \longrightarrow \overset{\vec{k}', \sigma', i'}{\longrightarrow} \vec{k}_1, \sigma_1, i_1$$

$$\vec{k}_2, \lambda_2, a_2$$

(a)

$$\vec{k}_{\bullet 1}, \sigma_{\bullet 1}, i_{\bullet 1} \longrightarrow \qquad \vec{k}_1, \sigma_1, i_1$$

$$\vec{k}_{\bullet 1}, \sigma_{\bullet 1}, i_{\bullet 1} \longrightarrow \qquad \vec{k}_1, \sigma_1, i_1$$

(b)

Figure 3: LCPT graphs contributing to the probability amplitude to find a quark in an initial quark; see Eq. (142). (a) causal diagram; (b) instantaneous diagrams.

This expression takes contributions from a causal loop diagram [first term in Eq. (142); see Fig. 3a] and diagrams involving the instantaneous part of the Hamiltonian [second term in Eq. (142); see Fig. 3b].

Generally speaking, the calculation of virtual corrections is tedious in LCPT. Fortunately, we will actually never need to compute these corrections through Eq. (142): we may get them from Eq. (19), namely as a consequence of probability conservation.

**Virtual corrections to the one-particle state from unitarity.** We specialize the formula (19) to the state of the quark, calling the corresponding wave packet $\phi_{q\bullet}$ and the renormalization constant $Z_q$. We can work in the mixed representation for the explicit calculation: the states $|\mathbb{m}_0\rangle$ and $|\mathbb{i}_\bullet\rangle$ are taken to be a quark-gluon and a quark state, respectively, of fixed transverse positions and "+" momenta, and we trade $\psi$ for $\widetilde{\psi}$. The sum over the states becomes an integral with a measure consistent with the completeness and orthogonality relations (87), (88).

From now on, we will limit ourselves to order $g^2$. Hence we may write

$$Z_q \simeq 1 - \sum_{\sigma_1, i_1; \lambda_2, a_2} \int_{\substack{(x_1^\perp, k_1^+) \\ (x_2^\perp, k_2^+)}} \lim_{\phi_{\bullet q} \to q} \left| \widetilde{\psi}_{qg, \mathbb{i}(\phi_{\bullet q})}^{(1)\sigma_1, i_1; \lambda_2, a_2}(x_1^\perp, k_1^+; x_2^\perp, k_2^+) \right|^2. \tag{143}$$

Since the virtual terms we are computing are actually the corrections to the emission of a soft gluon, we can afford the eikonal approximation. Therefore, we may replace $\widetilde{\psi}^{(1)}$ by its expression in Eq. (138). After having rearranged the terms, we get

$$Z_q = 1 - \int_{\substack{(x_1^\perp, k_1^+) \\ (x_2^\perp, k_2^+)}} \lim_{\phi_{\bullet q} \to q} \left| \phi_{\bullet q}(x_1^\perp, k_1^+) \right|^2 \frac{g^2}{\pi^2} \left( \sum_{a_2=1}^{N^2-1} \sum_{i_1=1}^{N} t_{i_1 i_{\bullet 1}}^{a_2} t_{i_{\bullet 1} i_1}^{a_2} \right)$$

$$\times \left( \sum_{\sigma_1 \in \{-\frac{1}{2}, \frac{1}{2}\}} \delta_{\sigma_1 \sigma_{\bullet 1}} \right) \sum_{\lambda_2 \in \{-1, 1\}} \left| \frac{x_{21}^\perp \cdot \varepsilon_{(\lambda_2)}^{*\perp}}{(x_{21}^\perp)^2} \right|^2 \tag{144}$$

(The sums over $a_2$ and $i_1$ could have been understood, according to our notation conventions). Taking into account the limit $\phi_{\bullet q} \to q$ and the normalization of the wave function $\phi_{\bullet q}$ set in Eq. (137), the integral of $|\phi_{\bullet q}|^2$ over $(x_1^\perp, k_1^+)$ is trivially performed, and can formally be dropped, while $(x_1^\perp, k_1^+)$ are replaced by $(x_{\bullet 1}^\perp, k_{\bullet 1}^+)$. On the other hand, the color factor is simply deduced from the quadratic Casimir operator of the fundamental representation of SU($N$):

$$\sum_{a, i} (t^a)_{i i_\bullet'} (t^a)_{i_\bullet i} = C_F \delta_{i_\bullet i_\bullet'}, \qquad \text{where} \qquad C_F \equiv \frac{N^2 - 1}{2N} \tag{145}$$

(We refer the reader who is not familiar with this calculation to Appendix A.3). As for the polarizations, the sum over the helicity of the quark is just 1. The modulus squared summed over the polarizations of the gluon yields

$$\sum_{\lambda_2 \in \{-1,1\}} \left| \frac{x_{2\bullet1}^{\perp} \cdot \varepsilon_{(\lambda_2)}^{*\perp}}{(x_{2\bullet1}^{\perp})^2} \right|^2 = \frac{x_{2\bullet1}^{\perp l} \left( \sum_{\lambda_2} \varepsilon_{(\lambda_2)}^{*\perp l} \varepsilon_{(\lambda_2)}^{\perp l'} \right) x_{2\bullet1}^{\perp l'}}{\left( x_{2\bullet1}^{\perp 2} \right)^2} = \frac{1}{x_{2\bullet1}^{\perp 2}}, \qquad x_{2\bullet1}^{\perp} \equiv x_2^{\perp} - x_{\bullet1}^{\perp}, \quad (146)$$

where we have used the completeness (63) of the transverse polarization vectors. We eventually arrive at

$$Z_q \simeq 1 - \frac{2\alpha_s C_F}{\pi} \int^{k_{\bullet1}^+} \frac{dk_2^+}{k_2^+} \int \frac{d^2 x_2^{\perp}}{2\pi x_{2\bullet1}^{\perp 2}}, \qquad (147)$$

where we have defined $\alpha_s \equiv g^2/(4\pi)$.

The upper bound $k_{\bullet1}^+$ in the integration over $k_2^+$ follows from the fact that the emitted gluon cannot have "+" momentum larger than that of the radiating quark. Note that $Z_q$ may depend on the transverse position of the initial quark only through possible bounds on the $x_2^{\perp}$ integration.

We observe that both integrals, the one over the transverse position and the one over the "+" momentum, are logarithmic, and hence may a priori diverge if no upper and/or lower bounds are set. The integral over $x_2^{\perp}$ diverges both when the gluon is emitted collinearly to the quark ($x_{21}^{\perp} \to 0^{\perp}$), and at very large distances ($|x_{21}^{\perp}| \to \infty$). The one over $k_2^+$ diverges when the gluon is very soft compared to the quark. For the time being, we will stick with this formal expression, and come back to the discussion of the singularities in Sec. 3.3 below, after having worked out the case of the onium.

We have thus derived the probability amplitude for an initial quark to be found in a single-quark state with definite quantum numbers:

$$\widetilde{\psi}_{q,\mathbb{i}(q)}^{\sigma_1,i_1}(x_1^{\perp}, k_1^+) = \left\langle x_1^{\perp}, k_1^+ \middle| x_{\bullet1}^{\perp}, k_{\bullet1}^+ \right\rangle \delta_{\sigma_1 \sigma_{\bullet1}} \delta_{i_1 i_{\bullet1}} \sqrt{Z_q}, \qquad (148)$$

the order-$\alpha_s$ expansion of $Z_q$ being given by Eq. (147).

## 3.2 Forward elastic $S$-matrix for an onium

We now turn to the main initial state of interest in these lectures, namely the onium. An onium in its ground state is a color-neutral quark-antiquark pair. We shall denote its bare state by the ket $|\phi_{\bullet}\rangle$. A dressed state of it is a complicated superposition of globally color-neutral partonic states.

As announced, we will focus on the forward elastic $S$-matrix element for such an onium, that we shall denote by $S_{\phi}$, meaning that we identify both the initial and the final asymptotic states to $|\phi_{\bullet}\rangle$. It is obtained from $S_{\mathbb{f}\mathbb{i}}$ in Eq. (111) replacing the wave functions by those of the onium. We can expand the contributions to $S_{\phi}$ of the different Fock states, corresponding to the different terms in Eq. (111), as

$$S_{\phi} = S_{\phi}^{\bar{q}q} + S_{\phi}^{\bar{q}qg} + \cdots. \qquad (149)$$

The subscript denotes the state that is expanded, the superscripts the content of the partonic Fock state. Each of the terms in the right-hand side possesses a perturbative expansion in powers of $\alpha_s$:

$$S_{\phi}^{\bar{q}q} = \underbrace{S_{\phi}^{\bar{q}q(0)}}_{\mathcal{O}(\alpha_s^0)} + \underbrace{S_{\phi}^{\bar{q}q(1)}}_{\mathcal{O}(\alpha_s^1)} + \cdots, \qquad S_{\phi}^{\bar{q}qg} = \underbrace{S_{\phi}^{\bar{q}qg(1)}}_{\mathcal{O}(\alpha_s^1)} + \underbrace{S_{\phi}^{\bar{q}qg(2)}}_{\mathcal{O}(\alpha_s^2)} + \cdots. \qquad (150)$$

The order $\alpha_s^0$ is the classical limit, in which the quark-antiquark pair interacts with the external field in its bare state. We will call order $\alpha_s$ the "leading order", thereby meaning "leading-order quantum correction", and order $\alpha_s^2$ the "next-to-leading order".

### 3.2.1 Bare state and its contribution to the $S$-matrix element

We take the bare state of the onium as the following color-neutral superposition of quark-antiquark states localized in transverse position and possessing fixed "+" momenta, colors and polarizations:

$$|\phi_\bullet\rangle \equiv \frac{1}{2} \sum_{\sigma_{\bullet 0},\sigma_{\bullet 1}\in\{-\frac{1}{2},\frac{1}{2}\}} \int_{(x_{\bullet 1}^\perp,k_{\bullet 1}^+)}^{(x_{\bullet 0}^\perp,k_{\bullet 0}^+)} \phi_\bullet^{\sigma_{\bullet 0};\sigma_{\bullet 1}}(x_{\bullet 0}^\perp,k_{\bullet 0}^+;x_{\bullet 1}^\perp,k_{\bullet 1}^+)$$
$$\times \frac{1}{\sqrt{N}} \sum_{i_\bullet=1}^N \left|\bar{q};x_{\bullet 0}^\perp,k_{\bullet 0}^+,\sigma_{\bullet 0},i_\bullet\right\rangle \otimes \left|q;x_{\bullet 1}^\perp,k_{\bullet 1}^+,\sigma_{\bullet 1},i_\bullet\right\rangle . \tag{151}$$

The function $\phi_\bullet$ is the probability amplitude to find the asymptotic onium as a quark and an antiquark which have fixed transverse positions, "+" momenta and helicities, and are globally color neutral. Here, it is defined in such a way that $|\phi_\bullet\rangle$ be normalized to unity, $\langle\phi_\bullet|\phi_\bullet\rangle = 1$, which requires

$$\frac{1}{4} \sum_{\sigma_{\bullet 0},\sigma_{\bullet 1}\in\{-\frac{1}{2},\frac{1}{2}\}} \int_{(x_{\bullet 1}^\perp,k_{\bullet 1}^+)}^{(x_{\bullet 0}^\perp,k_{\bullet 0}^+)} \left|\phi_\bullet^{\sigma_{\bullet 0};\sigma_{\bullet 1}}(x_{\bullet 0}^\perp,k_{\bullet 0}^+;x_{\bullet 1}^\perp,k_{\bullet 1}^+)\right|^2 \equiv 1 . \tag{152}$$

The wave function of a bare onium in a quark-antiquark state of fixed quantum numbers thus reads, in the absence of quantum fluctuations:

$$\widetilde{\psi}_{\bar{q}q,\mathfrak{i}(\phi)}^{(0)\sigma_0,i_0;\sigma_1,i_1}(x_0^\perp,k_0^+;x_1^\perp,k_1^+) \equiv \left(\langle q;x_1^\perp,k_1^+,\sigma_1,i_1| \otimes \langle\bar{q};x_0^\perp,k_0^+,\sigma_0,i_0|\right)|\phi_\bullet\rangle$$
$$= \frac{1}{2\sqrt{N}} \phi_\bullet^{\sigma_0;\sigma_1}(x_0^\perp,k_0^+;x_1^\perp,k_1^+)\,\delta_{i_0 i_1} . \tag{153}$$

Let us compute the $S$-matrix element for the eikonal elastic scattering of an onium in its bare state $|\phi_\bullet\rangle$, namely the lowest-order contribution to the first term in Eq. (149). According to Eq. (111),

$$S_\phi^{\bar{q}q(0)} \equiv \sum_{\sigma_0,\sigma_1} \sum_{i_0',i_1',i_0,i_1} \int_{(x_1^\perp,k_1^+)}^{(x_0^\perp,k_0^+)} \left(\widetilde{\psi}_{\bar{q}q,\mathfrak{i}(\phi)}^{(0)\sigma_0,i_0';\sigma_1,i_1'}(x_0^\perp,k_0^+;x_1^\perp,k_1^+)\right)^*$$
$$\times \left[W^F(x_1^\perp)\right]_{i_1' i_1} \left[W^{\bar{F}}(x_0^\perp)\right]_{i_0' i_0} \times \widetilde{\psi}_{\bar{q}q,\mathfrak{i}(\phi)}^{(0)\sigma_0,i_0;\sigma_1,i_1}(x_0^\perp,k_0^+;x_1^\perp,k_1^+) . \tag{154}$$

The wave functions are given by Eq. (153), and we may express the anti-fundamental Wilson line in terms of the fundamental one through Eq. (A.12). We get

$$S_\phi^{\bar{q}q(0)} = \sum_{\sigma_0,\sigma_1;i_1,i_1'} \int_{(x_1^\perp,k_1^+)}^{(x_0^\perp,k_0^+)} \left(\frac{1}{2\sqrt{N}} \phi_\bullet^{\sigma_0;\sigma_1}(x_0^\perp,k_0^+;x_1^\perp,k_1^+)\right)^*$$
$$\times [W(x_1^\perp)]_{i_1' i_1}[W^\dagger(x_0^\perp)]_{i_1 i_1'} \times \left(\frac{1}{2\sqrt{N}} \phi_\bullet^{\sigma_0;\sigma_1}(x_0^\perp,k_0^+;x_1^\perp,k_1^+)\right) \tag{155}$$

(We have dropped the "$F$" and "$\bar{F}$" superscripts as there is no ambiguity). The sum over the colors amounts to taking the trace of the product of the Wilson lines. We arrive at the simple

expression

$$S_\phi^{\bar{q}q(0)} = \int_{\substack{(x_0^\perp, k_0^+) \\ (x_1^\perp, k_1^+)}} \frac{1}{4} \sum_{\sigma_0, \sigma_1} \left| \phi_\bullet^{\sigma_0;\sigma_1}(x_0^\perp, k_0^+; x_1^\perp, k_1^+) \right|^2 \times \frac{1}{N} \operatorname{Tr}\left[ W(x_1^\perp) W^\dagger(x_0^\perp) \right]. \tag{156}$$

If $\phi_\bullet$ were localizing the antiquark and the quark at the fixed positions $x_0^\perp$ and $x_1^\perp$ in the transverse plane, then the $S$-matrix element would just read

$$S_\phi^{\bar{q}q(0)} \longrightarrow S_{\bar{q}q}^{\bar{q}q(0)} \equiv S_\bullet(x_0^\perp, x_1^\perp) \equiv \frac{1}{N} \operatorname{Tr}\left[ W^\dagger(x_0^\perp) W(x_1^\perp) \right]. \tag{157}$$

For the generic wave packet $\phi_\bullet$, $S_\phi^{\bar{q}q(0)}$ coincides with the $S$-matrix element $S_\bullet(x_0^\perp, x_1^\perp)$, corresponding to the eikonal elastic scattering of a localized dipole, averaged over the positions of the endpoints of the dipole with the measure

$$\frac{d^2 x_0^\perp}{2\pi} \frac{d^2 x_1^\perp}{2\pi} \int \frac{dk_0^+}{2k_0^+} \frac{dk_1^+}{2k_1^+} \times \frac{1}{4} \sum_{\sigma_0, \sigma_1} \left| \phi_\bullet^{\sigma_0;\sigma_1}(x_0^\perp, k_0^+; x_1^\perp, k_1^+) \right|^2. \tag{158}$$

This is the probability to find the antiquark and the quark with given transverse coordinates $x_0^\perp$, $x_1^\perp$ respectively (up to $d^2 x_0^\perp$, $d^2 x_1^\perp$), independently of the other quantum numbers.

It is not difficult to realize that this factorization propagates to all Fock states, at all orders in perturbation theory. The expansion (149) can be organized as

$$S_\phi = \frac{1}{4} \sum_{\sigma_0, \sigma_1} \int_{\substack{(x_0^\perp, k_0^+) \\ (x_1^\perp, k_1^+)}} \left| \phi_\bullet^{\sigma_0;\sigma_1}(x_0^\perp, k_0^+; x_1^\perp, k_1^+) \right|^2 S_{\bar{q}q}(x_0^\perp, k_0^+; x_1^\perp, k_1^+), \tag{159}$$

with $\quad S_{\bar{q}q}(x_0^\perp, k_0^+; x_1^\perp, k_1^+) = S_{\bar{q}q}^{\bar{q}q}(x_0^\perp, k_0^+; x_1^\perp, k_1^+) + S_{\bar{q}q}^{\bar{q}qg}(x_0^\perp, k_0^+; x_1^\perp, k_1^+) \cdots,$

in terms of the different Fock state contributions to the $S$-matrix element $S_{\bar{q}q}$ for an initial localized color-singlet $\bar{q}q$ pair. We have anticipated that the $S_{\bar{q}q}$'s will not depend on the helicities of the asymptotic quark and antiquark, because the eikonal approximation will be used for the interaction with the external field and also for the calculation of the quantum corrections. Each term admits a perturbative expansion as in Eq. (150): $S_{\bar{q}q}^{\bar{q}q} = S_{\bar{q}q}^{\bar{q}q(0)} + S_{\bar{q}q}^{\bar{q}q(1)} + \cdots$ etc, where the superscript indicates the order in $\alpha_s$, in addition to the partonic content of the interaction state.

We shall now address the lowest-order quantum correction. We start with a discussion of the Fock state of the onium, before turning to the calculation of the $S$-matrix element.

### 3.2.2 Fock state expansion to order $g^2$

The expansion of the interaction state of an onium in terms of partonic fluctuations schematically reads

$$|\mathtt{i}(\phi)\rangle = \sum \int |\bar{q}q\rangle \, \psi_{\bar{q}q, \mathtt{i}(\phi)} + \sum \int |\bar{q}qg\rangle \, \psi_{\bar{q}qg, \mathtt{i}(\phi)} + \{\text{higher-multiplicity states}\}. \tag{160}$$

Although we could in principle be satisfied with the computation of the $S$-matrix element for a localized color dipole, and hence consider the partonic expansion of such a state instead of that of a wave packet, it will prove convenient to keep wave packets at this stage. As usual, the discrete sums run over the helicities and the colors, and the integrals go over the 3-momenta – or alternatively, over the transverse positions and "+" components of the momenta if one is working in the mixed representation. At order $g^2$, $\psi_{\bar{q}q, \mathtt{i}(\phi)}$ includes virtual corrections, effectively in the form of a constant $\sqrt{Z_{\bar{q}q}}$ multiplying Eq. (153), that we will need to evaluate. (Although $Z_{\bar{q}q}$ does not renormalize a single-particle state, we shall anyway call it a "renormalization constant"). As for $\psi_{\bar{q}qg, \mathtt{i}(\phi)}$, we will denote by $\psi_{\bar{q}qg, \mathtt{i}(\phi)}^{(1)}$ the lowest-order contribution $\mathcal{O}(g)$ to it.

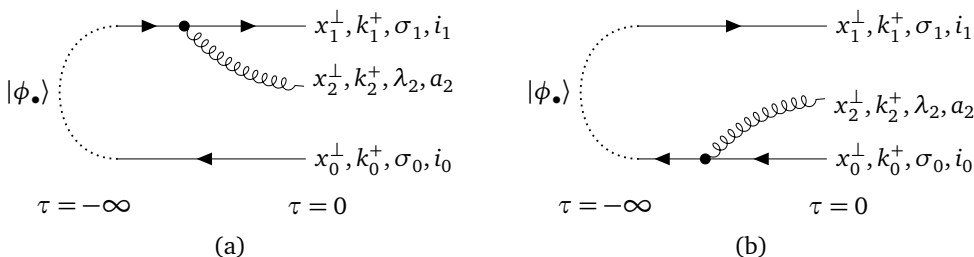

Figure 4: Diagrams contributing to the wave function $\psi^{(1)}_{\bar{q}qg,\mathtt{i}(\phi)}$ in Eq. (161). The dotted lines that connect the quark and the antiquark in the initial state $|\phi_\bullet\rangle$ (at $\tau = -\infty$) recall that these particles carry the same color index.

**Quark-antiquark-gluon state at lowest perturbative order.** The wave function $\psi^{(1)}_{\bar{q}qg,\mathtt{i}(\phi)}$ can easily be deduced from that of the quark (resp. antiquark) into a quark-gluon (resp. antiquark-gluon) pair. Indeed, going immediately to the mixed representation,

$$
\begin{aligned}
\widetilde{\psi}^{(1)}_{\bar{q}qg,\mathtt{i}(\phi)} &= \left(\left(\langle q; x_1^\perp, k_1^+, \sigma_1, i_1| \otimes \langle \bar{q}; x_0^\perp, k_0^+, \sigma_0, i_0| \otimes \langle g; x_2^\perp, k_2^+, \lambda_2, a_2|\right)|\mathtt{i}(\phi_\bullet)\rangle\right)\Big|_{\mathcal{O}(g)} \\
&\simeq \frac{1}{2\sqrt{N}} \sum_{\sigma_{\bullet 0}, \sigma_{\bullet 1}, i_\bullet} \int_{\substack{(x_{\bullet 0}^\perp, k_{\bullet 0}^+) \\ (x_{\bullet 1}^\perp, k_{\bullet 1}^+)}} \phi_\bullet^{\sigma_{\bullet 0}; \sigma_{\bullet 1}}(x_{\bullet 0}^\perp, k_{\bullet 0}^+; x_{\bullet 1}^\perp, k_{\bullet 1}^+) \\
&\quad \times \left[\widetilde{\psi}^{(1)\sigma_1, i_1; \lambda_2, a_2}_{qg, \mathtt{i}(q)}(x_1^\perp, k_1^+; x_2^\perp, k_2^+) \langle \bar{q}; x_0^\perp, k_0^+, \sigma_0, i_0| \bar{q}; x_{\bullet 0}^\perp, k_{\bullet 0}^+, \sigma_{\bullet 0}, i_\bullet\rangle \right. \\
&\quad \left. + \langle q; x_1^\perp, k_1^+, \sigma_1, i_1| q; x_{\bullet 1}^\perp, k_{\bullet 1}^+, \sigma_{\bullet 1}, i_\bullet\rangle \, \widetilde{\psi}^{(1)\sigma_0, i_0; \lambda_2, a_2}_{\bar{q}g, \mathtt{i}(\bar{q})}(x_0^\perp, k_0^+; x_2^\perp, k_2^+)\right].
\end{aligned}
\tag{161}
$$

In the left-hand side, we have not written down explicitly the variables upon which the wavefunction depends: they are understood from the right-hand side of the equation. In words, the two terms making up the latter, that we may denote by $\widetilde{\psi}^{(a)}_{\bar{q}qg,\mathtt{i}(\phi)}$ and $\widetilde{\psi}^{(b)}_{\bar{q}qg,\mathtt{i}(\phi)}$ respectively, correspond to the two possibilities to add a gluon to the state of the onium. Either the gluon is radiated off the quark (the corresponding probability amplitude is given in Eq. (135), up to appropriate replacements of variables) and the antiquark is spectator; or the gluon is radiated off the antiquark (the probability amplitude is deduced from Eq. (136)) and the quark is spectator. These two cases are represented diagrammatically in Fig. 4a and b respectively. We get the following expression:

$$
\begin{aligned}
&\widetilde{\psi}^{(1)\sigma_0, i_0; \sigma_1, i_1; \lambda_2, a_2}_{\bar{q}qg,\mathtt{i}(\phi)}(x_0^\perp, k_0^+; x_1^\perp, k_1^+; x_2^\perp, k_2^+) \\
&\simeq \frac{1}{2\sqrt{N}} \phi_\bullet^{\sigma_0; \sigma_1}(x_0^\perp, k_0^+; x_1^\perp, k_1^+) \times (t^{a_2})_{i_1 i_0} \frac{ig}{\pi} \left(\frac{x_{21}^\perp \cdot \varepsilon^{\perp *}_{(\lambda_2)}}{x_{21}^{\perp 2}} - \frac{x_{20}^\perp \cdot \varepsilon^{\perp *}_{(\lambda_2)}}{x_{20}^{\perp 2}}\right).
\end{aligned}
\tag{162}
$$

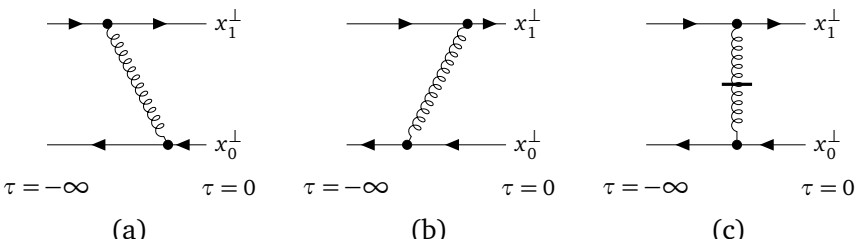

Figure 5: Virtual corrections to the dipole wave function. Only the connected graphs are shown. There are causal diagrams, (a) and (b), as well as an instantaneous exchange graph (c). [The term in the Hamiltonian whose matrix element is needed to compute this last graph can be found in Eq. (119)].

**Virtual corrections to the bare onium state.** As in the case of the quark initial state, we may get the virtual corrections, namely an expression for $Z_{\bar{q}q}$, just by enforcing unitarity:

$$Z_{\bar{q}q} \simeq 1 - \frac{1}{4} \sum_{\sigma_0,\sigma_1,\lambda_2} \sum_{i_0,i_1,a_2} \int_{\substack{(x_0^\perp,k_0^+) \\ (x_1^\perp,k_1^+) \\ (x_2^\perp,k_2^+)}} \lim_{\phi_\bullet \to \bar{q}q} \left| \widetilde{\psi}_{\bar{q}qg,\mathrm{i}(\phi)}^{(1)\sigma_0,i_0;\sigma_1,i_1;\lambda_2,a_2}(x_0^\perp,k_0^+;x_1^\perp,k_1^+;x_2^\perp,k_2^+) \right|^2$$

$$= 1 - \int_{\substack{(x_0^\perp,k_0^+) \\ (x_1^\perp,k_1^+)}} \frac{1}{4} \sum_{\sigma_0,\sigma_1} \lim_{\phi_\bullet \to \bar{q}q} \left| \phi_\bullet^{\sigma_0;\sigma_1}(x_0^\perp,k_0^+;x_1^\perp,k_1^+) \right|^2 \times \frac{g^2}{\pi^2} \left( \frac{1}{N} \sum_{i_0,i_1,a_2} (t^{a_2})_{i_1 i_0}(t^{a_2})_{i_0 i_1} \right) \quad (163)$$

$$\times \int \frac{dk_2^+}{k_2^+} \int_{(x_2^\perp,k_2^+)} \sum_{\lambda_2} \left| \frac{x_{21}^\perp \cdot \varepsilon_{(\lambda_2)}^{\perp*}}{x_{21}^{\perp 2}} - \frac{x_{20}^\perp \cdot \varepsilon_{(\lambda_2)}^{\perp*}}{x_{20}^{\perp 2}} \right|^2 .$$

The color factor is calculated in the same way as in the case of the single quark, see Eq. (145), and the modulus squared summed over the polarizations of the gluon stems from a calculation transposed from that of Eq. (146):

$$\sum_{\lambda=\pm1} \left| \frac{x_{21}^\perp \cdot \varepsilon_{(\lambda)}^{\perp*}}{x_{21}^{\perp 2}} - \frac{x_{20}^\perp \cdot \varepsilon_{(\lambda)}^{\perp*}}{x_{20}^{\perp 2}} \right|^2 = \frac{1}{x_{21}^{\perp 2}} - \frac{x_{21}^\perp \cdot x_{20}^\perp}{x_{21}^{\perp 2} x_{20}^{\perp 2}} - \frac{x_{20}^\perp \cdot x_{21}^\perp}{x_{21}^{\perp 2} x_{20}^{\perp 2}} + \frac{1}{x_{20}^{\perp 2}} = \frac{x_{10}^{\perp 2}}{x_{02}^{\perp 2} x_{21}^{\perp 2}} . \quad (164)$$

We then use Eq. (152) in order to perform the integral over the kinematical variables of the quark and of the antiquark. All in all, we arrive at the following expression for $Z_{\bar{q}q}$, in the limit of a localized onium:

$$Z_{\bar{q}q} \simeq 1 - \frac{2\alpha_s C_F}{\pi} \int \frac{dk_2^+}{k_2^+} \int \frac{d^2 x_2^\perp}{2\pi} \frac{x_{10}^{\perp 2}}{x_{02}^{\perp 2} x_{21}^{\perp 2}} \quad (165)$$

(We have traded $x_{\bullet i}^\perp$ for $x_i^\perp$ etc, for better lisibility). As for the renormalization constant of the one-quark state, the $k_2^+$ and $x_2^\perp$ integrations are plagued by logarithmic divergences. We have not put any bound on these integrals: since they are subtle, we will content ourselves with formal expressions here, and dedicate Sec. 3.3 below to a detailed discussion of the integration limits.

Note that $Z_{\bar{q}q}$ does not only amount to the sum of the self-energies of the quark and of the antiquark which make up the renormalization factor $Z_q$ (or $Z_{\bar{q}}$, which would be the same), in which the gluon is emitted and absorbed by the same fermion. Indeed, there are connected graphs in which a gluon is exchanged between the quark and the antiquark;[13] see Fig. 5. The

---

[13]Note that the graphs in Fig. 5 contribute to the renormalization factor $Z_{\bar{q}q}$ of a dipole when the gluon couples eikonally. Indeed, if the gluon is soft, its emission and absorption does not change the quantum numbers of the quark and of the antiquark, except for the color; but the latter is summed over.

disconnected graphs (not shown) correspond to the first and last terms in the second equality in Eq. (164). The connected ones correspond to the second and third terms.

### 3.2.3 $S$-matrix element at order $\alpha_s$

The calculation starts again from Eq. (111) and proceeds along the same lines as the calculation that led to Eq. (156), except that we need to include the virtual corrections to the $\bar{q}q$ Fock state that we have just calculated, as well as the $\bar{q}qg$ state whose wave function was calculated in Eq. (162).

Let us consider the contribution of these Fock states separately. We will show the calculation of $S_{\bar{q}q}$, namely for a localized color dipole: once this is known, it is trivial to get $S_\phi$ using Eq. (159).

**Contribution of the $\bar{q}q$ Fock state.** The wave function corresponding to the probability amplitude to find an initial localized dipole in a quark-antiquark state at the time of the interaction, that is needed in Eq. (111), is obtained by simply multiplying the order-zero wave function by the constant $\sqrt{Z_{\bar{q}q}}$. Then, we easily see that

$$S_{\bar{q}q}^{\bar{q}q} = Z_{\bar{q}q} S_\bullet . \tag{166}$$

To get an expression up to order $\alpha_s$, we just need to replace $Z_{\bar{q}q}$ by Eq. (165).

**Contribution of the $\bar{q}qg$ Fock state.** Starting again from Eq. (111), we write

$$
\begin{aligned}
S_\phi^{\bar{q}qg(1)} \equiv \sum_{\sigma_0,\sigma_1,\lambda_2}\sum_{i_0',i_1',a_2',i_0,i_1,a_2} \int_{\substack{(x_0^\perp,k_0^+)\\(x_1^\perp,k_1^+)\\(x_2^\perp,k_2^+)}} &\left( \widetilde{\psi}_{\bar{q}qg,\mathfrak{i}(\phi)}^{(1)\sigma_0,i_0';\sigma_1,i_1';\lambda_2,a_2'}(x_0^\perp,k_0^+;x_1^\perp,k_1^+;x_2^\perp,k_2^+) \right)^* \\
&\times \left[ W^F(x_1^\perp) \right]_{i_1'i_1} \left[ W^{\bar{F}}(x_0^\perp) \right]_{i_0'i_0} \left[ W^A(x_2^\perp) \right]_{a_2'a_2} \\
&\times \widetilde{\psi}_{\bar{q}qg,\mathfrak{i}(\phi)}^{(1)\sigma_0,i_0;\sigma_1,i_1;\lambda_2,a_2}(x_0^\perp,k_0^+;x_1^\perp,k_1^+;x_2^\perp,k_2^+).
\end{aligned}
\tag{167}
$$

We replace the wave functions by Eq. (162). After a straightforward calculation, we obtain an expression that we can write as the convolution of the squared wave function of the bare onium in a $\bar{q}q$ pair, and of the contribution to $S_{\bar{q}q}$ of the $\bar{q}qg$ Fock state at the lowest perturbative order. The latter reads

$$S_{\bar{q}q}^{\bar{q}qg(1)} = \frac{\alpha_s N}{\pi} \int \frac{dk_2^+}{k_2^+} \int \frac{d^2x_2^\perp}{2\pi} \frac{x_{10}^{\perp 2}}{x_{02}^{\perp 2} x_{21}^{\perp 2}} \times S_{\bar{q}qg\bullet}(x_0^\perp,x_1^\perp,x_2^\perp), \tag{168}$$

where we have grouped the factors that account for the interaction of a given quark-antiquark-gluon singlet by defining the new function

$$S_{\bar{q}qg\bullet}(x_0^\perp,x_1^\perp,x_2^\perp) \equiv \frac{2}{N^2} \text{Tr}\left[ W(x_1^\perp) t^a W^\dagger(x_0^\perp) t^b \right] \widetilde{W}(x_2^\perp)_{ba} . \tag{169}$$

To drop a few indices, we have again denoted by $W$ the Wilson line in the defining representation and by $\widetilde{W}$ that in the adjoint representation. We have arranged the terms in such a way that the overall numerical factor in Eq. (168) is $\alpha_s N/\pi$. Anticipating that this particular combination of constants will appear at all orders, we shall introduce the traditional notation

$$\bar{\alpha} \equiv \frac{\alpha_s N}{\pi} . \tag{170}$$

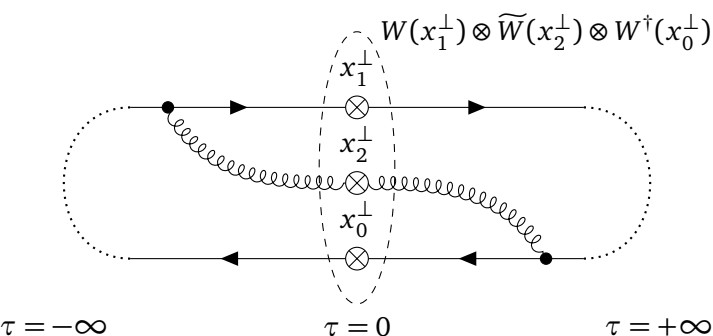

Figure 6: One of the four real diagrams of order $\alpha_s$ contributing to $S_{\bar{q}q}$, for one particular realization of the quantum numbers of the interaction state. The other three diagrams are obtained by exhausting all possibilities of hooking the gluon, at positive and negative light cone times, to the quark or to the antiquark. The crossed dots represent the insertion of Wilson lines.

One of the four diagrams contributing to $S_\phi^{\bar{q}qg(1)}$, resulting of the interference of the wave functions $\psi^{(b)}$ and $\psi^{(a)}$ (see Fig. 4), is displayed in Fig. 6. It reads

$$S_{\bar{q}q}^{\bar{q}qg(1)}\Big|_{\text{Fig. 6}} = \bar{\alpha} \int \frac{dk_2^+}{k_2^+} \int \frac{d^2x_2^\perp}{2\pi} \left(-\frac{x_{21}^\perp \cdot x_{20}^\perp}{x_{21}^{\perp 2} x_{20}^{\perp 2}}\right) \times S_{\bar{q}qg\bullet}(x_0^\perp, x_1^\perp, x_2^\perp). \tag{171}$$

We have organized the formula in such a way as to make the interaction factor $S_{\bar{q}qg\bullet}$ defined in Eq. (169) appear. The two-dimensional transverse factors are the same as the ones in the second term in the second member of equation (164).

**Reduction of the color structure.** We may actually express $S_{\bar{q}qg\bullet}$ in terms of $S_\bullet$ by manipulating Eq. (169) using the color algebra. It is useful to introduce graphical representations for such operations.

From the point of view of the color, all the graphs that we will need to evaluate contributing to the forward elastic $S$-matrix element for an onium, at any order in perturbation theory but in the high-energy limit, will have exactly one quark loop, to which gluons may attach. In addition, there are Wilson line insertions on the quark and on the antiquark, as well as on some gluon lines. An expression for the color factor (including the Wilson lines) associated to a given Feynman graph can be deduced from the topology of the graph using the following simple rules:

- Wind the quark loop in the opposite direction to that indicated by the arrow. Associate to each vertex a factor $t^{a_1}$, $t^{a_2}$ etc, and to each insertion of a Wilson line a factor $W$ or $W^\dagger$ (evaluated at the appropriate transverse position), depending on whether it occurs on a quark or an antiquark. Take the trace of the obtained SU($N$) matrix product.

- Replace three-gluon vertices by generators $-if^{b_ic_id_i}$, where the sequence of indices run counter-clockwise, and Wilson-line insertions on the gluon line propagators by $\widetilde{W}_{e_if_i}$, where the first index is assigned to the leg to the right of the insertion.

- Identify the color indices that are located at both ends of the same gluon propagators.

We may check that we indeed obtain the factor $S_{\bar{q}qg\bullet}$ in Eq. (169) from the Feynman graph shown in Fig. 6 applying the rules just stated, up to the normalization $2/N^2$ which we put in the definition of $S_{\bar{q}qg\bullet}$. How to represent systematically a SU($N$) group-theoretical factor by

a graph and how to reduce the former by manipulating the latter is explained in some more detail in Appendix A.3.

Now, algebraic identities on tensors can be translated into graphical manipulations. For example, the definition of the Lie bracket is equivalent to the following graphical substitution:

$$if^{abc}t^c = [t^a, t^b] \quad \longrightarrow \quad \text{(diagram)} \tag{172}$$

Each diagram that contains the "elementary brick" depicted in the left-hand side of the pictorial equation may be replaced by the difference of the two diagrams obtained by trading this subdiagram for the ones in the right-hand side of the equation.

Going back to Eq. (169), we may identify $S_{\bar{q}qg\bullet}$ to the following color diagram, which has the same topology as the Feynman graph in Fig. 6:

$$S_{\bar{q}qg\bullet}(x_0^\perp, x_1^\perp, x_2^\perp) = \frac{2}{N^2} \; \text{(diagram)} . \tag{173}$$

The crossed circles represent the insertion of a Wilson line: a fundamental one $W$ or $W^\dagger$ when it is placed on a fermion or antifermion propagator, or an adjoint one $\widetilde{W}$ when on a gluon propagator. We have connected the quark and antiquark of the initial and final dipoles by an index line, as they carry the same color, over which we sum through the trace. Note that this diagram is now only meant to represent the group-theoretical factors: the "time ordering" of the vertices is irrelevant, and hence, the latter can be moved around without altering the algebraic expression, provided one does not mix them up. We are going to perform the algebraic and graphical transformation in parallel for this simple case, and address the higher-order calculations in Sec. 4 purely graphically.

We shall first trade the adjoint Wilson line appearing in Eq. (169) for fundamental ones. To this aim, we use well-known relation

$$t^b \widetilde{W}_{ba} = W t^a W^\dagger , \tag{174}$$

see Appendix A.1 for a proof. This identity may be represented pictorially by the following transformation:

$$\text{(diagram)} = \text{(diagram)} . \tag{175}$$

The quark and the antiquark lines that replace the gluon line should actually be on the top of each other: we have separated them slightly for lisibility. Inserting this identity with $W \to W(x_2^\perp)$ into Eq. (169) enables us to rewrite the latter as

$$S_{\bar{q}qg\bullet}(x_0^\perp, x_1^\perp, x_2^\perp) = \frac{2}{N^2} \text{Tr}\left[ t^a W^\dagger(x_0^\perp) W(x_2^\perp) t^a W^\dagger(x_2^\perp) W(x_1^\perp)\right] = \frac{2}{N^2} \; \text{(diagram)} . \tag{176}$$

Breaking the trace into a sum over fundamental color indices of a product of matrix elements, this also reads

$$S_{\bar{q}qg\bullet}(x_0^\perp, x_1^\perp, x_2^\perp) = \frac{2}{N^2} (t^a)_{i'i} W^\dagger(x_0^\perp)_{ik} W(x_2^\perp)_{kj} (t^a)_{jj'} W^\dagger(x_2^\perp)_{j'k'} W(x_1^\perp)_{k'i'} . \tag{177}$$

We now use the Fierz identity,

$$(t^a)_{i'i}(t^a)_{jj'} = \frac{1}{2}\delta_{ji}\delta_{i'j'} - \frac{1}{2N}\delta_{jj'}\delta_{i'i} , \tag{178}$$

that we can depict graphically as the operation of substituting a gluon line by either a double fermion/antifermion line or no line at all,

$$
\begin{array}{c}
i \quad\quad i' \\
\\
j \quad\quad j'
\end{array}
= \frac{1}{2}
\begin{array}{c}
i \quad\quad i' \\
\\
j \quad\quad j'
\end{array}
- \frac{1}{2N}
\begin{array}{c}
i \quad\quad i' \\
\\
j \quad\quad j'
\end{array}
\tag{179}
$$

to cast this expression in the form of the sum of two terms:

$$
S_{\bar{q}qg\bullet}(x_0^\perp, x_1^\perp, x_2^\perp) = \frac{1}{N^2} \operatorname{Tr}\left[W^\dagger(x_0^\perp)W(x_2^\perp)\right] \operatorname{Tr}\left[W^\dagger(x_2^\perp)W(x_1^\perp)\right] - \frac{1}{N^3} \operatorname{Tr}\left[W^\dagger(x_0^\perp)W(x_1^\perp)\right].
\tag{180}
$$

We have used unitarity $W(x_2^\perp)W^\dagger(x_2^\perp) = \mathbb{I}$ to simplify the second term in the right-hand side. The combination of these operations is represented graphically as

$$
S_{\bar{q}qg\bullet}(x_0^\perp, x_1^\perp, x_2^\perp) = \frac{1}{N^2}
\begin{array}{c}\text{[diagram]}\end{array}
- \frac{1}{N^3}
\begin{array}{c}\text{[diagram]}\end{array}.
\tag{181}
$$

We see that the left-over objects that interact with the external field are all color dipoles. We can thus express $S_{\bar{q}qg\bullet}$ in terms of $S_\bullet$ only, evaluated at the relevant transverse positions,

$$
S_{\bar{q}qg\bullet}(x_0^\perp, x_1^\perp, x_2^\perp) = S_\bullet(x_0^\perp, x_2^\perp) S_\bullet(x_2^\perp, x_1^\perp) - \frac{1}{N^2} S_\bullet(x_0^\perp, x_1^\perp).
\tag{182}
$$

Note that the second term is formally suppressed by $1/N^2$ relatively to the first term in the limit in which the number $N$ of colors is taken large (and assuming that $S_\bullet$ does not have any dependence on $N$). This means that a quark-antiquark-gluon fluctuation scatters effectively like two independent quark-antiquark dipoles. This observation will allow for a drastic simplification of the computation of higher orders in the $N \to \infty$ limit.

Our final expression for the $\bar{q}qg$ Fock state contribution to the $S$-matrix element reads, at order $\alpha_s$,

$$
S_{\bar{q}q}^{\bar{q}qg(1)} \equiv \bar{\alpha} \int \frac{dk_2^+}{k_2^+} \int \frac{d^2 x_2^\perp}{2\pi} \frac{x_{01}^{\perp 2}}{x_{02}^{\perp 2} x_{21}^{\perp 2}} \left[ S_\bullet(x_0^\perp, x_2^\perp) S_\bullet(x_2^\perp, x_1^\perp) - \frac{1}{N^2} S_\bullet(x_0^\perp, x_1^\perp) \right].
\tag{183}
$$

Let us comment that the dominant term in the limit $N \to \infty$, namely

$$
S_{\bar{q}q}^{\bar{q}qg(1)} \Big|_{N\to\infty} \equiv \bar{\alpha} \int \frac{dk_2^+}{k_2^+} \int \frac{d^2 x_2^\perp}{2\pi} \frac{x_{01}^{\perp 2}}{x_{02}^{\perp 2} x_{21}^{\perp 2}} S_\bullet(x_0^\perp, x_2^\perp) S_\bullet(x_2^\perp, x_1^\perp),
\tag{184}
$$

amounts to the product of the $S$-matrix elements representing the independent scattering of two dipoles, integrated over the position of one of their endpoints. The integration measure,

$$
\bar{\alpha} \left( \int \frac{dk_2^+}{k_2^+} \right) \frac{d^2 x_2^\perp}{2\pi} \frac{x_{01}^{\perp 2}}{x_{02}^{\perp 2} x_{21}^{\perp 2}},
\tag{185}
$$

may be interpreted as the probability that the initial antiquark-quark pair at position $(x_0^\perp, x_1^\perp)$ is replaced by two color dipoles at positions $(x_0^\perp, x_2^\perp), (x_2^\perp, x_1^\perp)$, $x_2^\perp$ being found within the infinitesimal surface element $d^2 x_2^\perp$; see Fig. 7 for a diagrammatic representation.

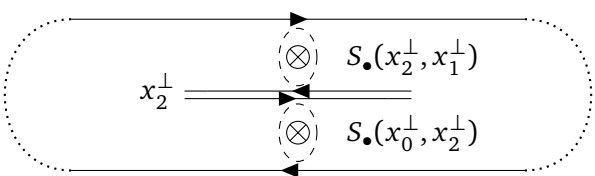

Figure 7: Scattering of quark-antiquark-gluon fluctuation in the large-$N$ limit. The displayed graph represents the sum of the diagram represented in Fig. 6 and of its three siblings, in the large-number-of-color limit in which a gluon can be thought of as a zero-size quark-antiquark pair. The crossed circles now stand for the interaction of the corresponding bare dipoles with the external field.

**Putting all terms together.** All in all, the order-$\alpha_s$ contribution to the $S$-matrix element for the forward elastic scattering of a localized quark-antiquark pair reads

$$S_{\bar{q}q}(x_0^\perp, k_0^+; x_1^\perp, k_1^+)\big|_{\mathcal{O}(\alpha_s)} = \bar{\alpha} \int \frac{dk_2^+}{k_2^+} \int \frac{d^2x_2^\perp}{2\pi} \frac{x_{01}^{\perp 2}}{x_{02}^{\perp 2} x_{21}^{\perp 2}} \left[ S_\bullet(x_0^\perp, x_2^\perp) S_\bullet(x_2^\perp, x_1^\perp) - S_\bullet(x_0^\perp, x_1^\perp) \right]. \quad (186)$$

Interestingly enough, had we kept consistently only the leading terms in the limit of a large number of colors, we would have obtained this very same expression for $S$ at this order. (This attractive property does however not go over to higher orders of perturbation theory; see Sec. 4 below).

We have already commented that the gluon emission kernel may be plagued with divergences. Let us discuss the potential divergences in this expression.

### 3.3 Singularities in $S$

#### 3.3.1 Properties of forward elastic $S$-matrix elements

In order to be able to analyze the singularities in Eq. (186), we need some knowledge of the analytic properties of $S_\bullet$.

It is well-known lore that in the regime of high energies, the forward elastic $S$-matrix elements are dominantly real, and bounded between 0 and 1. Moreover, the scattering amplitude,

$$T_{\bar{q}q}(x_0^\perp, x_1^\perp) \equiv 1 - S_{\bar{q}q}(x_0^\perp, x_1^\perp), \quad (187)$$

for an onium whose constituents have a fixed transverse position, turns out to vanish quadratically with the distance in the transverse plane between the quark and the antiquark. This property is known as color transparency. For the bare amplitude,

$$T_\bullet(x_0^\perp, x_1^\perp) \underset{x_0^\perp \to x_1^\perp}{\propto} Q_A^2 x_{01}^{\perp 2}. \quad (188)$$

We have introduced a momentum $Q_A$ on dimensional grounds. It is called the (nuclear) saturation scale, and characterizes the external potential (i.e., in the physical situation of interest, the target nucleus). This scale is proportional to the inverse transverse size of the onium above which $T_\bullet$ "saturates" to unity, meaning that its state gets destroyed in the interaction with probability of order one.

We are not going to prove these features, as it would take us too far from the focus of this review: we shall refer instead the reader to e.g. [47] (Sec. 3) or [5] for background on the general properties of hadronic amplitudes at very high energies. However, that the bare amplitude $S_\bullet$ is bounded is easy to check, starting from its definition (157). According to the latter, $S_\bullet$ identifies with the Frobenius inner product of (unitary) $N \times N$ matrices, normalized by $1/N$. The Cauchy-Schwartz inequality then yields the bound $\left| S_\bullet(x_0^\perp, x_1^\perp) \right| \leq 1$.

The fact that $T_{\bar{q}q}$ vanishes in the limit $x_0^\perp \to x_1^\perp$ in which the quark and the antiquark get close together is easy to understand physically: indeed, a color dipole of size zero is an effectively color-neutral point particle, and hence has no interaction with the color field.

### 3.3.2 Collinear and soft singularities

We shall first examine the possible infinities due to the integration over the transverse position in Eq. (186). Since the $S$-matrix elements of interest are bounded, the singularities are at most those of the integral

$$\int d^2 x_2^\perp \frac{x_{01}^{\perp 2}}{x_{02}^{\perp 2} x_{21}^{\perp 2}}, \qquad (189)$$

appearing in the inclusive gluon emission probability obtained by integrating (185). This integral converges at infinity: indeed, when the distances in the transverse plane between the trajectories of the gluon and of the quark or the antiquark, $|x_{21}^\perp|$ and $|x_{02}^\perp|$ respectively, are large compared to the distance $|x_{01}^\perp|$ between the quark and the antiquark, the integrand behaves as $x_{01}^{\perp 2}/(x_2^{\perp 2})^2$. Physically, this is a consequence of the global color neutrality of the dipole. Technically, the emission probability off a single quark is singular in the infrared; but in the case of a color-neutral quark-antiquark pair, the divergence cancels between the emission diagram off the quark and off the antiquark, thanks to the "−" sign difference between the two corresponding matrix elements that we have already stressed; see Eqs. (120).

Nevertheless, the transverse integral would a priori diverge logarithmically due to the short-size configurations in which $x_2^\perp$ gets close to $x_1^\perp$ or to $x_0^\perp$, namely when the gluon is emitted collinearly to the quark or to the antiquark. The renormalization constant $Z_{\bar{q}q}$ suffers from this ultraviolet divergence, as we would have expected. However, for these kinematical configurations and given that $S_\bullet(x_0^\perp, x_2^\perp) \to 1$ when $x_2^\perp \to x_0^\perp$ (and the same with $x_0^\perp$ replaced by $x_1^\perp$), the singularities cancel between the two terms of the integrand in Eq. (186), which correspond to the real and virtual contributions respectively. Hence the integral over $x_2^\perp$ in the expression of this physical observable is overall finite.

### 3.3.3 Integration over the "+" momentum and rapidity divergence

The dependence of $S_{\bar{q}q}$ on the "+" components of the momenta amounts to a logarithmic factor $\int dk_2^+/k_2^+$ in Eq. (186). We first need to discuss the bounds on this integral.

**Integration limits.** The "+" momentum of the gluon is necessarily bounded from above: it cannot be larger than that of the quark or the antiquark that has emitted it. Hence the integration cannot extend above $k_0^+$ or $k_1^+$. Let us set it conservatively to $\min(k_0^+, k_1^+)$: we will come back to this prescription below.

On the low-momentum side, we note that in the LCPT calculation of the wave function, nothing prevents a priori that $\bar{q}qg$ partonic states may be found in configurations in which the "+" momentum of the gluon is vanishingly small. Hence the integral over $k_2^+$ diverges a priori, due to the lack of a non-zero lower cutoff.

The fact that arbitrarily-soft gluons may contribute to the interaction amplitude stems from the initial assumption that the rapidity (or, equivalently, the "+" momentum) of the onium is infinite. Indeed, in this strict limit and from the point of view of the latter, the external potential acts in an infinitely thin slice of space, and hence has infinitely-accurate temporal resolution of any object crossing it at non-zero rapidity. It takes a literally instantaneous snapshot of the fluctuations of the onium. This is of course not right: crossing the interaction region takes a time on the order of the diameter of a nucleus, namely a few femtometers when the quantities are expressed in natural units. Only fluctuations that have coherence times larger than this

may be "seen", and hence should be considered in the summation in Eq. (186). We are going to check that this physical requirement is actually equivalent to enforcing a lower cutoff on $k_2^+$, by evaluating the coherence time of a quantum fluctuation.

**Lifetime of a fluctuation.** A two-particle fluctuation of a single asymptotic particle has a typical lifetime $\Delta\tau$ on the order of the inverse difference between the energy of the initial particle, of momentum $p_\bullet$, and that of the fluctuation, whose constituents have momenta $k$ and $p$. In the limit in which the particle of momentum $k$ is very soft compared to the other one, according to Eq. (127),

$$\Delta\tau = \left| \frac{1}{p_\bullet^- - k^- - (p-k)^-} \right| \simeq \frac{1}{k^-} = \frac{2k^+}{(k^\perp)^2}\,. \tag{190}$$

This lifetime should be larger than the size of the region in which the potential is acting, which implies a lower bound on the momentum of the emitted particle as follows:

$$\Delta\tau > 2R \quad \Longrightarrow \quad k^- < \frac{1}{2R} \quad \Longrightarrow \quad k^+ > R(k^\perp)^2 \sim RQ^2\,, \tag{191}$$

where $R$ is the typical size of the nucleus in its restframe. The factor 2 is for mere convenience: we do not control overall numerical $\mathcal{O}(1)$ factors in such estimates. $Q$ is the common scale of the transverse momenta.

In practice, $R$ is of order a few $1/\Lambda_{\text{QCD}}$. $Q$ instead is assumed to be typically in the multi-GeV range, namely much larger than $\Lambda_{\text{QCD}} \sim 200$ MeV. We also keep in mind that the "+" components of the initial objects are the largest momentum scales in this process: $k_0^+$, $k_1^+$ are assumed much larger than $Q$, and than $RQ^2$ as well.

**Leading logarithm.** Back to the integration over $k_2^+$, we apply the restriction on small momenta we have just argued, which amounts to putting a lower bound at $RQ^2$. The evaluation of the integral then yields a finite value:

$$\int_{RQ^2}^{\min(k_0^+,k_1^+)} \frac{dk_2^+}{k_2^+} = \ln \frac{\min(k_0^+,k_1^+)}{RQ^2}\,. \tag{192}$$

Let us now take a closer look at the integration limits. As for the upper one, we argued that the momentum of the gluon cannot be larger than that of the quark or of the antiquark that has emitted it. Hence setting the upper cutoff to $\min(k_0^+,k_1^+)$ is a reasonable prescription, but only up to a numerical factor. It is also in order to recall that Eq. (186) was established in the approximation in which the gluon was much softer than the quark and the antiquark. Hence we should actually even cut off the integration at a momentum much less than those of the quark and of the antiquark: the upper bound in the integral (192) should read $\epsilon \min(k_0^+,k_1^+)$, with $\epsilon \ll 1$. As for the lower bound, it is also not exact, as it was obtained from an order-of-magnitude estimate of the gluon coherence time. In order to evaluate the error that may result of a numerical uncertainty on this bound, let us replace it by $\kappa RQ^2$, where $\kappa$ is a number of order unity. With the upper and lower bounds modified in this way, the value of the integral over the "+" momentum of the gluon becomes

$$\ln \frac{\epsilon \min(k_0^+,k_1^+)}{\kappa RQ^2} = \ln \frac{\min(k_0^+,k_1^+)}{RQ^2} + \ln\epsilon - \ln\kappa\,. \tag{193}$$

Choosing $\epsilon$ much smaller than 1 but much larger than the argument of the first logarithm in this equation, it is the latter that dominates. Hence the bounds in Eq. (192) yield the correct result at the so-called *leading-logarithmic accuracy*.

At the same accuracy, since numerical factors do not contribute to the leading logarithm and since we assumed that the constituents of the initial onium had momenta of the same order, we may replace $\min(k_0^+, k_1^+)$ by $k_0^+ + k_1^+$, the total "+" momentum of the onium. Let us denote it by $p_\bullet^+$. The integral over the "+" momentum of the gluon finally reads

$$\int \frac{dk_2^+}{k_2^+} \longrightarrow \int_{RQ^2}^{\min(k_0^+, k_1^+)} \frac{dk_2^+}{k_2^+} \simeq \ln \frac{p_\bullet^+}{RQ^2} \qquad \text{at leading-logarithmic accuracy.} \tag{194}$$

**Switching to rapidity variables.**  Let us introduce the variable $y \equiv \ln p_\bullet^+/k_2^+$, which is just the logarithm of the inverse of the momentum fraction $z \equiv k_2^+/p_\bullet^+$ of the initial particle carried by the emitted gluon. We may rewrite it as follows:

$$y = \ln \frac{1}{z} = \ln \frac{p_\bullet^+}{k_2^+} = \left( \frac{1}{2} \ln \frac{p_\bullet^+}{p_\bullet^-} - \frac{1}{2} \ln \frac{k_2^+}{k_2^-} \right) + \ln \frac{|k_2^\perp|}{|p_\bullet^\perp|} . \tag{195}$$

Since all transverse momenta are assumed of similar magnitude $Q$, the last term is typically of order unity, and thus negligible when $p_\bullet^+$ and $k^+$ are very large and different enough from each other. The variable $y$ boils then down to the terms in parenthesis, which are literally the difference between the rapidity of the fast emitting particle,

$$Y \equiv \frac{1}{2} \ln \frac{p_\bullet^+}{p_\bullet^-} , \tag{196}$$

see Eq. (36), and that of the slow emitted gluon. The latter is positive. Indeed,

$$\frac{1}{2} \ln \frac{k_2^+}{k_2^-} = \ln \frac{|k_2^\perp|}{\sqrt{2} k_2^-} \geq \ln \left( \sqrt{2} RQ \right) \geq 0 . \tag{197}$$

This implies that $y \leq Y$. Furthermore, since the lower bound on the rapidity of the gluon just found is very small compared to the rapidity of the fast particle, one can consider that $y$ describes the full interval $[0, Y]$.

All in all, we have argued that the logarithmic integral over $k_2^+$ in Eqs. (165), (185) and (186) etc just leads to a factor $Y$:

$$\int \frac{dk_2^+}{k_2^+} \longrightarrow \int_{RQ^2}^{p_\bullet^+} \frac{dk_2^+}{k_2^+} \simeq \int_0^Y dy = Y . \tag{198}$$

### 3.4  Discussion of the leading-order calculation

Our discussion has just shown that $S$ acquires a dependence upon the rapidity $Y$ of the initial hadron due to quantum corrections:

$$S_\bullet(x_0^\perp, x_1^\perp) \to S_{\bar{q}q}(Y, x_0^\perp, x_1^\perp) = S_\bullet(x_0^\perp, x_1^\perp) + \bar{\alpha} Y \int \frac{d^2 x_2^\perp}{2\pi} \frac{x_{01}^{\perp 2}}{x_{02}^{\perp 2} x_{21}^{\perp 2}} \left[ S_\bullet(x_0^\perp, x_2^\perp) S_\bullet(x_2^\perp, x_1^\perp) \right.$$
$$\left. - S_\bullet(x_0^\perp, x_1^\perp) \right] + \mathcal{O}(\alpha_s^2) . \tag{199}$$

Let us comment that in the large-$N$ limit, this equation has an appealing interpretation. When $Y$ is very small, the first term obviously dominates: an onium almost at rest (formally, $Y \simeq 0$) interacts with the external field as a bare quark-antiquark state. If one boosts the onium to the rapidity $Y$, the phase space for the emission of a gluon opens up, meaning that there is now a non-zero probability that the onium interacts through a $\bar{q}qg$ state instead of the bare $\bar{q}q$

state. The onium may not branch, and still interact as a single dipole: this may happen with a probability given by the renormalization constant $Z_{\bar{q}q}(Y)$ in Eq. (165), which in the large-$N$ limit, reads

$$Z_{\bar{q}q}(Y) \simeq 1 - \bar{\alpha} Y \int \frac{d^2 x_2^\perp}{2\pi} \frac{x_{10}^{\perp 2}}{x_{02}^{\perp 2} x_{21}^{\perp 2}} \, . \tag{200}$$

This is the interpretation of the first and third terms in Eq. (199). If instead the initial onium branches, which may happen with probability $1 - Z_{\bar{q}q}(Y)$, then it is "seen" from the external field essentially as a two-dipole state, whose sizes are distributed as Eq. (185) (with proper normalizations). This corresponds to the second term in Eq. (199).

   We note that the probability that the initial dipole branches is parametrically on the order of $\bar{\alpha} Y$. For $Y \sim 1/\bar{\alpha}$, which are rapidities that are routinely vastly exceeded in modern colliders, this probability becomes of order unity. Hence the probability for a number $k$ of emissions, whose order of magnitude is expected to be $(\bar{\alpha} Y)^k$, may become large. Therefore, we need to be able to resum all orders in perturbation theory. This will be the object of the next section 4.

   Let us conclude with a short summary of what we have achieved in this section. We have computed the forward elastic $S$-matrix element for the interaction of an onium with an external field, including the lowest-order quantum fluctuation of the former. We have argued that when the rapidity of the incoming onium is large, then the dominant contributions are given by Eq. (199).

   The lowest-order quantum correction is due to quark-antiquark-gluon realizations of the interaction state of the onium, where the gluon may be assumed soft, in the considered Regge kinematics. It turns out to be proportional to the strong coupling constant $\alpha_s$, the number of colors $N$ and the rapidity of the onium $Y$. We have found that this lowest-order fluctuation is actually "seen" by the external potential as a set of two independent color dipoles – this interpretation being valid in the large-$N$ limit; see the calculation of higher orders below.

   Along the way, we have presented a graphical method to evaluate color factors, that will shortly prove handy to address higher-order graphs.

   The external field is "hidden" in the bare dipole $S$-matrix element $S_\bullet$. We will need to model a nucleus with the help of such a field if we want to describe physical processes. We shall postpone this discussion to the end of the next section.

# 4   Higher orders and the Balitsky-Kovchegov equation

In this section, we explain how to compute quantum corrections of higher order in the coupling constant. We start by studying the probability amplitude to find two gluons at the time of the interaction, first in an initial quark, and then in an onium (Sec. 4.1). In the case of the latter, we compute the forward elastic $S$-matrix element at order $\alpha_s^2$, retaining the dominant terms at high energy (Sec. 4.2). We shall see that the result becomes very simple in the limit of a large number of colors, and that the next-to-leading order can be understood as a simple iteration of the leading order discussed in the previous section. We deduce the resummation of all orders in the coupling constant (Sec. 4.3). Instead of a closed expression, the result will be presented as an evolution equation with the rapidity of the onium, which amounts to the renowned Balitsky-Kovchegov (BK) equation.

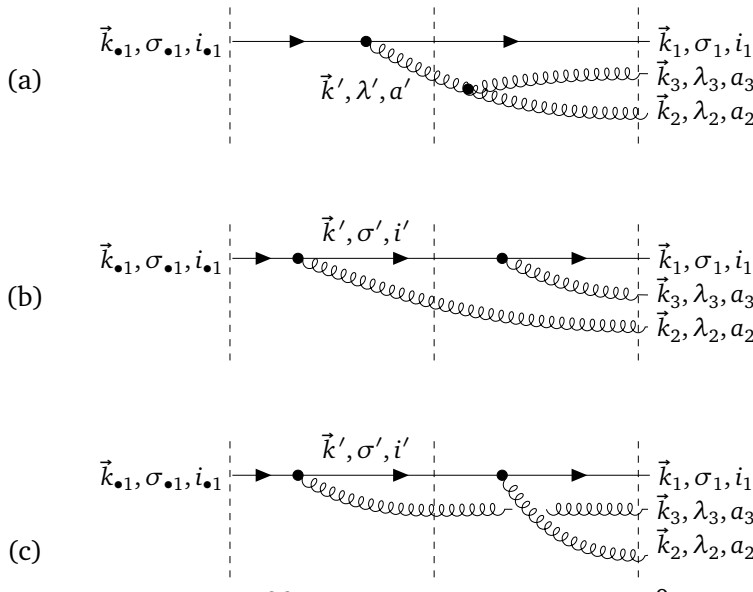

Figure 8: Diagrams contributing to the wave function $\psi_{qgg,\mathrm{i}(q)}$ at lowest order in the coupling. The dashed lines single out the intermediate states at which the energy denominators are evaluated.

## 4.1 Two-gluon Fock states at lowest perturbative order

### 4.1.1 Wave function of an initial quark at order $g^2$

The two-gluon causal real-emission diagrams contributing to the quark wave function at the lowest order $g^2$ are listed in Fig. 8. Applying Eq. (17), the corresponding wave functions read

$$
\begin{aligned}
\psi_{qgg,\mathrm{i}(q)}^{\mathrm{(a)}} &= \sum_{\lambda',a'} \int_{\vec{k}'} \frac{\langle \vec{k}_2,\lambda_2,a_2; \vec{k}_3,\lambda_3,a_3 | H_1 | \vec{k}',\lambda',a' \rangle \, \langle \vec{k}_1,\sigma_1,i_1; \vec{k}',\lambda',a' | H_1 | \vec{k}_{\bullet 1},\sigma_{\bullet 1},i_{\bullet 1} \rangle}{k_{\bullet 1}^- - k_1^- - k_2^- - k_3^-} \cdot \frac{1}{k_{\bullet 1}^- - k_1^- - k'^-}, \\
\psi_{qgg,\mathrm{i}(q)}^{\mathrm{(b)}} &= \sum_{\sigma',i'} \int_{\vec{k}'} \frac{\langle \vec{k}_1,\sigma_1,i_1; \vec{k}_3,\lambda_3,a_3 | H_1 | \vec{k}',\sigma',i' \rangle \, \langle \vec{k}',\sigma',i'; \vec{k}_2,\lambda_2,a_2 | H_1 | \vec{k}_{\bullet 1},\sigma_{\bullet 1},i_{\bullet 1} \rangle}{k_{\bullet 1}^- - k_1^- - k_2^- - k_3^-} \cdot \frac{1}{k_{\bullet 1}^- - k_1^- - k_2^-}, \\
\psi_{qgg,\mathrm{i}(q)}^{\mathrm{(c)}} &= \sum_{\sigma',i'} \int_{\vec{k}'} \frac{\langle \vec{k}_1,\sigma_1,i_1; \vec{k}_2,\lambda_2,a_2 | H_1 | \vec{k}',\sigma',i' \rangle \, \langle \vec{k}',\sigma',i'; \vec{k}_3,\lambda_3,a_3 | H_1 | \vec{k}_{\bullet 1},\sigma_{\bullet 1},i_{\bullet 1} \rangle}{k_{\bullet 1}^- - k_1^- - k_2^- - k_3^-} \cdot \frac{1}{k_{\bullet 1}^- - k_1^- - k_3^-}.
\end{aligned}
\tag{201}
$$

We are again restricting ourselves to soft gluons, that will dominate high-energy scattering. Moreover, we can show that the leading contribution will be given by configurations in which the "+" momenta of the particles are strictly ordered also among themselves. Let us argue this in some detail.

**Leading logarithms.** We choose to label the momenta in such a way that $k_2^+ \geq k_3^+$. From our experience of the leading-order calculation, as well as from general properties of unbroken gauge theories, we anticipate that the contribution of the emission of the two gluons to the $S$-matrix element will eventually result in a large factor stemming from nested logarithmic integrals over the "+" component of the gluon momenta. The latter will take the form

$$
\int_{RQ^2}^{k_1^+} \frac{dk_2^+}{k_2^+} \int_{RQ^2}^{k_2^+} \frac{dk_3^+}{k_3^+} = \frac{1}{2} \ln^2 \frac{k_1^+}{RQ^2} \simeq \frac{Y^2}{2} \,.
\tag{202}
$$

Restricting the integrals to strongly-ordered values of the momenta amounts to replacing the upper bounds by $\epsilon_{1,2}k_{1,2}^+$, for some $\epsilon_{1,2}$ such that $\epsilon_{1,2} \ll 1$ (but keeping $\epsilon_1 k_1^+ \geq RQ^2$):

$$
\begin{aligned}
\int_{RQ^2}^{\epsilon_1 k_1^+} \frac{dk_2^+}{k_2^+} \int_{RQ^2}^{\max(\epsilon_2 k_2^+, RQ^2)} \frac{dk_3^+}{k_3^+} &= \frac{1}{2}\ln^2 \frac{\epsilon_1 \epsilon_2 k_1^+}{RQ^2} \\
&= \frac{1}{2}\ln^2 \frac{k_1^+}{RQ^2} + \ln(\epsilon_1 \epsilon_2)\ln \frac{k_1^+}{RQ^2} + \frac{1}{2}\ln^2(\epsilon_1 \epsilon_2).
\end{aligned}
\tag{203}
$$

This results in additional terms, which can be made formally negligible compared to Eq. (202) by choosing $\epsilon_{1,2}$ not *too* small. Hence ordering strongly the momenta does not alter the coefficient of the leading power of the largest logarithm $\ln(k_1^+/RQ^2)$. In other words, we can indeed afford to assume that gluon 3 is soft with respect to both the quark and gluon 2: provided that the largest hierarchy of scales is between $k_1^+$ and $RQ^2$, we still get the "leading logarithms" right.

In Regge kinematics, the energy denominators are always dominated by the light cone energy of the particle that possesses the smallest "+" momentum. The matrix elements of the interaction Hamiltonian are taken in the eikonal approximation. We have already worked out the matrix element $\langle qg|H_1|q\rangle$, see Eq. (131). We now need the matrix element $\langle gg|H_1|g\rangle$ in the same approximation, namely in the limit in which one of the outgoing gluons very soft compared to the other one.

**Eikonal approximation for soft gluon emission off a gluon.** We start from the form of the $\langle gg|H_1|g\rangle$ matrix element given in Eq. (121), which reads in the variables of Eq. (201),

$$
\begin{aligned}
\left(\langle g;\vec{k}_2,\lambda_2,a_2| \otimes \langle g;\vec{k}_3,\lambda_3,a_3|\right) H_1 |g;\vec{k}',\lambda',a'\rangle = \\
-g\left(-if^{a_2 a_3 a'}\right) \Big\{ \left[(k_3 - k_2)\cdot\varepsilon_{(\lambda')}(k')\right]\left[\varepsilon_{(\lambda_3)}^*(k_3)\cdot\varepsilon_{(\lambda_2)}^*(k_2)\right] \\
+ \left[(k_2 + k')\cdot\varepsilon_{(\lambda_3)}^*(k_3)\right]\left[\varepsilon_{(\lambda_2)}^*(k_2)\cdot\varepsilon_{(\lambda')}(k')\right] \\
- \left[(k' + k_3)\cdot\varepsilon_{(\lambda_2)}^*(k_2)\right]\left[\varepsilon_{(\lambda')}(k')\cdot\varepsilon_{(\lambda_3)}^*(k_3)\right]\Big\}(2\pi)^3\delta^3(\vec{k}_2 + \vec{k}_3 - \vec{k}').
\end{aligned}
\tag{204}
$$

We will need to keep the largest term(s) in the limit $k_3^+ \ll k'^+ \simeq k_2^+$. Replacing the components of the polarization vectors by Eq. (114), we see that the scalar products of pairs of them reduce to products of their transverse parts, and hence are of order unity. Looking at the remaining factors,

$$
(k_2 + k')\cdot\varepsilon_{(\lambda_3)}^*(k_3) \simeq 2k_2^+ \frac{k_2^\perp \cdot \varepsilon_{(\lambda_3)}^{*\perp}(k_3)}{k_3^+}
\tag{205}
$$

obviously dominates in the Regge limit over the corresponding factors in the two other terms, parametrically by $k_2^+/k_3^+$. All in all, the $g \to gg$ matrix element has the very same form and features as that corresponding to the gluon emission off a quark, except for the color factor:

$$
\begin{aligned}
\left(\langle g;\vec{k}_2,\lambda_2,a_2| \otimes \langle g;\vec{k}_3,\lambda_3,a_3|\right) H_1 |g;\vec{k}',\lambda',a'\rangle \\
\simeq g\delta_{\lambda_2\lambda'}\left(-if^{a'a_2 a_3}\right) k_3^\perp \cdot \varepsilon_{(\lambda_3)}^{\perp*} \frac{2k_2^+}{k_3^+}(2\pi)^3\delta^2(k_2^\perp + k_3^\perp - k'^\perp)\delta(k_2^+ - k'^+).
\end{aligned}
\tag{206}
$$

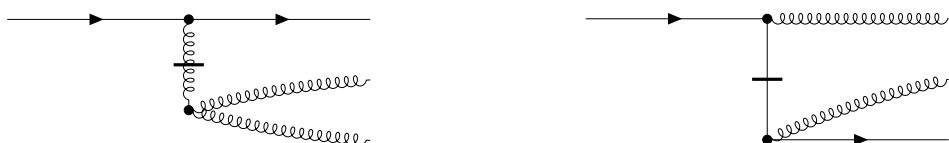

Figure 9: Instantaneous-exchange diagrams that do not contribute at leading-logarithmic accuracy.

**Wave functions.** We can immediately write down the different contributions to the wave functions. $\psi^{(a)}$ and $\psi^{(b)}$ have similar expressions:

$$
\begin{Bmatrix} \psi^{(a)}_{qgg,\mathfrak{i}(q)} \\[2mm] \psi^{(b)}_{qgg,\mathfrak{i}(q)} \end{Bmatrix} \simeq 4g^2 \delta_{\sigma_1 \sigma_{\bullet 1}} \times \begin{Bmatrix} if^{a'a_2a_3}(t^{a'})_{i_1 i_{\bullet 1}} \dfrac{(k_2^\perp + k_3^\perp)\cdot \varepsilon^{*\perp}_{(\lambda_2)}}{(k_2^\perp + k_3^\perp)^2} \\[4mm] (t^{a_3}t^{a_2})_{i_1 i_{\bullet 1}} \dfrac{k_2^\perp \cdot \varepsilon^{*\perp}_{(\lambda_2)}}{(k_2^\perp)^2} \end{Bmatrix} \times \dfrac{k_3^\perp \cdot \varepsilon^{*\perp}_{(\lambda_3)}}{(k_3^\perp)^2}
$$
$$
\times\, 2k_{\bullet 1}^+ (2\pi)^3 \delta^2\left(k_1^\perp + k_2^\perp + k_3^\perp - k_{\bullet 1}^\perp\right)\delta\left(k_1^+ - k_{\bullet 1}^+\right). \tag{207}
$$

As for $\psi^{(c)}$, it is obtained from the same calculation as for $\psi^{(b)}$, except for a crucial difference in the energy denominators. Overall, it is seen to be parametrically suppressed by $k_3^+/k_2^+ \ll 1$ with respect to the two other contributions. This fact can be understood physically through the lifetime arguments developed above (see Sec. 3.3.3). The gluon that has the lowest "+" momentum is also the shortest-lived parton. Therefore, it is dominantly emitted after the larger-momentum gluon, since we require that it survives until the time of the interaction with the potential. Thus $\psi^{(c)}$ is negligible compared to $\psi^{(a)}$ and $\psi^{(b)}$.

At this point, we may comment on the fact that we have excluded a priori instantaneous exchange diagrams, such as the ones in Fig. 9. In such graphs, the two gluons are produced at the same time, and thus, there cannot be a strong hierarchy in their "+" momenta. Hence such graphs do not contribute to leading-logarithmic accuracy.

Once again, it turns out useful to Fourier-transform to transverse positions. Associating the positions $x_i^\perp, x_{\bullet 1}^\perp$ to the momenta $k_i^\perp, k_{\bullet 1}^\perp$, we get

$$
\begin{Bmatrix} \widetilde{\psi}^{(a)}_{qgg,\mathfrak{i}(q)} \\[2mm] \widetilde{\psi}^{(b)}_{qgg,\mathfrak{i}(q)} \end{Bmatrix} \simeq \langle x_1^\perp, k_1^+ | x_\bullet^\perp, k_{\bullet 1}^+ \rangle\, \delta_{\sigma_1 \sigma_{\bullet 1}} \left(\dfrac{ig}{\pi}\right)^2 \dfrac{x_{21}^\perp \cdot \varepsilon^{*\perp}_{(\lambda_2)}}{(x_{21}^\perp)^2} \begin{Bmatrix} -\dfrac{x_{32}^\perp \cdot \varepsilon^{*\perp}_{(\lambda_3)}}{(x_{32}^\perp)^2} \\[4mm] \dfrac{x_{31}^\perp \cdot \varepsilon^{*\perp}_{(\lambda_3)}}{(x_{31}^\perp)^2} \end{Bmatrix} \begin{Bmatrix} (-if^{a'a_2a_3})(t^{a'})_{i_1 i_{\bullet 1}} \\[2mm] (t^{a_3}t^{a_2})_{i_1 i_{\bullet 1}} \end{Bmatrix}. \tag{208}
$$

The structure of these expressions has a clear and intuitive interpretation. The first two factors represent the probability amplitude to observe the quark at transverse position $x_1^\perp$, with "+" momentum $k_1^+$ and helicity $\sigma_1$. The next factors are the product of two emission factors defined in Eq. (140): $E_{21}\bar{E}_{32}$ for $\widetilde{\psi}^{(a)}$, and $E_{21}E_{31}$ for $\widetilde{\psi}^{(b)}$. They encode the emission of gluon 2 off the quark and, subsequently, that of gluon 3 off the gluon (respectively off the quark). Finally, there are color factors, that could easily be figured out independently, just from the nature of the emission vertices.

### 4.1.2 From quarks to onia

It is not difficult to generalize this discussion to an initial quark-antiquark color-neutral pair, to get the wave function $\widetilde{\psi}_{\bar{q}qgg,\mathfrak{i}(\phi)}$ at order $g^2$: it is enough to take account of the graphs in which one or the other gluon is emitted off the antiquark, see Fig. 10.

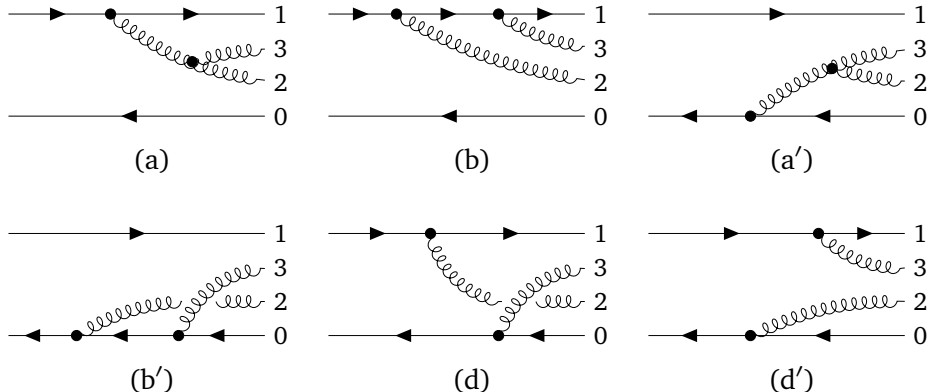

Figure 10: Graphs contributing to the wave function $\psi_{\bar{q}qgg,\mathbb{i}(q\bar{q})}$ at order $g^2$. (a) and (b) are the same as the ones in Fig. 8, supplemented by a spectator antiquark. In (a′) and (b′), the gluon(s) are emitted off the antiquark instead of the quark. In (d) and (d′), one gluon is radiated off the quark and one off the antiquark. The emission of the gluon labeled "3" which carries the smallest momentum is always posterior to the emission of the gluon "2".

Let us give a couple of examples. $\widetilde{\psi}^{(a)}_{\bar{q}qgg,\mathbb{i}(\phi)}$ is essentially $\widetilde{\psi}^{(a)}_{qgg,\mathbb{i}(q)}$ in Eq. (208), up to the wave function of the quark, which is replaced by the wave function of the onium:

$$\widetilde{\psi}^{(a)}_{\bar{q}qgg,\mathbb{i}(\phi)} = \frac{1}{2\sqrt{N}}\phi^{\sigma_0;\sigma_1}_\bullet(x_0^\perp,k_0^+;x_1^\perp,k_1^+)\left(\frac{ig}{\pi}\right)^2\frac{x_{21}^\perp \cdot \varepsilon^{*\perp}_{(\lambda_2)}}{\left(x_{21}^\perp\right)^2}\left(-\frac{x_{32}^\perp \cdot \varepsilon^{*\perp}_{(\lambda_3)}}{\left(x_{32}^\perp\right)^2}\right)(-if^{a'a_2a_3})(t^{a'})_{i_1i_0}. \quad (209)$$

The contribution to the wave function that corresponds to graph (d′), in which the quark and the antiquark emit one gluon each, reads

$$\widetilde{\psi}^{(d')}_{\bar{q}qgg,\mathbb{i}(\phi)} = \frac{1}{2\sqrt{N}}\phi^{\sigma_0;\sigma_1}_\bullet(x_0^\perp,k_0^+;x_1^\perp,k_1^+)\left(\frac{ig}{\pi}\right)^2\left(-\frac{x_{20}^\perp \cdot \varepsilon^{*\perp}_{(\lambda_2)}}{\left(x_{20}^\perp\right)^2}\right)\frac{x_{31}^\perp \cdot \varepsilon^{*\perp}_{(\lambda_3)}}{\left(x_{31}^\perp\right)^2}(t^{a_3}t^{a_2})_{i_1i_0}. \quad (210)$$

We are now in a position to discuss the contribution of such Fock states to the $S$-matrix element of an onium.

## 4.2 First subleading correction to the eikonal elastic $S$-matrix element for an onium

In order to get the real contributions of order $\alpha_s^2$ to the $S$-matrix element, we apply Eq. (111) using the wave functions whose graphs are displayed in Fig. 10, convoluted with the ad hoc Wilson lines. We shall then discuss the virtual corrections in order to get the complete expression for the $S$-matrix element at order $\alpha_s^2$ in perturbation theory.

### 4.2.1 Processing one particular real contribution

We shall first work out in detail one particular term, whose graph is displayed in Fig. 11. Applying Eq. (111),

$$S^{\bar{q}qgg(2)}_\phi\bigg|_{\text{Fig. 11}} = \sum_{\substack{i_0,i'_0,i_1,i'_1\\a_2,a'_2,a_3,a'_3}}\sum_{\substack{\sigma_0,\sigma_1\\\lambda_2,\lambda_3}}\int_{\substack{\{(x_i^\perp,k_i^+)\}_{i\in\{0\cdots3\}}\\k_3^+\le k_2^+}}\left(\widetilde{\psi}^{(d')i'_0i'_1a'_2a'_3}_{\bar{q}qgg,\mathbb{i}(\phi)}\right)^* \quad (211)$$

$$\times\left[W^{\bar{F}}(x_0^\perp)\right]_{i'_0i_0}\left[W^F(x_1^\perp)\right]_{i'_1i_1}\left[W^A(x_2^\perp)\right]_{a'_2a_2}\left[W^A(x_3^\perp)\right]_{a'_3a_3}\times\widetilde{\psi}^{(a)i_0i_1a_2a_3}_{\bar{q}qgg,\mathbb{i}(\phi)}.$$

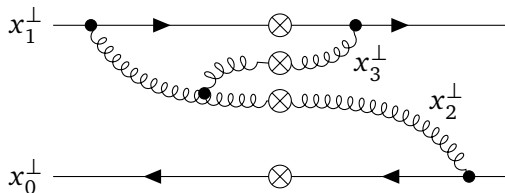

Figure 11: Diagram contributing to $S$ in the high-energy and large-number-of-color limits: the wave function of the initial state is $\widetilde{\psi}^{(a)}$, that of the final state $\widetilde{\psi}^{(d')}$ (see Fig. 10). The gluon at transverse position $x_2^\perp$ is the longest-lived one, and the gluon at $x_3^\perp$ is emitted off the first gluon at $x_2^\perp$ before the interaction with the external field and absorbed by the quark after the interaction.

The combinatorial factor of $1/2$, which would ordinarily be required due to the presence of two gluons in the interaction states, is effectively accounted for by our specific choice of momentum ordering. To improve readability, we have only kept track of the dependence of the wave functions upon the colors, which are the only quantum numbers that are modified by the eikonal interaction with the external field. Inserting Eqs. (209) and (210) into this equation, we get the convolution

$$S_\phi^{\bar{q}qgg(2)}\Big|_{\text{Fig. 11}} = \int_{\substack{(x_0^\perp,k_0^+) \\ (x_1^\perp,k_1^+)}} \frac{1}{4} \sum_{\sigma_0,\sigma_1} \left|\phi_\bullet^{\sigma_0;\sigma_1}(x_0^\perp,k_0^+;x_1^\perp,k_1^+)\right|^2 S_{\bar{q}q}^{\bar{q}qgg(2)}(x_0^\perp,k_0^+;x_1^\perp,k_1^+)\Big|_{\text{Fig. 11}}. \quad (212)$$

The wave function of the initial onium $\phi_\bullet$ is of course common to all contributions. The remaining term in the convolution is the $S$-matrix element for a dipole of fixed transverse positions

$$S_{\bar{q}q}^{\bar{q}qgg(2)}(x_0^\perp,k_0^+;x_1^\perp,k_1^+)\Big|_{\text{Fig. 11}} = \bar{\alpha}^2 \int \frac{dk_2^+}{k_2^+} \int^{k_2^+} \frac{dk_3^+}{k_3^+} \int \frac{d^2x_2^\perp}{2\pi} \frac{x_{21}^\perp \cdot x_{20}^\perp}{x_{21}^{\perp 2} x_{20}^{\perp 2}} \quad (213)$$

$$\times \int \frac{d^2x_3^\perp}{2\pi} \frac{x_{31}^\perp \cdot x_{32}^\perp}{x_{31}^{\perp 2} x_{32}^{\perp 2}} \times S_{\text{Fig. 11}}(x_0^\perp,x_1^\perp,x_2^\perp,x_3^\perp),$$

where the color factors and the matrix elements of the Wilson lines have been grouped in the function

$$S_{\text{Fig. 11}}(x_0^\perp,x_1^\perp,x_2^\perp,x_3^\perp) = \frac{4}{N^3} \left(-i f^{a_2 a_3 a'}\right) \text{Tr}\left[t^{a'} W(x_0^\perp)^\dagger t^{a_2'} t^{a_3} W(x_1^\perp)\right]$$

$$\times \widetilde{W}_{a_2' a_2}(x_2^\perp) \widetilde{W}_{a_3' a_3}(x_3^\perp). \quad (214)$$

We note that this expression, up to the numerical factor $4/N^3$ whose choice will be commented later on, is the one we would get by applying the pictorial rules listed in Sec. 3.2.3 (see also Sec. A.3) to the graph in Fig. 11.

The integration over the "+" components of the momenta can be performed, after having put an infrared cutoff in the way explained in Sec. 3.3.3. Assuming that there is no strong hierarchy between the momenta of the constituents of the initial onium, then the leading-logarithmic factor resulting from this integration is the same as that obtained in Eq. (202), namely $Y^2/2$, where $Y$ is now the rapidity of the onium.

We now address the factor $S_{\text{Fig. 11}}$ in Eq. (214).

**Evaluation of color factors.** Let us try and reduce the color factors associated to the diagram in Fig. 11. We are going to use the pictorial rules introduced above (Sec. 3.2.3) to address the case of the $\bar{q}qg$ Fock state.

We first transform all adjoint Wilson lines into fundamental ones, using twice the identity (174). The diagram in Fig. 11 is reduced as follows:

$$\text{(diagram)} \qquad (215)$$

We then trade the 3-gluon vertex for quark-gluon vertices using the algebra (172):

$$\text{(diagram)} \qquad (216)$$

We finally use the Fierz identity (178) for each of the terms, to eliminate adjoint lines (i.e. gluon propagators) in favor of pairs of fundamental/complex conjugate fundamental lines (i.e. superimposed propagators of quark/antiquark pairs). As for graph (I),

$$\text{(I)} = \frac{1}{4}\text{(diagram)} - \frac{1}{4N}\text{(diagram)} - \frac{1}{4N}\text{(diagram)} + \frac{1}{4N^2}\text{(diagram)}. \qquad (217)$$

As for graph (II), the Fierz identity also leads to a sum of four terms of which only one differs from the terms making up (I), the one in which both gluons are replaced by a quark-antiquark pair. Putting everything together, two terms are eventually left in the difference (I) − (II):

$$\text{(diagram)} = \text{(I)} - \text{(II)} = \frac{1}{4}\text{(diagram)} - \frac{1}{4}\text{(diagram)}. \qquad (218)$$

We just read off these diagrams a simplified expression of $S_{\text{Fig. 11}}$

$$
\begin{aligned}
S_{\text{Fig. 11}}(x_0^\perp, x_1^\perp, x_2^\perp, x_3^\perp) = {} & S_\bullet(x_0^\perp, x_2^\perp) S_\bullet(x_2^\perp, x_3^\perp) S_\bullet(x_3^\perp, x_1^\perp) \\
& - \frac{1}{N^2} S_{6\bullet}(x_0^\perp, x_3^\perp, x_2^\perp, x_1^\perp, x_3^\perp, x_2^\perp),
\end{aligned} \qquad (219)
$$

where we have defined the $S$-matrix for a bare sextupole as

$$
S_{6\bullet}(\chi_a^\perp, \chi_b^\perp, \chi_c^\perp, \chi_d^\perp, \chi_e^\perp, \chi_f^\perp) \equiv \frac{1}{N} \text{Tr}\left[ W(\chi_a^\perp)^\dagger W(\chi_b^\perp) W(\chi_c^\perp)^\dagger W(\chi_d^\perp) W(\chi_e^\perp)^\dagger W(\chi_f^\perp) \right]. \qquad (220)
$$

The normalization $1/N$ makes sure that $|S_{6\bullet}|$ describes the interval $[0, 1]$.

We see with this example that the next-to-leading order has a rather complicated color structure: sextupoles appear. This may hamper simple generalizations to higher orders of perturbation theory if the partonic system interacts through even more complicated representations of the color group. However, if we assume that the $S_\bullet$'s are of order 1 as far as the expansion in the number of colors is concerned, the first term in Eq. (219) dominates over the second one by $N^2$ when $N$ is large. This hints that going to the limit of large number of colors may simplify a lot the expressions: the leading term in the expansion $N \to +\infty$ effectively enables to reduce the color structure of all relevant diagrams to a tensor product of fundamental dipoles, as far as the scattering off the external field is concerned.

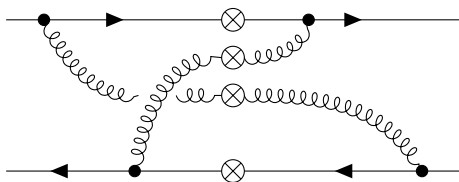

Figure 12: Diagram that does not contribute to $S$ in the limit $N \to +\infty$: it is related to its non-planar topology. Its expression is given in Eq. (223).

The final expression for the dominant contribution of Fig. 11 at large $N$ to the dipole $S$-matrix element can be written as

$$
S_{\bar{q}q}^{\bar{q}qgg(2)}(Y, x_0^\perp, x_1^\perp)\Big|_{\text{Fig. 11}} = \frac{(\bar{\alpha}Y)^2}{2} \int \frac{d^2 x_2^\perp}{2\pi} \left( -\frac{x_{21}^\perp \cdot x_{20}^\perp}{x_{21}^{\perp 2} x_{20}^{\perp 2}} \right) \int \frac{d^2 x_3^\perp}{2\pi} \left( -\frac{x_{31}^\perp \cdot x_{32}^\perp}{x_{31}^{\perp 2} x_{32}^{\perp 2}} \right)
$$
$$
\times S_\bullet(x_0^\perp, x_2^\perp) S_\bullet(x_2^\perp, x_3^\perp) S_\bullet(x_3^\perp, x_1^\perp). \tag{221}
$$

Let us comment on this formula. There is an overall $\alpha_s^2$, enhanced by two powers of $N$ and two powers of $Y$. The next factors are the ratios $-x_{21}^\perp \cdot x_{20}^\perp/(x_{21}^{\perp 2} x_{20}^{\perp 2})$ and $-x_{32}^\perp \cdot x_{31}^\perp/(x_{32}^{\perp 2} x_{31}^{\perp 2})$. They can be decomposed as follows with the help of the eikonal factors in Eq. (140):

$$
-\frac{x_{21}^\perp \cdot x_{20}^\perp}{x_{21}^{\perp 2} x_{20}^{\perp 2}} = \frac{\pi^2}{g^2} \sum_{\lambda_2} E_{21} \bar{E}_{20}^*, \qquad -\frac{x_{32}^\perp \cdot x_{31}^\perp}{x_{32}^{\perp 2} x_{31}^{\perp 2}} = \frac{\pi^2}{g^2} \sum_{\lambda_3} \bar{E}_{32} E_{31}^*, \tag{222}
$$

where the sum over the polarizations of the gluons is performed as in Eq. (146) and (164). The numerical factors $\pi^2/g^2$ are due to the fact that we had already factorized the numerical factors and couplings in Eq. (221). We have introduced "$-$" signs that eventually compensate, because in each of the products of $E$'s, one factor corresponds to the emission off the antiquark or off the "antiquark part of the gluon", and the other one off the quark.

The $S_\bullet$-factors in Eq. (221) account for the scattering of the interaction state, in the form of three independent color dipoles, which in the present case are defined by the three pairs of endpoint coordinates $(x_0^\perp, x_2^\perp)$, $(x_2^\perp, x_3^\perp)$ and $(x_3^\perp, x_1^\perp)$. Note that $S_{\text{Fig. 11}}$ reduces exactly to this product for $N \to +\infty$, which describes $[0, 1]$: this is the reason why we chose the peculiar normalization in the definition (214) of $S_{\text{Fig. 11}}$.

Finally, one integrates over the transverse phase space of each emitted gluon, with the normalization factor $1/(2\pi)$ for the measure of each integral.

### 4.2.2 Complete two-gluon Fock state contribution to $S$ at order $\alpha_s^2$

To compute the full set of graphs, we shall extrapolate simple rules deduced from our leading-order and next-to-leading order calculations (as for the latter, of the particular diagram of Fig. 11). Using these rules, we will evaluate all real contributions to the $S$-matrix element of interest, keeping only the leading-logarithms and the terms not suppressed when $N \to +\infty$.

We start by constructing all graphs from the complete set of wave functions listed in Fig. 10. Actually, only a subset of the 36 possible graphs have to be considered in the large-$N$ limit.

**Large-$N$ limit kills non-planar graphs.** The limit $N \to +\infty$, with $\alpha_s N$ kept fixed, is known to suppress the non-planar graphs [48]. Let us illustrate it on the example given in Fig. 12, which stems from the interference of the contributions (b$'$) and (d$'$) to the wave function of the onium shown in Fig. 10. The expression of the associated color factor reads

$$
S_{\text{Fig. 12}} = \frac{4}{N^3} \text{Tr} \left[ t^{a'} W(x_1^\perp) t^b t^a W(x_0^\perp)^\dagger t^{b'} \right] \widetilde{W}_{a'a}(x_3^\perp) \widetilde{W}_{b'b}(x_2^\perp), \tag{223}
$$

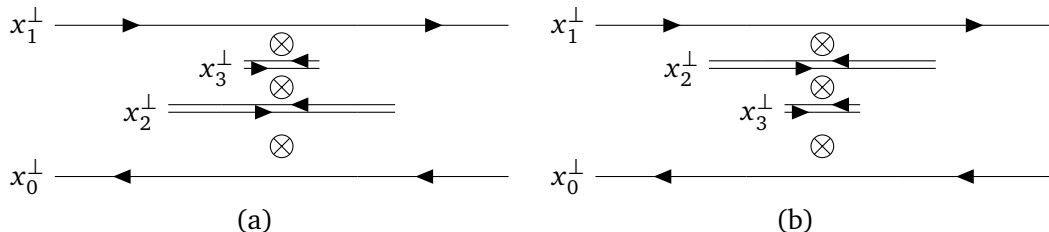

(a)  (b)

Figure 13: Representation of the large-$N$ limit of the sum of all graphs satisfying the condition that the gluon labeled 3 is the shortest-lived one (in Fig. 11 was displayed one of those). The interaction of the partonic system at time $\tau = 0$ is accounted for by the factor $S_\bullet(x_0^\perp, x_2^\perp)S_\bullet(x_2^\perp, x_3^\perp)S_\bullet(x_3^\perp, x_1^\perp)$ in graph (a), and $S_\bullet(x_0^\perp, x_3^\perp)S_\bullet(x_3^\perp, x_2^\perp)S_\bullet(x_2^\perp, x_1^\perp)$ in graph (b).

enforcing the same normalization as in Eq. (214). After using the identity (174) in its pictorial form (175) to transform Fig. 12, one easily sees that the latter becomes identical to (II) defined graphically in Eq. (216). It is thus suppressed by a factor $N^2$ with respect to the leading term of (I), and thus the contribution of the diagram of Fig. 12 to the $S$-matrix can safely be dropped in the limit $N \to +\infty$.

We then find by inspection that from the point of view of the color, all relevant planar graphs have topologies which fall into one of the following classes:

$$\begin{array}{l} x_1^\perp \dashrightarrow \\ x_3^\perp \dashrightarrow \\ x_2^\perp \dashrightarrow \\ \\ x_0^\perp \dashrightarrow \end{array} \qquad\qquad\qquad\qquad\qquad\qquad\qquad\qquad\qquad\qquad \text{(224)}$$

These graphs all reduce to products of three factors $S_\bullet$ in the large-$N$ limit:

$$\text{either}\quad S_{\bullet 02}S_{\bullet 23}S_{\bullet 31}\quad\text{(Fig. 13a)}\quad\text{or}\quad S_{\bullet 03}S_{\bullet 32}S_{\bullet 21}\quad\text{(Fig. 13b),} \tag{225}$$

with appropriate normalizations and the notation $S_{\bullet jk} \equiv S_\bullet(x_j^\perp, x_k^\perp)$, depending on which of the two dipoles $(x_0^\perp, x_2^\perp)$ or $(x_2^\perp, x_1^\perp)$ has emitted gluon 3. This is easy to see for the first graph: one just proceeds as in the leading-order case. The second one corresponds to that in Fig. 11 addressed in detail in the previous section. The third one is slightly more subtle: its large-$N$ limit leads to the sum

$$S_{\bullet 02}S_{\bullet 23}S_{\bullet 31} + S_{\bullet 03}S_{\bullet 32}S_{\bullet 21}\,, \tag{226}$$

as can be easily checked from the pictorial technique.

The full expressions of the graphs all have the same structure. First, there is an overall factor $(\bar{\alpha}Y)^2/2$. Second, there is a function of the difference of the transverse coordinates that may be found as follows: to each vertex, assign an appropriate eikonal factor (140), and sum over the polarization of the gluons. The overall sign is "$-$" if there is an odd number of couplings to antiquarks or to gluons "from above" (namely to their "antiquark part"). Third, there is a product of $S_\bullet$ as just discussed.

Adding the expressions of all graphs that survive when $N \to +\infty$, we get the following result for the contribution of $\bar{q}qgg$ Fock states to $S_{\bar{q}q}(x_0^\perp, x_1^\perp)$ at order $\alpha_s^2$:

$$\begin{aligned} S_{\bar{q}q}^{\bar{q}qgg(2)}(Y, x_0^\perp, x_1^\perp) = \frac{(\bar{\alpha}Y)^2}{2} \int \frac{d^2x_2^\perp}{2\pi} \frac{x_{01}^{\perp 2}}{x_{02}^{\perp 2}x_{21}^{\perp 2}} \Bigg( \int \frac{d^2x_3^\perp}{2\pi} \frac{x_{02}^{\perp 2}}{x_{03}^{\perp 2}x_{32}^{\perp 2}} S_{\bullet 03}S_{\bullet 32}S_{\bullet 21} \\ + \int \frac{d^2x_3^\perp}{2\pi} \frac{x_{21}^{\perp 2}}{x_{23}^{\perp 2}x_{31}^{\perp 2}} S_{\bullet 02}S_{\bullet 23}S_{\bullet 31} \Bigg). \end{aligned} \tag{227}$$

The interpretation of this a priori surprisingly simple formula is transparent. It is a convolution of three functions. First, the probability that the initial dipole $(x_0^\perp, x_1^\perp)$ branches into two dipoles $(x_0^\perp, x_2^\perp)$, $(x_2^\perp, x_1^\perp)$ by emitting a gluon whose rapidity lies in the interval $[0, Y]$. Second, the probability that subsequently, one or the other dipole also branches. Third, the $S$-matrix element for the independent scattering of the three dipoles effectively present in the state of the onium at the time of its interaction with the external potential; see Fig. 13 for an illustration.

We see that the probabilistic picture found in the leading-order calculation, in the leading logarithmic approximation and after the large-$N$ limit had been taken, still holds true in the next-to-leading order calculation: the interaction states of the onium are built through successive gluon emissions, interpreted as *independent* dipole branchings. We understand that higher orders will follow the same pattern.

In order to get a full expression for the $S$-matrix element up to order $\alpha_s^2$, one may think that one needs to consider also the $q\bar{q}q\bar{q}$ Fock states of the onium, since they can be generated by the conversion of the first-emitted gluon to a quark-antiquark pair. But this process turns out not to be singular when one of the produced particles, either the quark or the antiquark, is soft, and therefore, it does not contribute at leading-logarithmic accuracy.

Nevertheless, we need to include virtual corrections, namely, the contributions of the Fock states with no or one real gluon, which correspond to one or two dipoles respectively. This requires to calculate the constant $Z_{\bar{q}q}$ up to order $\alpha_s^2$, as well as one-loop corrections to the one-gluon emission diagrams. [Once we have calculated the latter, the former is obtained from Eq. (19)]. However, we may get these virtual terms at all orders in a much more straightforward way, building on the probabilistic picture we have just discovered in our fixed-order calculations of the real emissions.

### 4.3 All-order forward elastic $S$-matrix element and the Balitsky-Kovchegov equation

The calculation of the forward elastic $S$-matrix element up to order $\alpha_s^2$, in the high-energy and large-$N$ limits, has revealed the pattern that we have described just above.

At this point, it is clear that we will be able to compute contributions of still higher orders to the probability of a given interaction state of the onium by iteration of independent gluon emissions off dipoles, interpreted in turn as dipole branchings into pairs of dipoles. The dressed onium consists of the set of dipoles resulting from this branching process. A given state contributes to $S$ as a product of $S_\bullet$'s whose arguments are the sizes of the dipoles in the interaction state.

To these real gluon emissions are associated virtual corrections, corresponding to the probability of no branching. We shall first establish a recursion for this latter quantity, before going back to $S$.

#### 4.3.1 Recursion for $Z_{\bar{q}q}$

The constant $Z_{\bar{q}q}(Y)$ introduced in Sec. 3.2.2 [see Eq. (165) for its expression up to order $\alpha_s$] is interpreted as the probability that no gluon is radiated off a given dipole in a rapidity interval of size $Y$. Let us compute the change in $Z_{\bar{q}q}$ when one increases the interval by the infinitesimal quantity $dY$.

Since the emission of a gluon by a dipole, namely the decay of the latter, is a Poissonian process in rapidity (it has the same probability for all rapidities smaller than that of the emitting dipole), we can write the following factorization:

$$Z_{\bar{q}q}(Y + dY) = Z_{\bar{q}q}(Y) \times Z_{\bar{q}q}(dY). \tag{228}$$

At order $dY$, in the leading-logarithmic approximation, the quantity $Z_{\bar{q}q}(dY)$ is just given by the $\mathcal{O}(\alpha_s)$ expression (200). Therefore, expanding at first order in $dY$ and rearranging the terms, one gets

$$\frac{dZ_{\bar{q}q}(Y)}{dY} = -\bar{\alpha} \int \frac{d^2 x_2^\perp}{2\pi} \frac{x_{01}^{\perp 2}}{x_{02}^{\perp 2} x_{21}^{\perp 2}} \times Z_{\bar{q}q}(Y). \tag{229}$$

This differential equation is straightforward to solve. The initial condition is obvious: since there is no phase space for gluon emission when $Y = 0$, then $Z_{\bar{q}q}(Y = 0) = 1$. Thus:[14]

$$Z_{\bar{q}q}(Y) = \exp\left(-\bar{\alpha} Y \int \frac{d^2 x_2^\perp}{2\pi} \frac{x_{01}^{\perp 2}}{x_{02}^{\perp 2} x_{21}^{\perp 2}}\right). \tag{230}$$

This formula represents the resummation to all orders in $\alpha_s$ at leading-logarithmic accuracy and for $N \to +\infty$ of the virtual contributions to the probability that the initial dipole interacts as a bare dipole.

### 4.3.2 Recursion for $S$

We can now write a recursion for $S_{\bar{q}q}$ by tracking the first branching of the initial quark-antiquark pair in the evolution of the state of the latter.

A dipole $(x_0^\perp, x_1^\perp)$ possessing the rapidity $Y$ may decay into a pair of dipoles $(x_0^\perp, x_2^\perp)$, $(x_2^\perp, x_1^\perp)$ by emitting a gluon at position $x_2^\perp$. Let us denote by $y < Y$ the rapidity of this emission. The probability $\mathbb{P}(\text{decay} \in dy, d^2 x_2^\perp)$ that such an event occurs in a rapidity window of size $dy$ around $y$, the gluon being found at position $x_2^\perp$ up to $d^2 x_2^\perp$, is the product of two probabilities. First, the probability that there is no emission of any gluon with a rapidity larger than $y$, namely $Z_{\bar{q}q}(Y - y)$. Second, the probability that the dipole emits a gluon at position $x_2^\perp$ up to $d^2 x_2^\perp$ at any given rapidity, that can be deduced from Eq. (185). Putting these factors together,

$$\mathbb{P}\left(\text{decay} \in \{dy, d^2 x_2^\perp\}\right) = \bar{\alpha}\, dy\, Z_{\bar{q}q}(Y - y) \frac{d^2 x_2^\perp}{2\pi} \frac{x_{01}^{\perp 2}}{x_{02}^{\perp 2} x_{21}^{\perp 2}}. \tag{231}$$

The $S$-matrix element, conditioned on such a branching event, would be that of a Fock state of two dipoles of rapidites $y$ and sizes $(x_0^\perp, x_2^\perp)$, $(x_2^\perp, x_1^\perp)$, which is the product of the corresponding dipole $S$-matrix elements.

Alternatively, the initial dipole may not decay at all into dipoles that can be "seen" by the target: this would happen with probability $\mathbb{P}(\text{no decay}) = Z_{\bar{q}q}(Y)$. The $S$-matrix element would then coincide with the bare one of the initial dipole.

$S_{\bar{q}q}(Y, x_0^\perp, x_1^\perp)$ can thus be decomposed as the following weighted sum whose terms reflect the two possible events just described:

$$S_{\bar{q}q}(Y, x_0^\perp, x_1^\perp) = \int \mathbb{P}\left(\text{decay} \in \{dy, d^2 x_2^\perp\}\right) S_{\bar{q}q}(y, x_0^\perp, x_2^\perp) S_{\bar{q}q}(y, x_2^\perp, x_1^\perp)$$
$$+ \mathbb{P}(\text{no decay}) S_\bullet(x_0^\perp, x_1^\perp), \tag{232}$$

namely, after the appropriate replacements,

$$S_{\bar{q}q}(Y, x_0^\perp, x_1^\perp) = \bar{\alpha} \int_0^Y dy\, Z_{\bar{q}q}(Y - y) \int \frac{d^2 x_2^\perp}{2\pi} \frac{x_{01}^{\perp 2}}{x_{02}^{\perp 2} x_{21}^{\perp 2}} S_{\bar{q}q}(y, x_0^\perp, x_2^\perp) S_{\bar{q}q}(y, x_2^\perp, x_1^\perp)$$
$$+ Z_{\bar{q}q}(Y) S_\bullet(x_0^\perp, x_1^\perp). \tag{233}$$

---

[14]This expression needs of course regularization of the collinear divergences, for example in the form of cutoffs. We keep it implicit.

Taking the derivative with respect to the rapidity and rearranging the terms with the help of Eq. (229), we arrive at a nonlinear integro-differential equation:

$$\partial_Y S\left(Y, x_0^\perp, x_1^\perp\right) = \bar{\alpha} \int \frac{d^2 x_2^\perp}{2\pi} \frac{x_{01}^{\perp 2}}{x_{02}^{\perp 2} x_{21}^{\perp 2}} \left[ S(Y, x_0^\perp, x_2^\perp) S(Y, x_2^\perp, x_1^\perp) - S(Y, x_0^\perp, x_1^\perp) \right]. \quad (234)$$

We have dropped the "$\bar{q}q$" subscripts as there is no ambiguity. This is the famous Balitsky-Kovchegov equation [13, 16].

Solving this equation requires an initial condition $S(Y = 0, x_0^\perp, x_1^\perp)$, namely an expression for $S_\bullet$. Given the latter, let us point out the main property of the BK equation:

$$S_\bullet(x_0^\perp, x_1^\perp) \in [0, 1] \implies S(Y, x_0^\perp, x_1^\perp) \in [0, 1], \quad (235)$$

for all values of the parameters, as expected from general considerations (see Sec. 3.3.1). In particular, the scattering amplitude $T \equiv 1 - S$ never exceeds $T = 1$. This differs radically from equations written earlier to describe high-energy scattering in QCD, such as the Balitsky-Fadin-Kuraev-Lipatov (BFKL) equation [49, 50], which did not respect this unitarity bound, and which therefore had a more limited validity. We will come back to this discussion in Sec. 6.

### 4.3.3 Initial condition and model for the target nucleus

We have assumed that the target off which the onium scatters eikonally effectively consists of a classical Yang-Mills potential $\mathcal{A}$. This makes sense as a model for the field of a large nucleus, made of many nucleons, supposed independent. The large number of them enforces a classical limit. However, we have a priori no information on this field, and it surely fluctuates from event-to-event. Therefore, it would be in order to promote it to a stochastic field, and average the physical quantities over the realizations of the latter. The $S$-matrix element, the one related to cross sections, would be $S(Y, x_0^\perp, x_1^\perp)$ averaged over $\mathcal{A}$. We shall denote it by $\langle S(Y, x_0^\perp, x_1^\perp) \rangle_{\mathcal{A}}$. The BK equation (234) would also need to be averaged over the realizations of $\mathcal{A}$, leading to the following equation:

$$\partial_Y \left\langle S\left(Y, x_0^\perp, x_1^\perp\right)\right\rangle_{\mathcal{A}} = \bar{\alpha} \int \frac{d^2 x_2^\perp}{2\pi} \frac{x_{01}^{\perp 2}}{x_{02}^{\perp 2} x_{21}^{\perp 2}} \left[ \left\langle S(Y, x_0^\perp, x_2^\perp) S(Y, x_2^\perp, x_1^\perp)\right\rangle_{\mathcal{A}} - \left\langle S(Y, x_0^\perp, x_1^\perp)\right\rangle_{\mathcal{A}} \right]. \quad (236)$$

The problem is now that this equation is no longer closed: it calls for an evolution equation for the correlator $\langle SS \rangle_{\mathcal{A}}$, that, in turn, would call for an evolution equation for still higher-order correlators, and so on. We would end up with an infinite hierarchy of coupled integro-differential equations, which would correspond to the large-number-of-color limit of the Balitsky equations [13].

As a side note, it is worth highlighting that alternative approaches describe the rapidity evolution of dipole $S$-matrix elements (and more generally, of any correlator of Wilson lines at a finite number of colors) through closed-form equations known as the Jalilian-Marian-Iancu-McLerran-Weigert-Leonidov-Kovner (JIMWLK) equations [14, 15, 17, 18]. The latter, however, are formulated either as functional integro-differential equations or, equivalently, as stochastic differential equations for Wilson lines. The eventual connection to simple classical stochastic processes, which forms the central theme of this review, remains to be explored, if such a link can be established at all.

**Getting back a closed equation.** We may assume a "mean-field" approximation, factorizing this correlator as

$$\left\langle S(Y, x_0^\perp, x_2^\perp) S(Y, x_2^\perp, x_1^\perp)\right\rangle_{\mathcal{A}} = \left\langle S(Y, x_0^\perp, x_2^\perp)\right\rangle_{\mathcal{A}} \left\langle S(Y, x_2^\perp, x_1^\perp)\right\rangle_{\mathcal{A}}, \quad (237)$$

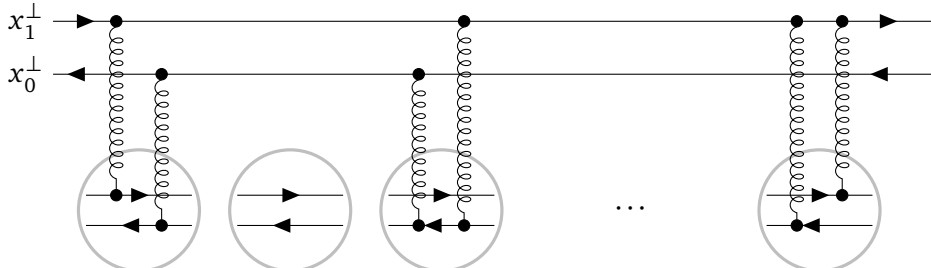

Figure 14: Graph contributing to $S_\bullet(x_0^\perp, x_1^\perp)$ in the Glauber-Mueller model. The circles in the lower part of the graph represent the nucleons, made of single independent color dipoles in this model. The latter interact with the projectile dipole by exchanging either two gluons or no gluon at all.

and get back to Eq. (234). What this factorization means physically is that the only source of correlations is the fact that the dipoles present in the Fock state at the time of the interaction with the target stem from the branching of a single initial dipole: their scatterings are totally uncorrelated. This makes sense if we have a nucleus made of a formally infinite number of nucleons, and if the latter are assumed independent of each other. Then, it is improbable that two different dipoles in the Fock state of the onium interact with the same nucleon.

The factorization (237) would actually be realized if the potential $\mathcal{A}$ were a solution of the Yang-Mills equations sourced by Gaussian-correlated color charges and after the large-$N$ limit has been taken (see Ref. [51], Appendix A for an explicit calculation). The former is the main assumption of the McLerran-Venugopalan (MV) model [52, 53] which leads to the following formula for the dipole forward elastic $S$-matrix element without quantum fluctuations:

$$S_\bullet(x_0^\perp, x_1^\perp) = \exp\left(-\frac{x_{01}^{\perp 2}Q_A^2}{4}\ln\frac{1}{\left|x_{01}^\perp\right|\Lambda_{\text{QCD}}}\right). \tag{238}$$

The constant $\Lambda_{\text{QCD}} \simeq 200$ MeV is the characteristic hadronic mass/momentum scale ("Landau pole"), and $Q_A$ is the so-called nuclear saturation scale. We have assumed translation and rotation invariance in the transverse plane, and the formula is valid for $|x_{01}^\perp| < 1/\Lambda_{\text{QCD}}$.

Let us comment on the form of $S_\bullet$ obtained in the MV model. We see that it is real, bounded by 0 and 1, and for $|x_{01}^\perp| \ll 2/Q_A$, it tends to 1: the amplitude $T_\bullet \equiv 1 - S_\bullet$ vanishes as

$$T_\bullet(x_0^\perp, x_1^\perp) \sim \frac{x_{01}^{\perp 2}Q_A^2}{4}\ln\frac{1}{\left|x_{01}^\perp\right|\Lambda_{\text{QCD}}}, \tag{239}$$

namely essentially quadratically up to the slowly-varying logarithm, as anticipated in Eq. (188). These features are in full agreement with the general properties of scattering amplitudes at high energy stated in Sec. 3.3.1.

Note also that when $|x_{01}^\perp| \gg 2/Q_A$, $S_\bullet$ goes to 0, meaning that the incoming dipole is fully absorbed ("black disk limit"). We note that the fall-off of $S_\bullet$ is Gaussian in $|x_{01}^\perp|$, which is actually a very fast decrease.

One could take this idea a step further and perform a calculation in a fully quantum setup. We have assumed that the target consists in a random classical field. But one may replace this external field by a large number of quantum particles. Assume that each nucleon is a single bare color dipole of size $|x_N^\perp| \sim 1/\Lambda_{QCD}$, and compute the forward elastic scattering amplitude of a quark-antiquark dipole state off this collection of target dipoles, supposing furthermore that each of the latter may exchange at most one pair of gluons (see Fig. 14). The exponential in Eq. (238) directly follows from the fact that the eikonal exchange of the pairs of gluons is

a Poisson point process, as a consequence of their eikonal coupling. A full discussion would be outside of the scope of this review: we shall refer the interested reader to the papers in Ref. [54, 55].

# 5 Branching random walks and the branching Brownian motion

We have just seen that the Fock state of a model hadron at the time of its interaction with an external field, in the approximations that underly the color dipole model, may be thought of as resulting of a binary branching process, and the scattering amplitude as a classical observable on this process. For observables such as the forward elastic scattering amplitude of a small dipole, it turns out that this branching process effectively belongs to the universality class of the one-dimensional branching Brownian motion (BBM) [23]. The latter is a simple stochastic process which evolves sets of particles completely characterized by their positions on an axis.

By belonging to the same universality class, we mean that observables on the color dipole model and on the BBM share asymptotic properties: analytical results obtained for the latter simpler model are easy to take over to QCD. That's the reason why the study of general classical branching processes is helpful to understand hadron scattering at high energies.

The main observable on the BBM is the probability that the particle with the largest position goes beyond some fixed threshold. The time evolution of this probability is given by the Fisher-Kolmogorov-Petrovsky-Piscounov (FKPP) equation [56, 57]. As we will see, the equivalent equation in QCD is precisely the BK equation.

In this self-contained section, we start by introducing a simple branching random walk (BRW), which is a particular lattice discretization of the branching Brownian motion. We define a class of observables on the BRW and on its continuous limit, the BBM, and show that they obey the FKPP equation (Sec. 5.1). We then discuss the solutions to the FKPP equation, or to a discretized version of it (Sec. 5.2). Applications to QCD will follow, in the next section 6.

Our aim here is to provide basic background on stochastic processes. The reader may learn more on stochastic processes in a textbook such as the one in Ref. [58], and on the BBM and on its relation with the FKPP equation e.g. in the review of Ref. [25].

## 5.1 Branching Brownian motion and the FKPP equation

### 5.1.1 A branching random walk and its continuous limit

We consider a set of particles whose state is completely characterized, at any given time $t$, by the positions of its constituents on a one-dimensional lattice with mesh size $\Delta x$. The system evolves in time, which we shall discretize in steps of size $\Delta t$, according to the following elementary processes. Between $t$ and $t + \Delta t$, independently of each other, a particle on site labeled $k$ may jump to the next site left or right (namely to the sites labeled $k - 1$ and $k + 1$ respectively), duplicate on the same site, or do nothing. The probabilities of these elementary processes are chosen to be constants, independent of the position $k$ on the lattice and of the time $t$:

$$\mathbb{P}(\overset{\frown}{\lfloor \bullet \rfloor}) = \mathbb{P}(\lfloor \bullet \rfloor^{\frown}) = \mu \,, \qquad \mathbb{P}(\bullet \to \overset{\bullet}{\bullet}) = r \,, \qquad \mathbb{P}(\bullet \to \bullet) = 1 - 2\mu - r \,. \tag{240}$$

The constants $\mu$ and $r$ are real numbers in the interval $[0, 1]$ and subject to the constraint $2\mu + r \leq 1$, in order to ensure that the three elementary probabilities are non-negative and well-defined. This stochastic process is, by definition, Markovian. We shall always assume that at the initial time $t = 0$, there is one single particle in the system, located on the site labeled 0.

We shall first investigate successively the cases $r = 0$, $\mu > 0$ (pure random walk) and $\mu = 0$, $r > 0$ (pure branching), before addressing the full model.

**Random walk.** If $r = 0$, the process defined above amounts to a discretization of the Brownian motion. In order to see this, let us compute the probability $p(x, t)$ for the particle to be on the site labeled by $x = k\Delta x$ after an evolution time $t = n\Delta t$ ($k$ and $n$ are integers). We may choose $k \geq 0$ without loss of generality. The probability $p(x, t)$ coincides with the probability that the initial particle jumps a number $j = n_l + n_r$ of times such that the number $n_r$ of right jumps be larger than the number $n_l$ of left jumps by exactly $k$.

The probability of a given pair of numbers $(n_l, n_r)$ of left and right jumps in $n \geq n_l + n_r$ steps total is the multinomial distribution

$$\mathbb{P}(n_l, n_r) = \binom{n}{n_l, n_r} \mu^{n_l + n_r} (1 - 2\mu)^{n - n_l - n_r} . \tag{241}$$

The probability $p$ then reads

$$p(x = k\Delta x, t = n\Delta t) = \sum_{n_l=0}^{n-k} \mathbb{P}(n_l, k + n_l) = \sum_{j=k}^{n} \left[ \binom{n}{j} (2\mu)^j (1 - 2\mu)^{n-j} \right] \left[ \binom{j}{\frac{j+k}{2}} \left( \frac{1}{2} \right)^j \right]. \tag{242}$$

In the second expression, we have rearranged the sum in such a way that the term $j$ has the interpretation of the probability that there are $j \geq k$ jumps exactly in $n$ time steps, independently of their direction, multiplied by the probability that $n_r - n_l = k$ given that $n_r + n_l = j$. From simple combinatorics, we have obtained this exact expression for $p(x, t)$.

It is possible to take the continuous limit, $\Delta t \to 0$ and $\Delta x \to 0$ with $x$ and $t$ fixed. It is enough to notice that the numbers appearing in the binomial coefficients become typically large, which justifies the use of the Stirling approximation. But overall, this limit is quite subtle to work out.

There is a simpler way to arrive at an expression for $p(x, t)$ in the continuous limit: establishing an evolution equation for $p$ with time, letting mesh sizes of the lattice that discretizes space and time vanish, and solving the obtained partial differential equation using known methods. We use the Markov property to write $p(x, t + \Delta t)$ as

$$p(x, t + \Delta t) = \sum_{k=-\infty}^{+\infty} \mathbb{P}(k\Delta x, \Delta t | 0, 0) \times \mathbb{P}(x, t + \Delta t | k\Delta x, \Delta t), \tag{243}$$

where $\mathbb{P}(x_1, t_1 | x_0, t_0)$ denotes the conditional probability that the particle is at $x_1$ at time $t_1$ given that it was at $x_0$ at time $t_0 < t$. We then recognize that the first factor in each term is the probability of one of the elementary processes listed in Eq. (240) starting with a particle on site labeled 0, and hence that only terms $k \in \{-1, 0, 1\}$ may contribute to the sum. The second factor can be rewritten as $\mathbb{P}(x - k\Delta x, t | 0, 0)$ thanks to translation invariance in space and time of the elementary processes, and thus identifies to $p(x - k\Delta x, t)$. Hence the equation reads, graphically,

$$p(x, t + \Delta t) = \mathbb{P}(\text{⌐•⌐}) \times p(x + \Delta x, t) + \mathbb{P}(\text{•} \to \text{•}) \times p(x, t) + \mathbb{P}(\text{⌐•⌐}) \times p(x - \Delta x, t), \tag{244}$$

namely after rearrangement of the terms,

$$p(x, t + \Delta t) - p(x, t) = \mu[p(x + \Delta x, t) + p(x - \Delta x, t) - 2p(x, t)]. \tag{245}$$

The initial condition corresponding to a single particle at position 0 is the indicator function

$$p(x, 0) = \mathbb{1}_{\{x=0\}} . \tag{246}$$

Let us now take the continuous limit $\Delta t, \Delta x \to 0$ of the evolution equation (245). The left-hand side obviously tends to the time derivative $p(x, t + \Delta t) - p(x, t) \to \Delta t \, \partial_t p(x, t)$. One

then recognizes that the right-hand side is a discretized Laplacian: in the limit $\Delta x \to 0$, it tends to the second-order differential operator $\mu \Delta x^2 \partial_x^2 p(x,t)$. Requiring that all terms of the evolution equation be relevant imposes that $\Delta t$ and $\mu \Delta x^2$ must be kept on the same order. Introducing the diffusion constant as the ratio of the latter to the former, $D = \mu \Delta x^2 / \Delta t$, we eventually arrive at the following partial differential equation:

$$\partial_t p(x,t) = D\partial_x^2 p(x,t). \tag{247}$$

The probability $p$ is now a distribution. At the initial time, it coincides with the Dirac distribution

$$p(x,0) = \delta(x). \tag{248}$$

Equation (247) is the Fokker-Planck (or forward Kolmogorov) equation for the one-dimensional Brownian motion. Its solution with the above initial condition and free boundary condition is obviously the Gaussian density

$$p(x,t) = \frac{1}{\sqrt{4\pi Dt}} e^{-x^2/(4Dt)}. \tag{249}$$

**Branching process.**    In order to get a pure branching process starting from the initial BRW, we may equivalently either set $\mu = 0$, or disregard the spatial distribution of the particles, namely consider the total number of particles at a given time $t$ independently of their positions. A recursion for the probabilities $q_n(t)$ that there are $n$ particles at time $t$ is readily established through simple combinatorics:

$$q_n(t+\Delta t) = \sum_{k=0}^{[n/2]} \binom{n-k}{k} r^k (1-r)^{n-2k} q_{n-k}(t), \qquad \text{with} \qquad q_n(0) = \delta_{n,1}, \tag{250}$$

where $[n/2] = n/2$ for even $n$ and $(n-1)/2$ for odd $n$. This equation is cumbersome to solve. But it simplifies drastically in the limit $\Delta t \to 0$.

In this limit, the branching probability must scale like $\Delta t$ in order to keep particle numbers finite at all times. Let us introduce the rate $\lambda \equiv r/\Delta t$, and take it constant. Then, only two terms survive in Eq. (250), the ones labeled by $k = 0$, and $k = 1$ for $n - k > 0$. We get the following master equation:

$$\frac{dq_n(t)}{dt} = \lambda[(n-1)q_{n-1}(t) - nq_n(t)], \qquad q_n(0) = \delta_{n,1}. \tag{251}$$

This (system of) equations is solved e.g. by introducing a generating function

$$G(z,t) = \sum_n z^n q_n(t), \qquad \text{which obeys} \qquad \begin{cases} \partial_t G(z,t) = \lambda z(z-1)\partial_z G(z,t), \\ G(z,0) = z. \end{cases} \tag{252}$$

The solution to this first-order partial differential equation can be found using the method of characteristics [59]. The obtained analytic function is then expanded in power series, and the coefficient of the term $z^n$ is identified to $q_n(t)$. Instead of working this out in detail, we simply check that

$$q_n(t) = e^{-\lambda t} \left(1 - e^{-\lambda t}\right)^{n-1}, \qquad \text{for} \qquad n \geq 1, \tag{253}$$

solves Eq. (251). The bottom line is that the number $n$ of particles is exponentially-distributed. For large $t$, its expectation value tends to $e^{\lambda t}$.

It bears mentioning that such "pure branching" models, also referred to as zero-space-dimensional models (potentially enhanced by a nonlinear recombination process), have been explored in the QCD literature, notably in Refs. [60, 61].

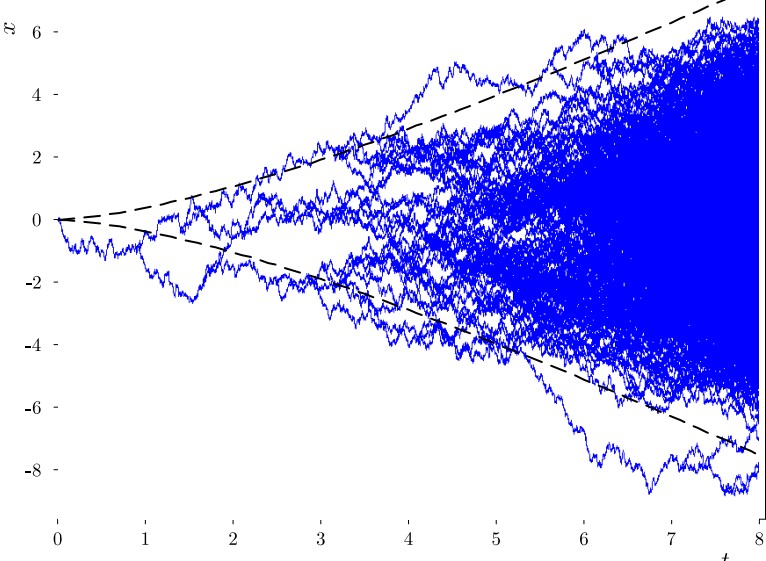

Figure 15: One typical realization of the branching Brownian motion with $D = \frac{1}{2}$ and $\lambda = 1$, until time $t = 8$. The dashed lines show the exact expectation values $m_t$ of the positions of the extreme particles as a function of time. An analytic asymptotic expression is given in Eq. (294) below.

**Branching Brownian motion.** The continuous limit $\Delta t, \Delta x \to 0$, with $\mu \Delta x^2 / \Delta t \equiv D$ and $r/\Delta t \equiv \lambda$ kept finite, of the full BRW model defined in Eq. (240) is the binary branching Brownian motion. This process rules the time evolution of a system of particles on the real line which undergo independent Brownian motions with diffusion constant $D$, and are replaced, independently of each other, by two particles at the same point, after a time distributed as $\lambda e^{-\lambda t}$. If one follows any given particle, picking randomly one offspring at each of its branchings, the observed trajectory is a realization of the Brownian motion. The total number of particles $n$ at a given time $t$ is distributed according to $q_n(t)$ in Eq. (253). One realization of the exact BBM is displayed in Fig. 15.

The positions of the particles at a given time are correlated, solely due to the fact that they have common ancestors. Consequently, the statistical properties of the set of particles generated by the BBM are non-trivial, but the source of the correlations is simple. The positions of the particles would be described by a time-dependent point measure. There is a priori no simple equation for the latter, as for the distribution of the position of the particle in the simple Brownian motion.

In order to determine observables on a BRW or on the BBM, one needs to solve finite-difference or partial differential equations respectively. In general, the latter turn out to be non-linear, and there is often no systematic method to find solutions.

We shall review below the formulation of a class of observables on the BBM, derive the equations they solve, and discuss the known properties of their solutions.

### 5.1.2 A partial differential equation for a class of observables on the BBM: The FKPP equation

Let us introduce the random number $N_t$ of particles in the BRW or BBM at time $t$, and denote their random positions by $X_1(t), \cdots, X_{N_t}(t)$. For any bounded function $z \mapsto f(z)$, we define

$$Q^{(f)}(t,x) = \mathbb{E}_0\left(\prod_{k=1}^{N_t} f(x - X_k(t))\right), \tag{254}$$

where $\mathbb{E}_0$ stands for the expectation value over realizations of the BRW or BBM. The index "0" means that they start with one single particle at position 0. Note that if the BBM started with a single particle at a generic position $x_0$ instead of 0, then the expectation value on the right-hand side could still be expressed in terms of $Q^{(f)}$, thanks to translation invariance of the BBM:

$$\mathbb{E}_{x_0}\left(\prod_{k=1}^{N_t} f(x - X_k(t))\right) = Q^{(f)}(t, x - x_0). \tag{255}$$

For the same reason, we may equivalently define $Q^{(f)}$ as

$$Q^{(f)}(t,x) = \mathbb{E}_x\left(\prod_{k=1}^{N_t} f(-X_k(t))\right). \tag{256}$$

In order to get a closed expression for $Q^{(f)}$, we again establish an evolution equation starting with the discrete model, as in the case of the simple random walk. We express $Q^{(f)}(t + \Delta t, x)$ with the help of $Q^{(f)}$ evaluated for an evolution time interval $t$, by distinguishing what happens to the initial particle during the very first time step. It may move left or right (in which case the subsequent evolution is a BRW starting at position $\mp\Delta x$ over the time interval $t$), do nothing, or branch to two particles on the initial site. In equation,

$$Q^{(f)}(t+\Delta t, x) = \mathbb{P}(\llcorner\hat{\bullet}\lrcorner) \times \mathbb{E}_{-\Delta x}\left(\prod_{k=1}^{N_t} f(x - X_k(t))\right) + \mathbb{P}(\llcorner\hat{\bullet}\lrcorner) \times \mathbb{E}_{\Delta x}\left(\prod_{k=1}^{N_t} f(x - X_k(t))\right)$$

$$+ \mathbb{P}(\boxed{\bullet} \to \boxed{\bullet}) \times \mathbb{E}_0\left(\prod_{k=1}^{N_t} f(x - X_k(t))\right) \tag{257}$$

$$+ \mathbb{P}(\boxed{\bullet} \to \boxed{\overset{\bullet}{\bullet}}) \times \mathbb{E}_0\left(\prod_{k_1=1}^{N_t^{(1)}} f\left(x - X_{k_1}^{(1)}(t)\right) \prod_{k_2=1}^{N_t^{(2)}} f\left(x - X_{k_2}^{(2)}(t)\right)\right).$$

The expectation values in the first three terms in the right-hand side identify respectively to $Q^{(f)}(t, x + \Delta x)$, $Q^{(f)}(t, x)$, $Q^{(f)}(t, x - \Delta x)$, see Eq. (255). These terms are of the same form as the ones we had in Eq. (244) for the random walk. The last term is new: it accounts for the possibility that the initial particle is replaced by two particles on site 0 in the first time step. The latter further evolve into two sets of particles: one made up of $N_t^{(1)}$ particles at positions $\{X_{k_1}^{(1)}(t)\}$ at time $t + \Delta t$, namely after $t$ additional time steps, and another one made up of $N_t^{(2)}$ particles at positions $\{X_{k_2}^{(2)}(t)\}$ at the final time. Now we observe that these sets are statistically independent, by definition of the BRW. Thus the expectation value factorizes as the product of expectation values over the two realizations of the BRW seeded respectively by the two particles at position 0 at time $\Delta t$. It thus identifies to $[Q^{(f)}(t,x)]^2$.

After the appropriate replacements, we find

$$Q^{(f)}(t + \Delta t, x) - Q^{(f)}(t, x) = \mu\left[Q^{(f)}(t, x + \Delta x) + Q^{(f)}(t, x - \Delta x) - 2Q^{(f)}(t, x)\right]$$
$$+ r\left\{[Q^{(f)}(t, x)]^2 - Q^{(f)}(t, x)\right\}, \tag{258}$$

endowed with the initial condition

$$Q^{(f)}(0, x) = f(x). \tag{259}$$

The continuous limit $\Delta t, \Delta x \to 0$ of this evolution equation is taken in the same way as in the case of the simple random walk. We need to require that the branching rate $\lambda \equiv r/\Delta t$ stay finite. The equation we obtain reads

$$\partial_t Q^{(f)}(t, x) = D\partial_x^2 Q^{(f)}(t, x) + \lambda \left\{ \left[ Q^{(f)}(t, x) \right]^2 - Q^{(f)}(t, x) \right\}, \tag{260}$$

with the initial condition (259).

This non-linear partial differential equation is the celebrated Fisher-Kolmogorov-Petrosky-Piscounov (FKPP) equation [56, 57]. It was introduced in the context of population genetics, but has turned out to have many potential applications across various scientific contexts. The connection between a stochastic process, the BBM, and a partial differential equation, the FKPP equation, was first established by McKean [62], and has become a fruitful point of contact between analysis and the theory of probabilities.

Before addressing the solutions to the FKPP equation (266), let us discuss a few relevant choices for the initial condition $f$, and the associated observables. In several contexts, the probability of the positions of the lead particles in BRWs or their probability density in the BBM, as well as the particle distribution close to the left or right lead, are of great interest. It is the case, for example, when the BBM is used to model the configurations of polymers in a random medium: the position variable in this case stands for the energy of the configurations, thus the position of the lead in the BBM corresponds to the energy of the ground state of the polymer [63].

**Probability density of the position of the rightmost particle.** If $f(z)$ is set to the step function,

$$f(z) = \mathbb{1}_{\{z \geq 0\}} \tag{261}$$

then, replacing $f$ in Eq. (254), one gets

$$Q^{(f)}(t, x) = \mathbb{E}_0 \left( \prod_{k=1}^{N_t} \mathbb{1}_{\{X_k(t) \leq x\}} \right) = \mathbb{P}_0 \left( X_k(t) \leq x \text{ for all } k \in \{1, \cdots, N_t\} \right). \tag{262}$$

Hence $Q^{(f)}(t, x)$ for this choice coincides with the probability that all the particles (and in particular the rightmost one) are found at positions less than $x$ at time $t$, starting the process at position 0. An appropriate spatial derivative leads to the probability density of the position of the rightmost particle:

$$\mathbb{P} \left( \max_{1 \leq k \leq N_t} X_k(t) \in dx \right) = \frac{\partial Q^{(f)}(t, x)}{\partial x} dx. \tag{263}$$

**Probability of the number of particles within a given distance behind the lead.** Let us now consider a two-step function:

$$f(z) = (1 - \mu) \mathbb{1}_{\{z \geq \Delta\}} + \mu \mathbb{1}_{\{z \geq 0\}}, \tag{264}$$

where $\mu$ and $\Delta$ are positive constants, with $\mu \leq 1$. It is then easy to see that

$$
\begin{aligned}
Q^{(f)}(t,x) &= \mathbb{E}_0\left(\prod_{k=1}^{N_t}\mathbb{1}_{\{X_k(t)\leq x\}}\mu^{\mathbb{1}_{\{x-\Delta<X_k(t)\leq x\}}}\right) \\
&= \sum_{j=0}^{\infty}\mathbb{P}_0\left(\max_{1\leq k\leq N_t}X_k(t)\leq x; j \text{ particles in } (x-\Delta,x]\right)\times\mu^j.
\end{aligned}
\tag{265}
$$

This is the generating function of the joint probability to have a given number of particles in an interval $(x-\Delta,x]$ and no particle to the right of $x$. One may extract from $Q^{(f)}$ the probability density of the number of particles at a distance $\Delta$ from the lead.

Taking as an initial condition $f$ a more complicated step function, one may infer from the solution to the FKPP equation joint probabilities of particle numbers at different distances from the lead, useful to characterize the statistics of the number of particles in the tip of the BBM [64,65].

## 5.2 Solutions to the FKPP equation

While there is no analytical expression for the solution of the FKPP equation (258), a lot is known on its large-time asymptotics. The FKPP equation is best discussed for the function $u \equiv 1-Q^{(f)}$:

$$
\partial_t u(t,x) = D\partial_x^2 u(t,x) + \lambda\left\{u(t,x)-[u(t,x)]^2\right\}, \qquad \text{with} \qquad u(0,x)=1-f(x). \tag{266}
$$

**Preliminary considerations.** Our first observation is that the spatially constant functions $u(t,x)=1$ and $u(t,x)=0$ are two fixed points of this evolution equation, but of different nature: $u=1$ is stable while $u=0$ is unstable. This means that any perturbation $u=\epsilon\ll 1$ to the latter grows exponentially in time, while any perturbation $u=1-\epsilon$ to the former eventually disappears, $u$ being driven back to 1.

For non-constant initial conditions such that $0 \leq u(0,x) \leq 1$, the diffusion term $D\partial_x^2 u$ tends to smoothen the irregularities, the linear term $\lambda u$ induces locally an exponential growth in time, while the non-linear term $-\lambda u^2$ compensates the growth when $u \sim 1$, in such a way that $u \leq 1$ at all times.

The relevant initial conditions for the problems we want to address will be monotonically decreasing functions connecting 1 at $x \to -\infty$ to 0 at $x \to +\infty$. Diffusion and growth will result in the progressive invasion of the unstable state $u=0$ by the stable one $u=1$, through a right-moving wave front, whose shape converges to a universal function at large times, called a "traveling wave"; see Fig. 16. This convergence is formalized by a mathematical theorem [66] that we shall now quote.

### 5.2.1 The Bramson theorem

If one chooses an initial condition such that $u(0,x) \underset{x\to\infty}{\sim} e^{-\beta x}$ for some $\beta > 0$, for example

$$
u(0,x) = e^{-\beta x}\mathbb{1}_{\{x>0\}} + \mathbb{1}_{\{x\leq 0\}}, \tag{267}
$$

then, at large times, $u$ converges to a traveling wave, namely a front connecting 1 at $x \to -\infty$ to 0 at $x \to +\infty$. Its time evolution amounts to a translation along the real axis:

$$
u(t,x) \underset{t\to\infty}{\sim} U^{(\beta)}\left(x-m_t^{(\beta)}\right). \tag{268}
$$

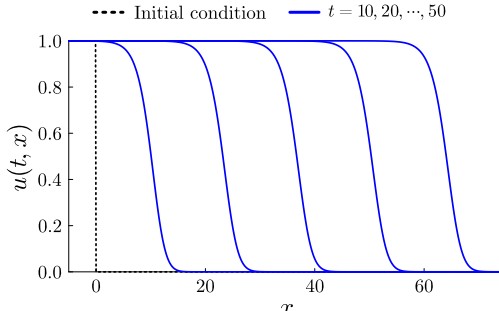

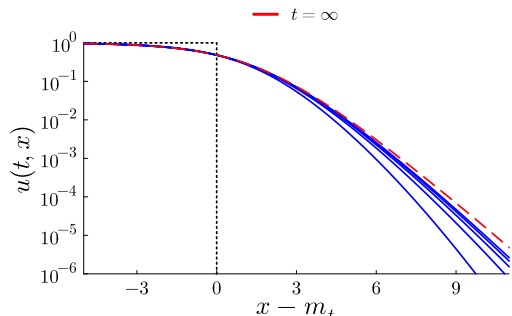

(a) $u(t,x)$ as a function of $x$ for five equally-spaced times between $t = 10$ and $t = 50$ (full lines, from left to right). The initial condition is a step function and is displayed in dotted lines.

(b) The same, but as a function of $x - m_t$ (i.e. in the frame of the front), in order to see the convergence to the asymptotic traveling wave (dashed line labeled "$t = \infty$"). The position $m_t$ solves $u(t, m_t) = \frac{1}{2}$.

Figure 16: Numerical solution of (a discretization of) the FKPP equation with $D = \frac{1}{2}$ and $\lambda = 1$. The model used for this calculation is the one introduced in Ref. [67].

The single-variable function $U^{(\beta)}$ is the asymptotic shape of the traveling wave. The function $m_t^{(\beta)}$ is the position of the wave front as a function of the time $t$. Let us introduce

$$v(\gamma) = D\gamma + \frac{\lambda}{\gamma}, \qquad \text{and } \gamma_0 \text{ such that } v'(\gamma_0) = 0. \tag{269}$$

Explicitly, as a function of the parameters of our equation (266),

$$\gamma_0 = \sqrt{\frac{\lambda}{D}}, \qquad \text{and} \qquad v(\gamma_0) = 2\sqrt{\lambda D}. \tag{270}$$

Then, according to Bramson's theorem [66], the position of the wave front depends on time as

$$m_t^{(\beta)} = \begin{cases} v(\beta)t + \mathcal{O}(1), & \text{for } \beta < \gamma_0, \\ v(\gamma_0)t - \frac{1}{2\gamma_0}\ln t + \mathcal{O}(1), & \text{for } \beta = \gamma_0, \\ v(\gamma_0)t - \frac{3}{2\gamma_0}\ln t + \mathcal{O}(1), & \text{for } \beta > \gamma_0, \end{cases} \tag{271}$$

in the limit of large $t$. (In the last case, we will afford to drop the superscript in $m_t^{(\beta)}$, denoting it by $m_t$, as it effectively has no $\beta$-dependence).

A few remarks are in order:

- The detailed form of the partial differential equation solved by $u$, and in particular the non-linearity, is not relevant for these asymptotics, as long as it falls within a certain class. One may for example replace $u^2$ by $u^3$ in Eq. (266): $m_t^{(\beta)}$ in Eq. (271) would remain the same.

- Initial conditions that decay asymptotically more rapidly than $e^{-\gamma_0 x}$, such as e.g. a Gaussian of variance $\sigma^2$,

$$u(0, x) = e^{-x^2/(2\sigma^2)}\mathbb{1}_{\{x>0\}} + \mathbb{1}_{\{x\leq 0\}}, \tag{272}$$

or a step function $u(0, x) = \mathbb{1}_{\{x\leq 0\}}$, all fall within the case $\beta > \gamma_0$.

- The large-$t$ asymptotics of the traveling wave velocity does not depend in a more detailed manner on the initial condition, except in the critical case $\beta = \gamma_0$ (second case in

Eq. (271)). In that case, the coefficient of $\ln t$ in the front position $m_t^{(\beta)}$ depends on the prefactor of the asymptotic exponential decay in the initial condition $U^{(\beta)}$. Had we, for example, a power $x^\nu$ with $\nu \in (-2, \infty)$ instead of a constant multiplying $e^{-\gamma_0 x}$, then the coefficient $-1/(2\gamma_0)$ would be replaced by $-(1-\nu)/(2\gamma_0)$. These subasymptotics have been further investigated recently; see e.g. Ref. [68].

Let us try to understand heuristically Bramson's theorem. We shall first investigate exact stationary solutions in a frame moving at a constant velocity, and then explain how these solutions are reached asymptotically at large times when one starts from some initial condition different from these solutions.

**Exact traveling waves.** We observe that $U(x - wt)$ is a solution of the FKPP equation of constant velocity $w$ if $\xi \mapsto U(\xi)$ obeys the ordinary non-linear differential equation

$$D\,U''(\xi) + w\,U'(\xi) + \lambda\,U(\xi)[1 - U(\xi)] = 0\,. \tag{273}$$

For $\xi \to +\infty$, $U(\xi) \to 0$, and this equation may be linearized about $U = 0$ by just dropping the last term, yielding an elementary second-order differential equation with constant coefficients. Solving the latter enables us to find the large-$\xi$ asymptotics of the shape of the front as a function of $w$.

We denote by $U_{\text{lin}}$ the solutions to the linearized equation, with $U_{\text{lin}}(\xi) \underset{\xi \to +\infty}{\simeq} U(\xi)$. There are three cases to distinguish:

- $w < v(\gamma_0)$: there is no real root of the characteristic polynomial of the differential equation, and hence no solution to the latter for which $U_{\text{lin}}$ is positive for all values of $\xi$.

- $w > v(\gamma_0)$: $U_{\text{lin}}$ is a linear combination of exponentials,

$$U_{\text{lin}}(\xi) = a e^{-\beta \xi} + b e^{-[\lambda/(D\beta)]\xi}\,, \qquad \text{with} \qquad a, b = \text{constants}, \tag{274}$$

and $\beta \equiv \left(w - \sqrt{w^2 - [v(\gamma_0)]^2}\right)/(2D)$. It is dominated by the first of these terms when $\xi$ is large. Note that solving this last equation for $w$ yields $w = v(\beta)$.

- $w = v(\gamma_0)$: this is the critical case in which the characteristic polynomial of the differential equation has a double root, yielding

$$U_{\text{lin}}(\xi) = (A\xi + B)e^{-\gamma_0 \xi}\,, \qquad \text{with} \qquad A, B = \text{constants}. \tag{275}$$

Note that the constant $B$ may be eliminated by a redefinition of the front position.

For completeness, let us mention that for large negative values of $\xi$, for which $U(\xi)$ approaches 1, the shape of the traveling wave may also be determined. Indeed, the equation for $U$ can be linearized about this stable fixed point $U = 1$, and solved as a combination of exponentials. We then see that the wave front approaches 1 exponentially fast:

$$1 - U(\xi) \simeq \text{const} \times \exp\left(\frac{\sqrt{w^2 + [v(\gamma_0)]^2} - w}{2D} \times \xi\right), \qquad \text{for large negative } \xi\,. \tag{276}$$

The transition region $\xi \approx 0$ is complicated to address analytically, since all terms in Eq. (273) are equally important.

**Convergence to the exact traveling wave and velocity selection.** We have found a one-parameter family of stationary solutions to the FKPP equation, parametrized by the velocity $w \geq v(\gamma_0)$. We shall now show heuristically that these exact solutions are reached asymptotically at infinite times, when one starts from one of the initial conditions underlying the Bramson theorem.

This can actually be understood from the linearized FKPP equation

$$\partial_t u_{\text{lin}}(t,x) = D\partial_x^2 u_{\text{lin}}(t,x) + \lambda u_{\text{lin}}(t,x). \tag{277}$$

We observe that any function of the form

$$u_\gamma(t,x) = e^{-\gamma[x-v(\gamma)t]} \tag{278}$$

is a solution, provided that $v(\gamma)$ is given by Eq. (269). The latter is interpreted as the velocity of the partial wave of "wave number" $\gamma$, and $\gamma v(\gamma)$ is the eigenvalue of the kernel of Eq. (277) that corresponds to the eigenfunction $e^{-\gamma x}$.

Any solution can be expanded on the basis (278):

$$u_{\text{lin}}(t,x) = \int \frac{d\gamma}{2i\pi} \tilde{f}(\gamma) u_\gamma(t,x). \tag{279}$$

The function $\tilde{f}(\gamma)$ is obviously the Mellin transform of the initial condition $u(0,x)$. If one picks the initial condition displayed in Eq. (267), then

$$\tilde{f}(\gamma) = \int_{-\infty}^{+\infty} dx\, u(0,x)\, e^{\gamma x} = \frac{\beta}{\gamma(\beta - \gamma)}. \tag{280}$$

The integration contour in Eq. (279) runs parallel to the imaginary axis and intersects the real axis between the two poles of $\tilde{f}$. Inserting the different elements in Eq. (279) and going to a frame moving at a constant velocity $w$,

$$u_{\text{lin}}(t,wt+\xi) = \int \frac{d\gamma}{2i\pi} \frac{\beta}{\gamma(\beta-\gamma)} e^{-\gamma[\xi+(w-v(\gamma))t]}. \tag{281}$$

For $t \to +\infty$, one may try to approximate $u_{\text{lin}}$ by a saddle-point estimate of the integral over $\gamma$. Treating the initial condition as a prefactor and keeping $\xi$ finite, the saddle-point equation reads $\gamma_s v'(\gamma_s) = w - v(\gamma_s)$. The saddle-point contribution to $u_{\text{lin}}$ then reads

$$u_{\text{lin}}(t,v(\gamma_0)t+\xi)\big|_{\text{saddle point}} \sim e^{-\gamma_s[\xi+(w-v(\gamma_s))t]}. \tag{282}$$

Requiring stationarity in the frame moving at velocity $w$ imposes that $w = v(\gamma_s)$. Then, the saddle-point equation boils down to $v'(\gamma_s) = 0$, meaning that $\gamma_s$ minimizes the velocity of the partial wave labeled by $\gamma$: it turns out to coincide with $\gamma_0$. Hence the velocity is $w = v(\gamma_0)$, which corresponds to the second and third cases of Bramson's theorem (271). The front has the following shape:

$$u_{\text{lin}}(t,v(\gamma_0)t+\xi)\big|_{\text{saddle point}} \sim e^{-\gamma_0\xi}. \tag{283}$$

Note that while we get the correct exponential, we do not get any prefactor linear in $\xi$ as in Eq. (275). A more detailed analysis is in order; see the next section.

Considering the prefactor encoded in $\tilde{f}(\gamma)$, we see that the saddle-point may actually not be the dominant contribution. Indeed, if $\gamma_0 > \beta$, then one first needs to move the contour across the pole at $\gamma = \beta$. This yields a contribution

$$u_{\text{lin}}(t,wt+\xi)\big|_{\text{pole at }\gamma=\beta} = e^{-\beta[\xi+(w-v(\beta))t]} \underset{w=v(\beta)}{=} e^{-\beta\xi}, \tag{284}$$

where we have again required stationarity to obtain the last equality. This pole contribution dominates over the saddle-point contribution for large enough $\xi$. The wave-front velocity is now $w = v(\beta)$: we have recovered the first case of Bramson's theorem (271).

To conclude, we have found a family of exact traveling wave solutions to the FKPP equation, that we have characterized by their constant velocities and asymptotic shapes, which are functions of the latter. From an analysis of the linearized FKPP equation in the infinite-time limit, we have then shown that a given initial condition evolves to a stationary function in a frame moving at the velocity given by the infinite-time limit of the Bramson theorem. The shapes of these functions are the ones of the exact traveling wave solutions, up to prefactors. A more detailed analysis is needed to determine subasymptotics of the velocity and front shape: this is what we will investigate in the next section.

From now on, we will focus on the last case in Eq. (271) (i.e. $\beta > \gamma_0$), which will be the only one of interest for the applications in particle physics that we shall discuss in this review. A convenient initial condition in this class is the simple step function.

### 5.2.2   Finite-time corrections

We now discuss finite-time corrections to the asymptotic velocity of the front as well as to the shape ahead of its position, where $u \ll 1$.

Let us solve more accurately the linearized FKPP equation (277) with the step function $u_{\text{lin}}(0, x) = \mathbb{1}_{\{x<0\}}$ as an initial condition, corresponding to $\tilde{f}(\gamma) = 1/\gamma$. Equation (279) can then be evaluated exactly in terms of the complementary error function $\text{erfc}(z) \equiv \frac{2}{\sqrt{\pi}} \int_z^\infty dt\, e^{-t^2}$,

$$u_{\text{lin}}(t,x) = \frac{\sqrt{\pi}}{2}\text{erfc}\left(\frac{x}{\sqrt{4Dt}}\right) \times e^{\lambda t} \underset{x \to \infty}{\simeq} \frac{\sqrt{\pi Dt}}{x} e^{-x^2/(4Dt)+\lambda t}\,. \tag{285}$$

In order to find a frame in which the solution looks stationary, we follow lines of constant $u_{\text{lin}}$. Namely, we define the position $m_{t,\text{lin}}$ such that

$$u_{\text{lin}}(t, m_{t,\text{lin}}) = 1\,. \tag{286}$$

An easy calculation leads to the following expressions for the large-time behaviors of $m_{t,\text{lin}}$ and of $u_{\text{lin}}$ around the latter position read

$$m_{t,\text{lin}} = v(\gamma_0)t - \frac{1}{2\gamma_0}\ln t + \cdots, \qquad \text{and} \qquad u_{\text{lin}}(t, m_{t,\text{lin}} + \xi) \simeq e^{-\gamma_0 \xi}\,, \tag{287}$$

where $v(\gamma_0)$ and $\gamma_0$ are given in Eq. (270). Compare to Eq. (271) and Eq. (275): one sees that the leading term in the front position is the one given by Bramson's theorem, and the leading front shape is also the correct one. The subleading terms are however not correct: instead of $-1/(2\gamma_0)\ln t$ one should actually have $-3/(2\gamma_0)\ln t$ in the front position, and there should be an extra linear factor $\xi$ multiplying the exponential in the front shape.

This is a motivation for analyzing the FKPP equation (266) in the frame of the asymptotic traveling wave moving at velocity $v(\gamma_0)$, and for the rescaled function

$$h(t, \xi) = e^{\gamma_0 \xi} \times u(t, v(\gamma_0)t + \xi)\,. \tag{288}$$

Equation (266) translates into the following exact equation for $h$:

$$\partial_t h(t, \xi) = D\partial_\xi^2 h(t, \xi) - \lambda e^{-\gamma_0 \xi} \times [h(t, \xi)]^2\,. \tag{289}$$

Because of the behavior of $u$ for positive $\xi$, see Eq. (287), it is clear that the last term in the right-hand side vanishes exponentially for $\xi \gg 1$. In this region, the equation for $h$ essentially

reduces to the diffusion equation. For negative values of $\xi$ instead, the exponential factor grows very steeply as one dials $-\xi$ up, and thus the time evolution drives $h$ fastly to zero in this region. Hence the nonlinearity effectively acts as an absorptive boundary condition on the linear diffusion equation, localized at $\xi = 0$.

We thus need to solve the diffusion equation

$$\partial_t h(t,\xi) = D\partial_\xi^2 h(t,\xi), \qquad \text{with the initial condition} \qquad h(0,\xi) = e^{\gamma_0 \xi}\mathbb{1}_{\{\xi<0\}}, \qquad (290)$$

localized near $\xi = 0$, and with the boundary condition $h(t,0) = 0$. We can start from the solution without boundaries, which is essentially a Gaussian:

$$h(t,\xi) = \frac{1}{2}e^{\gamma_0(\xi+\gamma_0 Dt)}\text{erfc}\left(\frac{\xi}{\sqrt{4Dt}} + \gamma_0\sqrt{Dt}\right) \underset{t\to\infty}{\simeq} \frac{1}{\gamma_0}\frac{1}{\sqrt{4Dt}}e^{-\xi^2/(4Dt)}. \qquad (291)$$

We then implement the absorptive boundary at $\xi = 0$ through the method of images. We find

$$h(t,\xi) \simeq \text{const} \times \frac{\xi}{t^{3/2}}e^{-\xi^2/(4Dt)} \qquad (292)$$

[It is straightforward to show directly that this function solves Eq. (290), with an initial condition $h(0,\xi) \propto \delta(\xi)$ which is a good approximation to the actual initial condition (290) viewed from large times]. We see that the main finite-time correction is an effective Gaussian cutoff at a distance $\mathcal{O}(\sqrt{2Dt})$ to the right of the origin of this moving frame. The second observation is that $h$ does not tend to a stationary function: it vanishes with $t$ like $1/t^{3/2} = \exp\left[\gamma_0(-\frac{3}{2\gamma_0}\ln t)\right]$. Intuitively, this means that the absorptive boundary is "too fast": it absorbs too much mass.

In order to fix the non-stationarity of $h$, one may pull the frame back by $\frac{3}{2\gamma_0}\ln t$, which amounts to changing to a frame moving exactly at the Bramson velocity. One checks that

$$\boxed{\begin{aligned} u(t,x) &\simeq \text{const} \times (x-m_t)e^{-\gamma_0(x-m_t)}\exp\left(-\frac{(x-m_t)^2}{2\gamma_0 v''(\gamma_0)t}\right), \\ \text{with} \quad m_t &= v(\gamma_0)t - \frac{3}{2\gamma_0}\ln t + \text{const} + o(1), \end{aligned}} \qquad (293)$$

solves the FKPP equation (266) in the region $x-m_t \gg 1$, up to terms suppressed by $1/t$. We have written the diffusion coefficient $D$ as $\gamma_0 v''(\gamma_0)/2$, where in the present case, $v(\gamma)$ is the function defined in Eq. (269), because the latter form turns out to be more general.

The detailed shape of the traveling wave, obtained from a numerical solution of a (discretized) FKPP equation, is displayed in Fig. 17a. One sees in particular that $h(t,\xi)$ converges to a straight line $h(t \gg 1,\xi) \propto \xi$ at large times, which is the sign that the non-linearity acts as an absorptive boundary. The time-dependence of the position of the wave front is shown in Fig. 17b.

Let us underline once again that the fact that the traveling waves are universal, in the sense that their very existence as asymptotic solutions, their shape and their position, depend neither on the detailed form of the evolution equation nor on the precise form of the initial conditions. While it is not possible to give a precise and complete mathematical definition of the universality class, we can say informally that many evolution equations, continuous or discrete, that encode a form of diffusion and an exponential growth mechanism, supplemented by a nonlinearity which limits the growth when the evolved function becomes on the order of some predefined constant, would admit Bramson's traveling waves as large-time solutions. The parameters of the latter are deduced from an analysis of the solutions to the linearized equation: it is enough to work out the expression of $v(\gamma)$ such that $u_\gamma(t,x)$ defined in Eq. (278)



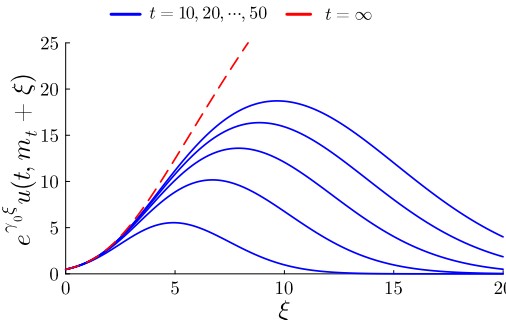

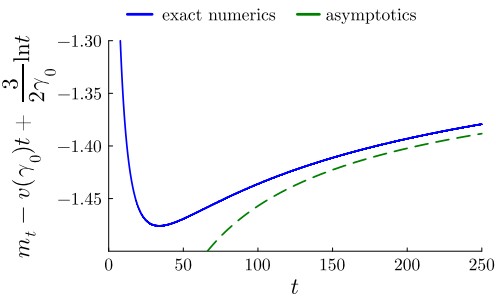

(a) Shape of the front for different times, rescaled by the leading exponential. [The displayed function actually amounts to $h(t, \xi)$ defined in Eq. (288)].

(b) Position of the front as a function of time, of which we have subtracted the leading terms. *Dashed line*: $\text{const} - \frac{3}{2\gamma_0^2} \sqrt{\frac{2\pi}{\gamma_0 v''(\gamma_0)}} \frac{1}{\sqrt{t}}$ [see Eq. (294)].

Figure 17: Detailed properties of the traveling waves displayed in Fig. 16.

solves the linearized FKPP equation. Then, Eq. (293) is an asymptotic solution to the non-linear evolution equation.

A systematic approach to the finite-$t$ corrections along the line of thought sketched out here, for the FKPP equation and for more general equations, can be found in the appendix of Ref. [69] and in Ref. [70]. In the latter reference, it was shown heuristically that, beyond Bramson's theorem, at least another subleading correction to the front position, of order $1/\sqrt{t}$, is also universal. Including this term, the time dependence of the position of the front reads

$$m_t = v(\gamma_0)t - \frac{3}{2\gamma_0} \ln t + \text{const} - \frac{3}{2\gamma_0^2} \sqrt{\frac{2\pi}{\gamma_0 v''(\gamma_0)}} \frac{1}{\sqrt{t}} + \mathcal{O}(1/t). \qquad (294)$$

There is no universal expression for the constant term: it obviously depends on the definition of the front position, and it is a functional of the initial condition.

## 6 Applications to scattering in particle physics

In this section, we shall connect the effective picture of quantum evolution at high energy developed in Secs. 2 through 4 to the discussion of general stochastic processes in the last section, in order to better understand high-energy scattering in particle physics. First, we will show that the physics of the forward elastic $S$-matrix element in high-energy scattering is essentially that of the statistics of extremes in binary branching processes, and that the BK equation can be established in exactly the same way (Sec. 6.1). Once this parallel has been established, we will be able to transpose the results obtained on the BBM to QCD (Sec. 6.2). We will then show how to map mathematically the BK equation to the FKPP equation to make the link between the two equations even sharper (Sec. 6.3). Finally, we will briefly discuss the applications to the interpretation of the data on the deep-inelastic scattering total cross section (Sec. 6.4).

We shall generally assume the nucleus large enough, compared to the onium, that it may be considered uniform and homogeneous in the transverse plane. In other words, the $S$-matrix element for the forward elastic scattering of an onium of definite transverse size depends only on the modulus of the latter and on the rapidity:

$$S(Y, x_0^\perp, x_1^\perp) \longrightarrow S(Y, |x_{01}^\perp|). \qquad (295)$$

In order to simplify the notations, we shall write $x_{kl} \equiv |x_{kl}^\perp|$.

## 6.1 Onium-nucleus scattering amplitude as an extreme-value problem

### 6.1.1 BK as an evolution equation for observables on a binary branching process

In the dipole model derived in Secs. 3 and 4, the forward elastic $S$-matrix element for an onium takes the form of the convolution of two quantities: on the one hand, the probability that the Fock state of the onium at rapidity $Y$ consists in a set of dipoles of given sizes; on the other hand, of the $S$-matrix element for the scattering of these dipoles as bare states. The latter are produced in a binary branching process in rapidity. The former turns out to boil down to the product of the $S$-matrix elements $S_\bullet$ for the scattering of bare dipoles, given e.g. by the McLerran-Venugopalan model (238).

The strong parallel between the BK and the FKPP equations and between the underlying physics they describe can already be appreciated by simply putting this observation in equation. Let us call $\mathbb{E}_{\ln[4/(x_{01}^2 Q_A^2)]}$ the expectation value over realizations of dipole evolution at rapidity $Y$ starting with a dipole of size $x_{01}$, and $\{r_k\}$ the set of the sizes of the dipoles in the Fock state at the time of the interaction. Then

$$S(Y, x_{01}) = \mathbb{E}_{\ln[4/(x_{01}^2 Q_A^2)]}\left(\prod_k S_\bullet(r_k)\right). \tag{296}$$

Next, write $x \equiv \ln[4/(x_{01}^2 Q_A^2)]$, $X_k \equiv \ln[4/(r_k^2 Q_A^2)]$, and set $f(z) = S_\bullet\left(2e^{z/2}/Q_A\right)$. Then, in terms of the latter function and of the former variables, the right-hand side reads

$$\mathbb{E}_x\left(\prod_k f(-X_k)\right), \tag{297}$$

which identifies with the right-hand side of Eq. (256). Hence $S$ here is like $Q^{(f)}$ there, up to the substitution $\ln[4/(x_{01}^2 Q_A^2)] \leftrightarrow x$. The evolution variable was the time $t$ in Sec. 5, and is the rapidity $Y$ here, up to a constant that we will choose in the most convenient way.

We took the logarithm of the dipole sizes for the following reason. The evolution of the particle position in the BBM consists in a diffusion, namely is an additive process (and the kernel is translation invariant). By contrast, the evolution of the dipole size is a multiplicative process (and the kernel is scale invariant): it becomes additive on a logarithmic scale.

Once we have formulated $S$ in this manner, we recognize that the expectation value in Eq. (296) is over a Markovian binary stochastic process. An evolution equation can then be derived in the same way as the FKPP equation was established for $Q^{(f)}$; see Sec. 5.1.2. We express $S$ at rapidity $Y + dY$ with the help of $S$ at $Y$ by singling out a first step $dY$ of rapidity in the evolution of an onium of initial size $x_{01}$. In this infinitesimal rapidity interval, either the onium branches (with probability proportional to $dY$) into two dipoles of sizes $x_{02}$ and $x_{21}$, or it does not branch (with probability $Z_{\bar{q}q}(Y)$). In the first case, the subsequent evolution over the rapidity interval $Y$ is that of two independent dipoles, which eventually interact as statistically-independent sets of dipoles of sizes $\{r_{k_1}^{(1)}\}$ and $\{r_{k_2}^{(2)}\}$ respectively. In the second case, the subsequent evolution is that of a single dipole of size $x_{01}$. This leads to the following equation for $S$:

$$S(Y + dY, x_{01}) = dY\, \bar{\alpha} \int \frac{d^2 x_2^\perp}{2\pi} \frac{x_{10}^2}{x_{02}^2 x_{21}^2} \mathbb{E}_{\ln[4/(x_{02}^2 Q_A^2)]}\left(\prod_{k_1} S_\bullet\left(r_{k_1}^{(1)}\right)\right)$$

$$\times \mathbb{E}_{\ln[4/(x_{21}^2 Q_A^2)]}\left(\prod_{k_2} S_\bullet\left(r_{k_2}^{(2)}\right)\right) + Z_{\bar{q}q}(dY) S(Y, x_{01}). \tag{298}$$

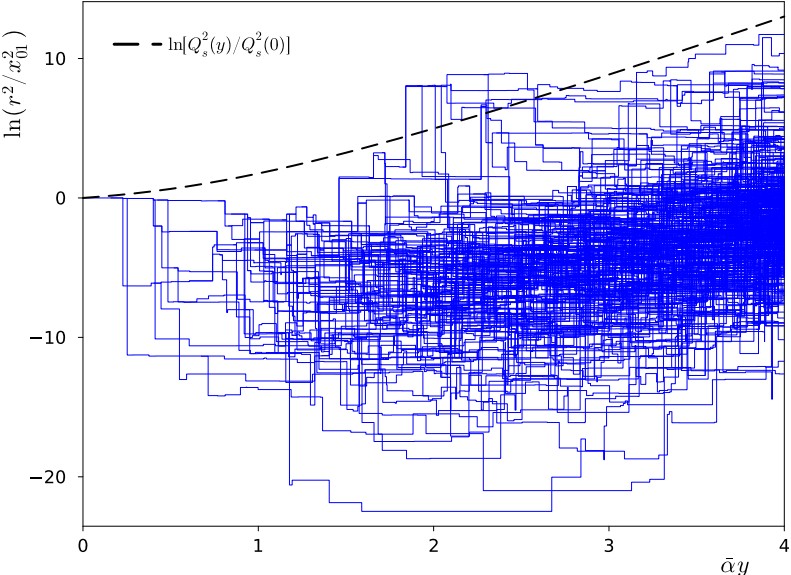

Figure 18: Size of the dipoles (on a logarithmic scale) present in a typical realization of the QCD evolution, as a function of the rapidity, starting with a dipole of size $x_{01}$. Only the branches which end at the final rapidity $\bar{\alpha}Y = 4$ with dipoles whose size-to-impact-parameter ratio is less than $\frac{1}{2}$ are shown, in order to make the figure less busy: they are the dominant ones in the scattering of a small onium off a nucleus.

After substitutions according to Eq. (296) and (165), we take the limit $dY \to 0$, and recover the BK equation (234).

As for the initial condition $S(Y = 0, x_{01})$, which determines $f$ in Eq. (297), we anticipate that on the relevant scale, the MV model (238) exhibits a very sharp transition (in a sense that we will explain below) between $S_\bullet = 1$ (for $x_{01} \ll 2/Q_A$) and $S_\bullet = 0$ (for $x_{01} \gg 2/Q_A$). It may actually be well-approximated by the indicator function

$$S_\bullet(r) \simeq \mathbb{1}_{\{rQ_A/2 \le 1\}}. \tag{299}$$

Within this approximation, $S(Y, x_{01})$ solves exactly the same equation as the *probability that there is no dipole of size larger than $2/Q_A$ in the Fock state of the onium at rapidity $Y$*; see again the parallel with Sec. 5.1.2. In this sense, the calculation of high-energy scattering amplitudes becomes an extreme-value problem: a given realization of the Fock state of the onium brings a non-zero contribution to the amplitude $T = 1 - S$, if the largest dipole in that realization is larger than the length $2/Q_A$ that characterizes the target.[15]

This elegant statistical interpretation of high-energy scattering is also practical. Indeed, since the physics of BK and FKPP is essentially the same, the methods to understand the former should apply to the latter. Let us investigate this, before mapping more precisely the BK equation to the FKPP equation.

### 6.1.2 Color dipole model versus BBM

At this point, let's say a word about the differences between the color dipole model and the BBM. In the latter, there are two distinct elementary processes: diffusion in space and branching into two particles at a given point. In the former, dipoles branch into two dipoles of dif-

---

[15]Note however that if $T$ can be interpreted as an abstract probability on the Fock state of the onium, the physical probability that a dipole $(x_0^\perp, x_1^\perp)$ scatters at rapidity $Y$, namely the inelastic cross section differential in the impact parameter $(x_0^\perp + x_1^\perp)/2$, is of course $1 - |S(Y, x_0^\perp, x_1^\perp)|^2$.

ferent sizes: the equivalent of diffusion occurs in the same process as splitting. But this turns out not fundamental. More surprisingly maybe, the fact that the kernel of the BK equation is a non-local integral operator does not preclude the parallel with FKPP either, as we will see technically in more detail later.

However, if we want to compare the realizations of the dipole evolution and of the BBM, there is one property of the dipole model that requires more attention: the fact that the emission rate of small dipoles can be arbitrarily large. This means, in effect, that at all rapidities, each realization has an infinite number of small dipoles, which is very different from what happens in the BBM. The point is that arbitrarily-small dipoles do not have offspring of size larger than the typical size $2/Q_A$ associated with the target with non-negligible probability. Therefore, they do typically not contribute to the overall onium-nucleus $S$-matrix element. Thus only a "BBM-like" subtree of the quantum evolution, of which such dipoles are excluded, effectively contributes to the observables of interest. The complete discussion is quite sophisticated, and would go well beyond the scope of these lectures. We shall refer the interested reader to e.g. Refs. [71,72].

One typical realization of dipole evolution generated with the Monte Carlo code described in Appendix A of Ref. [73][16] is shown in Fig. 18. It may be compared qualitatively to the realization of the BBM displayed in Fig. 15.

## 6.2 Applying general methods for branching processes to analyze the BK equation in position space

The BK equation (234) is best analyzed for the amplitude $T \equiv 1 - S$. For this quantity, it becomes the following non-linear integro-differential equation:

$$\partial_Y T(Y, x_{01}) = \bar{\alpha} \int \frac{d^2 x_2^{\perp}}{2\pi} \frac{x_{01}^2}{x_{02}^2 x_{21}^2} \left[ T(Y, x_{02}) + T(Y, x_{21}) - T(Y, x_{01}) - T(Y, x_{02}) T(Y, x_{21}) \right]. \quad (300)$$

The non-linearity is negligible compared to the linear terms when $T \ll 1$ (weak scattering limit). The equation obtained by dropping the non-linear term is the Balitsky-Fadin-Kuraev-Lipatov (BFKL) equation [49,50]:

$$\partial_Y T_{\text{BFKL}}(Y, x_{01}) = \bar{\alpha} \int \frac{d^2 x_2^{\perp}}{2\pi} \frac{x_{01}^2}{x_{02}^2 x_{21}^2} \left[ T_{\text{BFKL}}(Y, x_{02}) + T_{\text{BFKL}}(Y, x_{21}) - T_{\text{BFKL}}(Y, x_{01}) \right]. \quad (301)$$

We are first going to investigate this weak-scattering regime, solving the BFKL equation. From our experience with general branching processes in Sec. 5, we know that it is also a first step to understand the solutions to the BK equation. As a matter of fact, the method we shall use in this section to solve the BK equation is completely parallel to the heuristic calculations presented in Sec. 5.2.

### 6.2.1 Solution to the BFKL equation

As a general rule, integro-differential equations such as (301) are solved like an imaginary-time Schrödinger equation, by first looking for eigenfunctions of the integral kernel. The latter plays the role of a Hamiltonian operator.

Since the BFKL kernel is scale-invariant, the power functions $x_{01}^{2\gamma}$ diagonalize it. The latter actually form an (over)complete basis of eigenfunctions labeled by the continuous parameter $\gamma$, which we may use to expand the solutions of the BFKL equation. Denoting by $\chi(\gamma)$ the corresponding eigenvalues, any solution is given in terms of a superposition of such eigenfunctions,

---

[16]See also the earlier implementation described in Ref. [74].

that we write

$$T_{\text{BFKL}}(Y, x_{01}) = \int \frac{d\gamma}{2i\pi} \tilde{f}(\gamma) \left( \frac{x_{01} Q_A}{2} \right)^{2\gamma} e^{\bar{\alpha} Y \chi(\gamma)}. \tag{302}$$

The integration contour runs in the complex plane, parallel to the imaginary axis. We may choose for example $(\frac{1}{2} - i\infty, \frac{1}{2} + i\infty)$. We have introduced the natural scale $Q_A/2$ in order to make the coefficient function $\tilde{f}(\gamma)$ dimensionless.

The calculation of the eigenvalues $\chi(\gamma)$ can be performed in different ways (see e.g. [5, 12]). We first write the definition of $\chi(\gamma)$ as the eigenvalue of the BFKL kernel associated to the eigenfunction $x_{01}^{2\gamma}$, introducing a regularization for the collinear divergences:

$$\chi(\gamma) = \lim_{\epsilon \to 0} \int \frac{dz d\bar{z}}{4i\pi} \frac{1}{|z|^{2-2\epsilon} |1-z|^{2-2\epsilon}} \left( |z|^{2\gamma} + |1-z|^{2\gamma} - 1 \right), \tag{303}$$

where we have used scale invariance to make the integrand dimensionless, and we have switched from $x_2^\perp \in \mathbb{R}^2$ to the complex variable $z \in \mathbb{C}$. We then introduce the following integral:

$$I_{\alpha,\beta} = \int \frac{dz d\bar{z}}{2i\pi} |z|^{2\alpha-2} |1-z|^{2\beta-2} = \frac{\Gamma(\alpha)\Gamma(\beta)}{\Gamma(\alpha+\beta)} \frac{\Gamma(1-\alpha-\beta)}{\Gamma(1-\alpha)\Gamma(1-\beta)} \tag{304}$$

(See [75] and references therein; an elementary proof of this formula may be found in the Appendix of Ref. [76]). In terms of $I_{\alpha,\beta}$,

$$\chi(\gamma) = \lim_{\epsilon \to 0} \left[ \frac{1}{2} \left( I_{\gamma+\epsilon,\epsilon} + I_{\epsilon,\gamma+\epsilon} - I_{\epsilon,\epsilon} \right) \right]. \tag{305}$$

A straightforward calculation leads to

$$\chi(\gamma) = 2\psi(1) - \psi(\gamma) - \psi(1-\gamma), \qquad \text{where} \qquad \psi(z) = \frac{d \ln \Gamma(z)}{dz}. \tag{306}$$

This meromorphic function has simple poles for all integer values of $\gamma$ which give rise to essential singularities in the integrand in Eq. (302). The most important ones are those closest to the integration contour, at $\gamma = 0$ and $\gamma = 1$. The following expansion ("collinear model" [77]) is known to describe quite accurately the principal branch of $\chi(\gamma)$:

$$\chi(\gamma) \simeq \frac{1}{\gamma} + \frac{1}{1-\gamma} + 4(\ln 2 - 1). \tag{307}$$

Note that we may use the eigenvalues $\chi(\gamma)$ in order to write the BFKL kernel in a handy synthetic way. Since $\gamma$ is conjugate to $\ln x_{01}^2$, we check that

$$\partial_Y T_{\text{BFKL}}(Y, x_{01}) = \bar{\alpha} \chi \left( \partial_{\ln x_{01}^2} \right) T_{\text{BFKL}}(Y, x_{01}), \tag{308}$$

is tantamount to the BFKL equation: we simply verify that $T_{\text{BFKL}}(Y, x_{01})$ in Eq. (302) solves Eq. (308).

**Initial condition.** Setting $Y = 0$ and inverting Eq. (302), we see that $\tilde{f}(\gamma)$ is the Mellin transform of the initial condition $T_{\text{BFKL}}(Y = 0, x_{01})$,

$$\tilde{f}(\gamma) = \int_0^{+\infty} \frac{dx_{01}^2}{x_{01}^2} \left( \frac{x_{01} Q_A}{2} \right)^{-2\gamma} T_{\text{BFKL}}(Y = 0, x_{01}). \tag{309}$$

Let us consider a couple of toy initial conditions that are physically relevant, and for which we have a simple explicit Mellin transform:

- Saturated initial condition:

$$T_{\text{BFKL}}(Y = 0, x_{01}) = 1 - \exp\left[-\left(\frac{x_{01}^2 Q_A^2}{4}\right)^\beta\right] \implies \tilde{f}(\gamma) = \frac{1}{\gamma}\Gamma\left(1 - \frac{\gamma}{\beta}\right). \qquad (310)$$

For $\beta = 1$, this function can be understood as an asymptotic limit of the MV model (238). We added a parameter $\beta$ in order to control the steepness of the small-$x_{01}$ asymptotics: we will see that this will enable us to probe the different regimes of the Bramson theorem that we stated in Sec. 5.2.1.

- "Dilute" initial condition:

$$T_{\text{BFKL}}(Y = 0, x_{01}) = \alpha_s^2 \min\left[\left(\frac{x_{01}^2 Q_A^2}{4}\right)^\beta, 1\right] \implies \tilde{f}(\gamma) = \alpha_s^2 \frac{\beta}{\gamma(\beta - \gamma)}. \qquad (311)$$

For $\beta = 1$, this form of initial condition coincides aymptotically with the forward elastic scattering of a dipole of size $x_{01}$ off a dipole of size $2/Q_A$ at the lowest order $\alpha_s^2$ of perturbation theory. Up to the factor $\alpha_s^2$, this is the same function as the one in Eq. (280) studied in Sec. 5.

In both cases, the two most important poles of the meromorphic function $\tilde{f}(\gamma)$, the ones closest to the integration contour, are located at $\gamma = 0$ and $\gamma = \beta$.

**Large-rapidity limit.** For large values of the parameters, the integrand in Eq. (302) is dominated by a saddle point $\gamma_s$ located in the interval $(0, 1)$. Its position depends on the region of the parameters $Y$ and $x_{01}$ in which one is interested. Ignoring the prefactors for the moment, the saddle-point equation reads

$$\bar{\alpha}Y \chi'(\gamma_s) + \ln\left(\frac{x_{01}^2 Q_A^2}{4}\right) = 0, \qquad (312)$$

and the leading contribution to $T_{\text{BFKL}}$ at the saddle point obviously writes

$$T_{\text{BFKL}}(Y, x_{01}) \sim \left(\frac{x_{01} Q_A}{2}\right)^{2\gamma_s} e^{\bar{\alpha}Y \chi(\gamma_s)}. \qquad (313)$$

We shall assume $\bar{\alpha}Y$ large in any case. If we keep $x_{01}$ finite while dialing the rapidity up, the saddle-point $\gamma_s$ gets closer to the minimum of $\chi(\gamma)$ on the real axis located at $\frac{1}{2}$. $T_{\text{BFKL}}$ then exhibits an exponential growth with $\bar{\alpha}Y$ for all values of $x_{01}$. Hence it eventually overshoots the unitarity limit $T = 1$, first in the regions of large $x_{01}$, with the initial conditions we have picked. This is a sign of the breakdown of the linearization leading to the BFKL equation, since as we have already pointed out, the main virtue of the BK equation is to preserve unitarity [see Eq. (235)].

However, with initial conditions which exhibit the small-$x_{01}$ asymptotics of the MV model, the BFKL equation may be valid above some $Y$-dependent value of $x_{01}$. Accordingly, we pick $x_{01}$ such that $T_{\text{BFKL}}$ be approximately constant and of order 1, while $\bar{\alpha}Y \to +\infty$. This new condition to enforce also reads $\ln T_{\text{BFKL}} \simeq 0$. It is implemented by equating the logarithm of Eq. (313) to zero:

$$\bar{\alpha}Y \chi(\gamma_s) + \gamma_s \ln\left(\frac{x_{01}^2 Q_A^2}{4}\right) \simeq 0. \qquad (314)$$

Combining this equation to the saddle-point equation (312), we find that $\gamma_s$ is a pure number that solves the following equation:

$$\chi(\gamma_s) = \gamma_s \chi'(\gamma_s) \implies \gamma_s = 0.627\cdots. \tag{315}$$

Let us denote by $2/Q_s(Y)$ the value of $x_{01}$ for which $T_{\text{BFKL}} = 1$:

$$T_{\text{BFKL}}(Y, 2/Q_s(Y)) = 1 \implies \ln \frac{Q_s^2(Y)}{Q_A^2} \simeq \bar{\alpha}Y \chi'(\gamma_s) + o(\bar{\alpha}Y). \tag{316}$$

$Q_s(Y)$ is the saturation momentum at rapidity $Y$, which controls the transition between the linear and non-linear regimes of the BK equation. The logarithm of $Q_s(y)$ normalized by $Q_A$ may also be interpreted as the expectation value of the logarithm of the size of the largest dipole present in the realization at a given rapidity $y$, normalized by $x_{01}$. We have plotted it in Fig. 18 (in dashed lines), for comparison with the logarithm of the dipole sizes present in an actual realization.

In terms of the saturation momentum, the saddle-point contribution (313) writes

$$T_{\text{BFKL}}(Y, x_{01})|_{\text{saddle point}} \sim \left( \frac{x_{01}^2 Q_s^2(Y)}{4} \right)^{\gamma_s} = \exp\left[ -\gamma_s \left( \ln \frac{4}{x_{01}^2 Q_A^2} - \ln \frac{Q_s^2(Y)}{Q_A^2} \right) \right]. \tag{317}$$

If $\beta < \gamma_s$, then there is a contribution from the pole at $\gamma = \beta$ in the initial condition which is larger than the saddle-point contribution

$$T_{\text{BFKL}}(Y, x_{01})|_{\text{pole at } \gamma = \beta} \sim \left( \frac{x_{01}^2 Q_A^2}{4} \right)^{\beta} e^{\bar{\alpha}Y \chi(\beta)} = \exp\left[ -\beta \left( \ln \frac{4}{x_{01}^2 Q_A^2} - \ln \frac{Q_s^{(\beta)2}(Y)}{Q_A^2} \right) \right], \tag{318}$$

with

$$\ln \frac{Q_s^{(\beta)2}(Y)}{Q_A^2} = \bar{\alpha}Y \frac{\chi(\beta)}{\beta}. \tag{319}$$

In both cases, we see that ahead of the saturation scale (i.e. for $1/x_{01} \gg Q_s(Y)$), a solution that shares the properties of traveling waves emerges. The correspondence with the BBM follows from the identification

$$\bar{\alpha}Y \leftrightarrow t, \qquad \ln \frac{4}{x_{01}^2 Q_A^2} \leftrightarrow x, \qquad \ln \frac{Q_s^{(\beta)2}(Y)}{Q_A^2} \leftrightarrow m_t^{(\beta)}. \tag{320}$$

The asymptotic (large-rapidity or large-time) velocities of the traveling waves respectively read

$$\begin{cases} \chi'(\gamma_s) = \chi(\gamma_s)/\gamma_s \leftrightarrow v(\gamma_s), & \text{in the case in which } \beta > \gamma_s, \\ \chi(\beta)/\beta \leftrightarrow v(\beta), & \text{for } \beta < \gamma_s. \end{cases} \tag{321}$$

$\gamma_s$ identifies of course with $\gamma_0$ discussed in Sec. 5.2.1.

The relevant initial condition in QCD is $\beta = 1$ (or $\beta$ close to 1 to account phenomenologically for subleading effects in the form of a small anomalous dimension $2(\beta - 1)$ fitted to the data; see e.g. [78]). Thus the dominant contribution is always the saddle point (317). The case $\beta < \gamma_s$ matches to one of the Bramson's cases, see Eq. (271). We shall focus exclusively on the former in what follows.

**Expanding about the saddle-point.** Through our saddle-point analysis, we have obtained the leading term in the saturation scale (316) at large rapidities, and the asymptotic shape of the front (317). We may try to get subleading terms refining the asymptotic solution to the BFKL equation by expanding systematically the integrand of Eq. (302) about the saddle point. We replace $\chi(\gamma)$ by its second-order expansion around $\gamma_s$:

$$\chi(\gamma) = \chi(\gamma_s) + \chi'(\gamma_s)(\gamma - \gamma_s) + \frac{\chi''(\gamma_s)}{2}(\gamma - \gamma_s)^2 + \mathcal{O}\left[(\gamma - \gamma_s)^3\right]. \tag{322}$$

We then change variable $\gamma \to \nu$, with $\gamma = \gamma_s + i\nu$ and where $\nu$ describes the real axis. Equation (302) becomes a Gaussian integral, which can readily be performed. We get

$$T_{\text{BFKL}}(Y, x_{01}) = \frac{f(\gamma_s)}{\sqrt{2\pi\chi''(\gamma_s)\bar{\alpha}Y}} \exp\left\{-\gamma_s\left(\ln\frac{4}{x_{01}^2 Q_A^2} - \chi'(\gamma_s)\bar{\alpha}Y\right)\right.$$
$$\left. - \frac{\left\{\ln[4/(x_{01}^2 Q_A^2)] - \chi'(\gamma_s)\bar{\alpha}Y\right\}^2}{2\chi''(\gamma_s)\bar{\alpha}Y}\right\}. \tag{323}$$

The $Y$-dependence of the prefactor may be incorporated to the exponent. Defining

$$\ln\frac{Q_{s(\text{lin})}^2(Y)}{Q_A^2} = \chi'(\gamma_s)\bar{\alpha}Y - \frac{1}{2\gamma_s}\ln\bar{\alpha}Y, \tag{324}$$

whose leading term is the one of $\ln[Q_s^2(Y)/Q_A^2]$ in Eq. (316), we may rewrite the BFKL amplitude as

$$T_{\text{BFKL}}(Y, x_{01}) \simeq \frac{f(\gamma_s)}{\sqrt{2\pi\chi''(\gamma_s)}}\left(\frac{x_{01}^2 Q_{s(\text{lin})}^2(Y)}{4}\right)^{\gamma_s} \exp\left(-\frac{\ln^2\left\{4/[x_{01}^2 Q_{s(\text{lin})}^2(Y)]\right\}}{2\chi''(\gamma_s)\bar{\alpha}Y}\right), \tag{325}$$

up to subleading terms at large $Y$.

### 6.2.2 Full BK: Effect of the non-linearity

In order to take properly account of the non-linearity in the BK equation, we may proceed as for the FKPP equation, trading the non-linear term for a moving absorptive boundary on the linearized equation.

At this point, instead of redoing the whole calculation, we shall just take over the results obtained for the FKPP equation listed in Eq. (293), after identification of the variables using Eqs. (320),(321). The solution to the BK equation eventually reads, for $x_{01}Q_s(Y)/2 \ll 1$

$$T(Y, x_{01}) = \text{const} \times \ln\frac{4}{x_{01}^2 Q_s^2(Y)}\left(\frac{x_{01}^2 Q_s^2(Y)}{4}\right)^{\gamma_s} \exp\left(-\frac{\ln^2\{4/[x_{01}^2 Q_s^2(Y)]\}}{2\chi''(\gamma_s)\bar{\alpha}Y}\right), \tag{326}$$

where the saturation momentum depends on the rapidity as follows:

$$\ln\frac{Q_s^2(Y)}{Q_A^2} = \chi'(\gamma_s)\bar{\alpha}Y - \frac{3}{2\gamma_s}\ln\bar{\alpha}Y + \mathcal{O}\left(1/\sqrt{\bar{\alpha}Y}\right). \tag{327}$$

At very large rapidities, $T$ becomes a function of a single variable, $x_{01}^2 Q_s^2(Y)/4$, which is typical of a traveling wave. The traveling waves are clearly seen in numerical solutions of the BK equation; compare Fig. 16 and Fig. 19. Rescaling $T$ by $\{4/[x_{01}^2 Q_s^2(Y)]\}^{\gamma_s}$ shows the

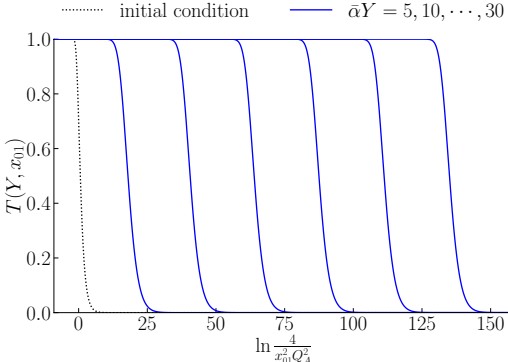

(a) $T$ as a function of $\ln[4/(x_{01}^2 Q_A^2)]$, for different values of $\bar{\alpha}Y$ (increasing from left to right).

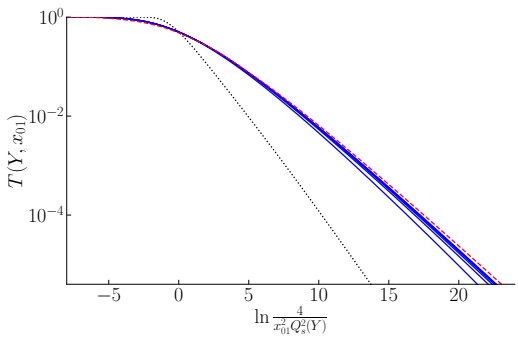

(b) $T$ as a function of $\ln\{4/[x_{01}^2 Q_s^2(Y)]\}$, in order to show to convergence to a traveling wave.

Figure 19: Numerical solution of the BK equation for different values of the rapidity, with the MV model (238) as an initial condition.

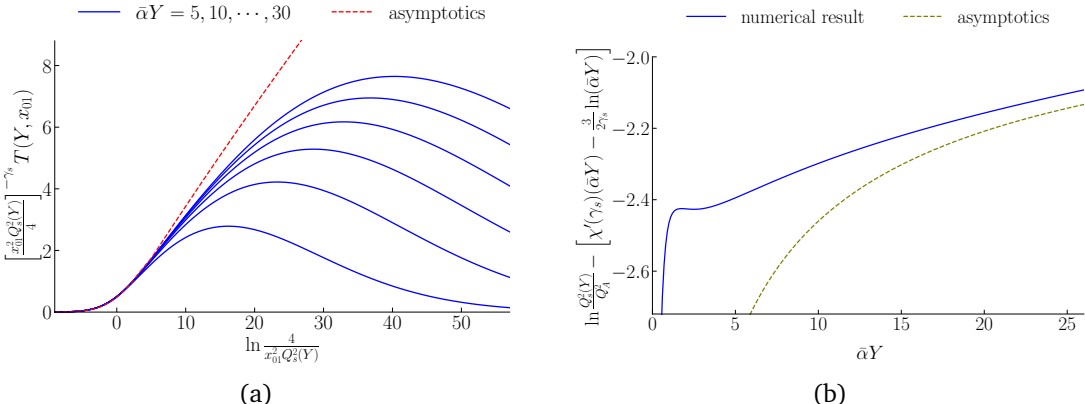

(a)                                          (b)

Figure 20: Equivalent of Fig. 17, but for the BK equation.

convergence to a linear function of the scaling variable, which is a characteristic feature of solutions to FKPP-like equations; see Fig. 20a (see also Ref. [79]). The numerical evaluation of the front position clearly matches very well the analytical formula (327); see Fig. 20b. Subleading corrections, of order $\mathcal{O}(1/\sqrt{\bar{\alpha}Y})$, may also be considered: we refer the reader to [80] for detailed studies.

Finally, a historical comment. Our approach has been to show that general methods developed in mathematics to solve the FKPP equation can quite naturally be taken over to the BK equation. The reason is that in the dipole model, the interaction state of an onium results from a binary Markov branching process, similar to the BBM, and the $S$-matrix element of interest amounts to an observable on this classical process, similar to the statistics of extremes in the BBM.

However, it must be acknowledged that a version of the calculation presented in this section was carried out independently of these considerations in the particle physics community, before the tight connection between the BK and FKPP equations was unveiled. In Ref. [81], the BK equation was approximated by the BFKL equation for dipole sizes smaller than the inverse saturation scale, and following the lines of constant amplitude, the correct leading rapidity-dependence of the latter (namely the linear term in the logarithm of the saturation scale) was found. In Ref. [82], the non-trivial effect of the nonlinearity was understood. Remarkably

enough, the calculation done in there consisted precisely in trading the latter for a moving absorptive boundary condition. It was only a posteriori that an approximate mapping between the two equations was exhibited [83], which, in turn, allowed to clarify the above-mentioned approaches, and more generally, the physics of the BK equation. Let us discuss this mapping.

### 6.3   Mapping BK to FKPP in momentum space

So far, we have discussed the BK equation in parallel to the FKPP equation, applying the methods known for the latter to the former. We may actually map the former to the latter, in a way that becomes exact for some asymptotic values of the parameters.

Following [16], we introduce the transformation

$$\widetilde{T}(Y,k) = \int \frac{d^2 x_{01}^\perp}{2\pi x_{01}^2} e^{ik^\perp \cdot x_{01}^\perp} T(Y, x_{01}),\tag{328}$$

where $k = \left|k^\perp\right|$. This particular map has the advantage of deconvoluting the non-linear term in the BK equation (300). Indeed, a straightforward calculation shows that

$$\int \frac{d^2 x_{01}^\perp}{2\pi x_{01}^2} e^{ik^\perp \cdot x_{01}^\perp} \int \frac{d^2 x_2^\perp}{2\pi} \frac{x_{01}^2}{x_{02}^2 x_{21}^2} T(Y, x_{02}) T(Y, x_{21}) = \left[\widetilde{T}(Y,k)\right]^2.\tag{329}$$

As for the linear part in the right-hand side of Eq. (300), we may write it as an operator acting on $\widetilde{T}$ constructed from the eigenvalues of the BFKL kernel. Starting from Eq. (308) and recognizing that $x_{01}^\perp$ is conjugate to $k^\perp$ through the Fourier transform, the BFKL kernel operator in $k^\perp$-space reads $\bar{\alpha}\chi(-\partial_{\ln k^2})$. The BK equation then becomes

$$\partial_Y \widetilde{T}(Y,k) = \bar{\alpha}\left\{\chi\left(-\partial_{\ln k^2}\right)\widetilde{T}(Y,k) - \left[\widetilde{T}(Y,k)\right]^2\right\}.\tag{330}$$

If the kernel were an appropriate second-order differential operator, then this equation would map to the FKPP equation, through possible changes of variables and function redefinition. The kernel is integro-differential, but we can expand it in the form of a differential operator.

Let us pick some $\bar{\gamma} \in (0,1)$ and write the following formal expansion of the kernel, obtained by replacing $\gamma$ by $-\partial_{\ln k^2}$ in Eq. (322):

$$\chi(-\partial_{\ln k^2}) = \chi(\bar{\gamma}) + \chi'(\bar{\gamma})(-\partial_{\ln k^2} - \bar{\gamma}) + \frac{\chi''(\bar{\gamma})}{2}(-\partial_{\ln k^2} - \bar{\gamma})^2 + \cdots.\tag{331}$$

Keeping only terms up to the second order, this is a diffusive approximation. There is one better choice for $\bar{\gamma}$ when one is interested in the transition region to saturation: $\bar{\gamma} = \gamma_s$. The expanded BK equation then writes

$$\partial_Y \widetilde{T}(Y,k) = \bar{\alpha}\left\{-\chi'(\gamma_s)\partial_{\ln k^2}\widetilde{T}(Y,k) + \frac{\chi''(\gamma_s)}{2}(\partial_{\ln k^2} + \gamma_s)^2 \widetilde{T}(Y,k) - \left[\widetilde{T}(Y,k)\right]^2\right\}.\tag{332}$$

We define the new variables

$$x \equiv \frac{\gamma_s}{\sqrt{2}}\left\{\ln k^2 + \left[\gamma_s\chi''(\gamma_s) - \chi'(\gamma_s)\right]\bar{\alpha}Y\right\}, \qquad t \equiv \frac{\gamma_s^2\chi''(\gamma_s)}{2}\bar{\alpha}Y.\tag{333}$$

The function $u \equiv \frac{2}{\gamma_s^2\chi''(\gamma_s)}\widetilde{T}$ then satisfies the FKPP equation (266) with $D = \frac{1}{2}$ and $\lambda = 1$:

$$\partial_t u(t,x) = \frac{1}{2}\partial_x^2 u(t,x) + u(t,x) - [u(t,x)]^2.\tag{334}$$

Taking over the solution (293) and changing variables according to Eq. (333), one finds for $\widetilde{T}$

$$\widetilde{T}(Y,k) = \text{const} \times \ln \frac{k^2}{Q_s^2(Y)} \left( \frac{k^2}{Q_s^2(Y)} \right)^{-\gamma_s} \exp \left( -\frac{\ln^2 \left[ k^2/Q_s^2(Y) \right]}{2\chi''(\gamma_s)\bar{\alpha}Y} \right), \qquad (335)$$

with the saturation momentum $Q_s(Y)$ given by Eq. (327), up to constant irrelevant numerical terms in $\ln[Q_s^2(Y)/Q_A^2]$. This formula is valid for $k \gg Q_s(Y)$. As expected from the universality of the solutions to traveling wave equations (see the discussion at the end of Sec. 5.2.2), $\widetilde{T}$ has exactly the same form as $T$, up to the substitution $k \longleftrightarrow 2/x_{01}$.

## 6.4 Consequences for physical observables: Total cross section in deep-inelastic scattering

This section aims at establishing a link between our theoretical calculations and collider observables. We choose to focus on deep-inelastic scattering (DIS), namely, the process of scattering of an electron off an atomic nucleus at very high energy. The electron does not interact directly with the components of the nucleus, but through an electroweak gauge boson. We shall assume that the electron interacts through a virtual photon $\gamma^*$, and we will consider the latter as the initial state.[17] This channel turns out to be the dominant one in the kinematical regime of interest, namely that of high center-of-mass energies and moderately large momentum transfers.

We shall first relate the dipole $S$-matrix element we have computed to the DIS cross section (Sec. 6.4.1). We shall then examine briefly one of the main measurable consequence of the form of the $S$-matrix element we have found, namely a phenomenon known under the name "geometric scaling" (Sec. 6.4.2).

We take as an initial state a normalized wave packet constructed from the superposition of fixed-momentum photon states of polarization $\lambda_\bullet$:

$$|\phi_\bullet; \lambda_\bullet\rangle \equiv \int_{\vec{q}_\bullet} \phi_\bullet(\vec{q}_\bullet) |\gamma^*; \vec{q}_\bullet, \lambda_\bullet\rangle, \qquad \text{with} \qquad \int_{\vec{q}_\bullet} |\phi_\bullet(\vec{q}_\bullet)|^2 = 1, \qquad (336)$$

keeping in mind that we will eventually take the limit of a perfectly monochromatic initial state. We introduce the impact parameter $b_\bullet^\perp$ by translating the wave packet:

$$\left| \phi_\bullet; b_\bullet^\perp, \lambda_\bullet \right\rangle \equiv e^{-i b_\bullet^\perp \cdot P^\perp} |\phi_\bullet; \lambda_\bullet\rangle = \int_{\vec{q}_\bullet} \phi_\bullet(\vec{q}_\bullet) e^{-i b_\bullet^\perp \cdot q_\bullet^\perp} |\gamma^*; \vec{q}_\bullet, \lambda_\bullet\rangle, \qquad (337)$$

where $P^\perp$ stands for the representation on the space of states of the infinitesimal generator of translations in the transverse plane.

### 6.4.1 From the forward elastic $S$-matrix element to deep-inelastic scattering cross sections

The total cross section $\sigma^{\gamma^*(\lambda_\bullet)A}$ for the interaction of a virtual photon of polarization $\lambda_\bullet$ off a nucleus can be defined as the transition probability between the virtual photon and any final state, integrated over the impact parameter of the initial wave packet in order to account for the uncertainty in the precise position of the incident particles in the transverse plane.[18]

---

[17]Note that a virtual photon is a priori not a proper asymptotic state, being off-shell. But trading it for an initial state leads to the same result as if one starts with the electron; see e.g. Ref. [84].

[18]See Ref. [85] (Chap. 3) for a detailed discussion of the scattering cross section of a particle off a scattering center in non-relativistic quantum mechanics, and Ref. [86] (Sec. 4.5) for its relativistic extension in a modern formulation.

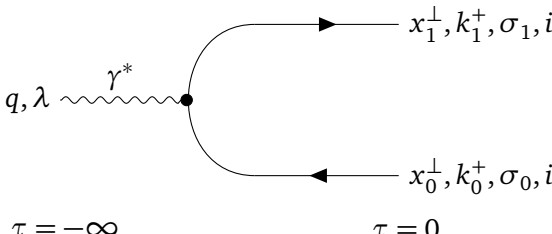

Figure 21: Diagram contribution to the wave function $\widetilde{\psi}_{\bar{q}q,\text{i}(\phi_{\bullet})}$ and $\widetilde{\varphi}$ of the virtual photon in quark-antiquark pairs.

Introducing a complete set of normalized final states $\{|\mathbb{f}_0\rangle\}$, it reads

$$\sigma^{\gamma^*(\lambda_{\bullet})A} = \int d^2 b_{\bullet}^{\perp} \sum_{|\mathbb{f}_0\rangle} \left| \langle \mathbb{f}_0 | T | \phi_{\bullet}; b_{\bullet}^{\perp}, \lambda_{\bullet} \rangle \right|^2 = \int d^2 b_{\bullet}^{\perp} \langle \phi_{\bullet}; b_{\bullet}^{\perp}, \lambda_{\bullet} | T^{\dagger} T | \phi_{\bullet}; b_{\bullet}^{\perp}, \lambda_{\bullet} \rangle . \quad (338)$$

The unitarity of the $S$-matrix $S^{\dagger} S = \mathbb{I}$ implies the operator identity $T^{\dagger} T = 2(1 - \operatorname{Re} S)$, from which the optical theorem follows. Inserting this identity into the previous equation, we arrive at a relation between the forward elastic $S$-matrix element and the total cross section:

$$\sigma^{\gamma^*(\lambda_{\bullet})A} = 2 \int d^2 b_{\bullet}^{\perp} \left[ 1 - \operatorname{Re} S_{\phi_{\bullet}}^{(\lambda_{\bullet})}(b_{\bullet}^{\perp}) \right] . \quad (339)$$

We shall now compute the $S$-matrix element appearing in this formula using Eq. (111). To this aim, let us first introduce the Fock expansion of the dressed state of the virtual photon up to the lowest-order partonic states that couple to gluons, namely quark-antiquark pairs:

$$\left| \text{i}(\phi_{\bullet}; b_{\bullet}^{\perp}, \lambda_{\bullet}) \right\rangle = \sqrt{Z_{\phi_{\bullet}}^{(\lambda_{\bullet})}(b_{\bullet}^{\perp})} \left| \phi_{\bullet}; b_{\bullet}^{\perp}, \lambda_{\bullet} \right\rangle + \sum_{q; \sigma_0, \sigma_1} \int_{\substack{(x_0^{\perp}, k_0^+) \\ (x_1^{\perp}, k_1^+)}} \widetilde{\psi}_{\bar{q}q,\text{i}(\phi_{\bullet})}^{\sigma_0; \sigma_1; \lambda_{\bullet}}(x_0^{\perp}, k_0^+; x_1^{\perp}, k_1^+; b_{\bullet}^{\perp})$$

$$\times \sum_{i=1}^{N} \left| \bar{q}; x_0^{\perp}, k_0^+, \sigma_0, i \right\rangle \otimes \left| q; x_1^{\perp}, k_1^+, \sigma_1, i \right\rangle + \cdots . \quad (340)$$

Each $\bar{q}q$ pair is understood to come in a flavor $q$ (of course not to be confused with the four-momentum of the virtual photon), and the sum over $q$ runs over all active flavors. The lowest-order non-trivial diagram contributing to the wave function $\widetilde{\psi}_{\bar{q}q,\text{i}(\phi_{\bullet})}$ is shown in Fig. 21.

The bare virtual photon does not couple to the vector potential $\mathcal{A}$: the Wilson line associated to its trajectory through the external field is trivial. Equation (111) then reads, taking into account the two states in Eq. (340),

$$S_{\phi_{\bullet}}^{(\lambda_{\bullet})}(b_{\bullet}^{\perp}) = Z_{\phi_{\bullet}}^{(\lambda_{\bullet})}(b_{\bullet}^{\perp}) + \sum_{q; \sigma_0, \sigma_1} \int_{\substack{(x_0^{\perp}, k_0^+) \\ (x_1^{\perp}, k_1^+)}} \left| \widetilde{\psi}_{\bar{q}q,\text{i}(\phi_{\bullet})}^{\sigma_0; \sigma_1; \lambda_{\bullet}}(x_0^{\perp}, k_0^+; x_1^{\perp}, k_1^+; b_{\bullet}^{\perp}) \right|^2 N S_{\bullet}(x_0^{\perp}, x_1^{\perp}) . \quad (341)$$

The renormalization constant $Z_{\phi_{\bullet}}$ can be obtained from unitarity [see Eq. (19)], using the fact that the states in Eq. (340) are all normalized to unity. At order $\alpha_{\text{em}} \alpha_s^0$, it is enough to notice that if $S_{\bullet}$ is set to 1 in the right-hand side, meaning that we switch off the external field, then the left-hand side of this equation must also equate unity. Inserting the value of $Z_{\phi_{\bullet}}$ thus determined back into Eq. (341), we get

$$1 - S_{\phi_{\bullet}}^{(\lambda_{\bullet})}(b_{\bullet}^{\perp}) = \sum_{q; \sigma_0, \sigma_1} N \int_{\substack{(x_0^{\perp}, k_0^+) \\ (x_1^{\perp}, k_1^+)}} \left| \widetilde{\psi}_{\bar{q}q,\text{i}(\phi_{\bullet})}^{\sigma_0; \sigma_1; \lambda_{\bullet}}(x_0^{\perp}, k_0^+; x_1^{\perp}, k_1^+; b_{\bullet}^{\perp}) \right|^2 \left[ 1 - S_{\bullet}(x_0^{\perp}, x_1^{\perp}) \right] . \quad (342)$$

It is then enough to plug this equation into Eq. (339) to get an expression for the DIS cross section in terms of the dipole forward elastic $S$-matrix element and of the light cone wave function of the virtual photon:

$$\sigma^{\gamma^*(\lambda_\bullet)A} = \sum_{q;\sigma_0,\sigma_1} N \int_{\substack{(x_0^\perp,k_0^+)\\(x_1^\perp,k_1^+)}} \left( \int d^2 b_\bullet^\perp \left| \widetilde{\psi}_{\bar{q}q,\mathtt{i}(\phi_\bullet)}^{\sigma_0;\sigma_1;\lambda_\bullet}(x_0^\perp,k_0^+;x_1^\perp,k_1^+;b_\bullet^\perp) \right|^2 \right) \times 2\left[1 - S_\bullet(x_0^\perp,x_1^\perp)\right]. \quad (343)$$

In order to simplify this expression, we first factorize the Dirac distributions that enforce momentum conservation from the momentum-space wave function, defining new functions $\varphi_q$ as follows:

$$\psi_{\bar{q}q,\mathtt{i}(\gamma^*)}^{\sigma_0;\sigma_1;\lambda_\bullet}(\vec{k}_0,\vec{k}_1;\vec{q}_\bullet,b_\bullet^\perp) = \left[(2\pi)^3 2q_\bullet^+ \delta^3(\vec{k}_0 + \vec{k}_1 - \vec{q}_\bullet)\right] \varphi_q^{\sigma_0;\sigma_1;\lambda_\bullet}(\vec{k}_0,\vec{k}_1;\vec{q}_\bullet) e^{-ib_\bullet^\perp \cdot q_\bullet^\perp}. \quad (344)$$

Convoluting with $\phi_\bullet$ and going to the mixed representation through the Fourier transformation (110), a straightforward calculation leads to

$$\widetilde{\psi}_{\bar{q}q,\mathtt{i}(\phi_\bullet)}^{\sigma_0;\sigma_1;\lambda_\bullet}(x_0^\perp,k_0^+;x_1^\perp,k_1^+;b_\bullet^\perp) = \int \frac{d^2 q_\bullet^\perp}{(2\pi)^2} \phi_\bullet(q_\bullet^\perp, k_0^+ + k_1^+)$$
$$\times e^{-i[b_\bullet^\perp - (x_0^\perp + x_1^\perp)/2]\cdot q_\bullet^\perp} \widetilde{\varphi}_q^{\sigma_0;\sigma_1;\lambda_\bullet}(x_{10}^\perp,k_1^+;\vec{q}_\bullet), \quad (345)$$

where $\widetilde{\varphi}_q$ is the following Fourier transform of $\varphi_q$

$$\widetilde{\varphi}_q^{\sigma_0;\sigma_1;\lambda_\bullet}(x_{10}^\perp,k_1^+;\vec{q}_\bullet) = \int \frac{d^2 k_1^\perp}{(2\pi)^2} e^{ix_{10}^\perp \cdot (k_1^\perp - q_\bullet^\perp/2)} \varphi_q^{\sigma_0;\sigma_1;\lambda_\bullet}(\vec{q}_\bullet - \vec{k}_1,\vec{k}_1;\vec{q}_\bullet). \quad (346)$$

Taking the modulus squared and integrating over the impact parameter,

$$\int d^2 b_\bullet^\perp \left| \widetilde{\psi}_{\bar{q}q,\mathtt{i}(\phi_\bullet)}^{\sigma_0;\sigma_1;\lambda_\bullet}(x_0^\perp,k_0^+;x_1^\perp,k_1^+;b_\bullet^\perp) \right|^2$$
$$= \int \frac{dq_\bullet^\perp}{(2\pi)^2} \left| \phi_\bullet(q_\bullet^\perp,k_0^+ + k_1^+) \right|^2 \left| \widetilde{\varphi}_q^{\sigma_0;\sigma_1;\lambda_\bullet}(x_{10}^\perp,k_1^+;q_\bullet^\perp,k_0^+ + k_1^+) \right|^2. \quad (347)$$

We then insert this result into Eq. (343), and take the limit of a monochromatic wave packet, with the choice $q^\perp = 0^\perp$. To this aim, we let $|\phi_\bullet|^2$ tend to an appropriately-normalized Dirac distribution:

$$|\phi_\bullet(\vec{q}_\bullet)|^2 \to (2\pi)^3 \delta^2(q_\bullet^\perp) 2q^+ \delta(q_\bullet^+ - q^+). \quad (348)$$

It is also convenient to change variables as follows:

$$(x_0^\perp,x_1^\perp) \to \left( x_{10}^\perp, b^\perp \equiv \frac{x_1^\perp + x_0^\perp}{2} \right), \qquad k_1^+ \to z \equiv \frac{k_1^+}{q^+}. \quad (349)$$

Simple algebra eventually leads us to the following expression of the cross section:

$$\boxed{\begin{aligned} \sigma^{\gamma^*(\lambda_\bullet)A} = N \int_0^1 \frac{dz}{4\pi z(1-z)} \int d^2 x_{01}^\perp \sum_{q;\sigma_0,\sigma_1} \left| \widetilde{\varphi}_q^{\sigma_0;\sigma_1;\lambda_\bullet}(x_{10}^\perp,z;q^+) \right|^2 \\ \times \int d^2 b^\perp 2\left[1 - S_\bullet(b^\perp - x_{10}^\perp/2, b^\perp + x_{10}^\perp/2)\right]. \end{aligned}} \quad (350)$$

It is now trivial to arrive at the cross sections that are effectively measured, namely of a transverse initial photon averaged over its two possible states and of a scalar photon, and the unpolarized cross section. The relationship of these observables to $\sigma^{\gamma^*(\lambda_\bullet)A}$ reads

$$\sigma_T^{\gamma^*A} = \frac{1}{2} \sum_{\lambda_\bullet = \pm 1} \sigma^{\gamma^*(\lambda_\bullet)A}, \qquad \sigma_L^{\gamma^*A} = \sigma^{\gamma^*(\lambda_\bullet=0)A}, \qquad \sigma^{\gamma^*A} = \sigma_T^{\gamma^*A} + \sigma_L^{\gamma^*A}. \tag{351}$$

The wave functions of the virtual photon $\widetilde{\varphi}_q$ in the particular reference frame we have chosen are computed in Appendix B.

These formulas relate the DIS cross sections to the forward elastic amplitude for the scattering of a color dipole [87]. We have established it for the bare dipole amplitude, which should be a good approximation for not too large rapidities $Y$. In that regime, the dipole amplitude may be modeled e.g. by the McLerran-Venugopalan model (238). For larger rapidities, higher quantum fluctuations will have to be included, resulting in a dependence upon the rapidity. This dependence may be modeled (see e.g. Refs. [88–91]). But quantum fluctuations can also be accounted for from first principles using the color dipole model. It is actually enough to replace $T_\bullet = 1 - S_\bullet$ in Eq. (350) by $T$ given by the solution (326) to the BK equation. Phenomenological studies show a good agreement with the data on electron-proton scattering (see e.g. Refs. [78, 92, 93]).

### 6.4.2 Geometric scaling

Let us present the most striking phenomenological consequence of the traveling wave solution to the BK equation. To simplify the discussion, one may integrate over the impact parameter assuming that the nucleus is a homogeneous uniform object of cross section $\pi R^2$. We get

$$\sigma^{\gamma^*(\lambda_\bullet)A} = \int d^2 x_{01}^\perp \left( N \int_0^1 \frac{dz}{4\pi z(1-z)} \sum_{q;\sigma_0,\sigma_1} \left| \widetilde{\varphi}_q^{\sigma_0;\sigma_1;\lambda_\bullet}(x_{10}^\perp, z; q^+) \right|^2 \right) 2\pi R^2 [1 - S(Y, x_{10})]. \tag{352}$$

Analyzing the probability density of the dipole size appearing in parentheses in the previous equation whose explicit expressions can be found in Appendix B [see Eqs. (B.9) and (B.12)], one may show that the scaling property of the solutions to the BK equation in the $x_{01}^2 Q_s^2(Y)$ variable goes over to the electron-nucleus total cross section $\sigma^{\gamma^*A}$ in the $Q^2/Q_s^2(Y)$ variable (see e.g. [88]). The latter becomes a function of a single variable at large rapidities:

$$\boxed{\sigma^{\gamma^*A}(Y,Q) \xrightarrow[\text{large } \bar{\alpha}Y]{} \text{ function of } \frac{Q^2}{Q_s^2(Y)}.} \tag{353}$$

This feature is called "geometric scaling". It was first postulated in a simple saturation model [88], and shortly after, it was discovered in the deep-inelastic data [94]; see Fig. 22. This actually happened before geometric scaling was recognized to follow from the mathematics of the solutions to the BK equation [83]: on the contrary, it was this experimental observation that motivated the more formal studies reported in these lectures.

Note however that the fact that the scaling looks so good in the data might appear a bit surprising, for a couple of reasons. First, the data were taken from electron-proton scattering, while one important assumption of the BK equation is that the target be a large nucleus. Second, the traveling wave property of the solution to FKPP-like equations such as BK is only expected to be good for large values of the evolution parameter. In QCD, the latter is $\bar{\alpha}Y$, which is definitely not very large in any realistic experiment! Finally, an infinite number of colors is also assumed explicitly in the derivation of the BK equation, and, as we have seen, this assumption plays a crucial role for the very existence of the whole probabilistic picture.

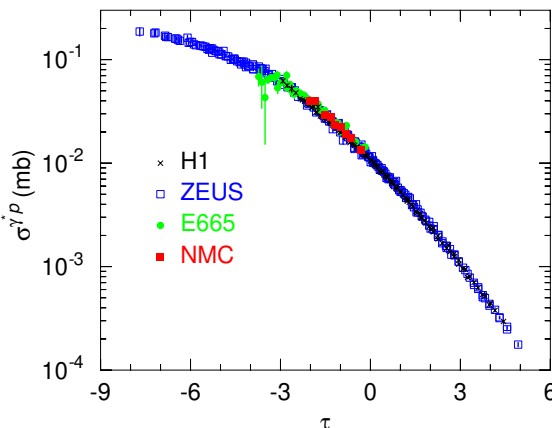

Figure 22: Virtual photon-proton cross section extracted from deep-inelastic electron-proton data, plotted as a function of $\tau = \ln Q^2/Q_s^2(Y)$. Geometric scaling [94] is clearly visible. [Plot from Ref. [95]. In there, a parametrization of $Q_s(Y)$ that incorporates the known properties of the latter was used].

$N = 3$ is far from infinite. But it turns out that the subasymptotic corrections are expected of order $1/N^2$, which amounts to a reasonable $\mathcal{O}(10\%)$ systematic error induced by the limit of infinite number of colors.[19]

# 7 Conclusions

## 7.1 Summary

The main goal of these lectures was to show, starting from the basics, how the forward $S$-matrix element for the scattering of an onium off a classical field at high energy and in the large-number-of-color limit can be understood as an observable on a classical branching process, and how one may take advantage of this fact to better understand high-energy onium or electron-nucleus scattering.

We derived the infinite-rapidity limit of the amplitudes for the scattering of a quantum particle off a classical field (Sec. 2), arriving at Eq. (111). The latter consisted in a convolution of light cone wave functions (describing the partonic Fock states of the quantum particles) and Wilson lines (accounting for the interaction of the partons with the external field).

Next, we specialized to forward elastic scattering of onia. We first derived a formula, that turned out to be very simple, for the corresponding $S$-matrix element at lowest order in perturbation theory and keeping only the leading contribution in the high-energy limit (Sec. 3); see Eq. (199). The next order in perturbation theory became also very simple in the infinite-number-of-color limit (Sec. 4): see Eq. (227). We have seen that, within the framework of these approximations, the partonic Fock states of an onium could be understood as collections of color dipoles, and the interaction with the external field as the independent scatterings of these dipoles. A pattern for constructing interaction states as the result of independent stochastic dipole branchings emerged. This enabled the perturbation theory for the $S$-matrix element to be iterated at all orders: this is how we argued the Balitsky-Kovchegov equation (234).

We then stepped back, digressing into one of the simplest stochastic branching process, namely the branching Brownian motion (Sec. 5). We showed that the Fisher-Kolmogorov-

---

[19]Another mystery however is that empirically, the finite-$N$ corrections to QCD high-energy evolution turn out to be rather on the order of 1%; see e.g. the numerical calculations in Ref. [96].

Petrovsky-Piscounov equation (260) that governs a large class of observables on the BBM, and in particular, the statistics of extremes, belongs to the same universality class as the BK equation. This eventually enabled us to place the BK equation in a more general context and to better understand the essence of the physics it describes, and its solutions (Sec. 6). We also briefly mentioned some phenomenological applications.

## 7.2 To go beyond

**Other nuclear observables.** The relationship between the dipole model and stochastic processes has more recently enabled to understand and calculate other observables beyond the total onium-nucleus cross-section. Indeed, it has been shown that the hard diffraction process in deeply inelastic scattering can be understood in terms of genealogies of the parton branching process [97, 98]. This made it possible to find an asymptotic analytical solution to the Kovchegov-Levin equation [99] that governs the diffractive cross-section [100, 101].

Even more recently, the relevance of the FKPP equation to high-energy physics was pointed out in a quite different context. Indeed, it was shown that the equation governing the quantum corrections on transverse momentum broadening of a fast parton passing through dense QCD matter admitted traveling wave solutions [102, 103].

**Beyond nuclear targets.** The connection between onium scattering and stochastic processes has made it possible to understand what might change when the target is not a nucleus which can be modeled by a classical field, but a "dilute" hadron, for example another onium. In this case, a crucial assumption for the derivation of the BK equation is no longer justified: that dipoles scatter independently with the target. This assumption implies that the only source of correlations is the non-trivial genealogy of the dipoles present in the interaction state as a result of the quantum evolution.

After a ground-breaking calculation of dipole-dipole scattering amplitudes based on unitarity arguments only [104], it was conjectured that this particle-physics process is similar to reaction-diffusion processes, or to branching Brownian motion supplemented with a selection mechanism (the so-called "$N$-BBM" process) [105]. Relevant equations for observables on such processes belong to the universality class of the *stochastic* FKPP equation. Results obtained for the latter [106, 107] can be transposed to QCD, taking advantage of the conjectured universality.

The study of the $N$-BBM has become a very active field of research in statistical physics and mathematics (see [25] and references therein), with many interdisciplinary applications. On the QCD side, an explicit stochastic extension of the BK equation has not been exhibited to date, despite very interesting attempts (see e.g. [108–111]). Thus, the theoretical understanding of onium-onium scattering at very high energy, as opposed to the scattering of an onium off a large nucleus that may be modeled with the help of a classical external field, remains an outstanding challenge.

# Acknowledgments

SM warmly thanks François Arleo and Stéphane Peigné for the invitation to lecture at the QCD Master Class 2021 (Saint-Jacut-de-la-Mer, France). These notes represent an expanded version of the material presented there. He also congratulates them for the perfect organization, and thanks the participants for their interest and for their multiple questions.

He thanks François Gelis for helpful discussions and for his feedback on the manuscript, and Anne-Sophie de Suzzoni for her advice on a mathematical point. Above all, he is deeply

grateful to Al Mueller for his teaching and collaborative research over the past two decades, from which these lecture notes are heavily inspired.

The Feynman diagrams were produced using the TikZ-Feynman package [112].

**Funding information** This work was partly supported by the Agence Nationale de la Recherche (France) under grants ANR-16-CE31-0019 (DenseQCD@LHC) and ANR-18-CE31-0024 (COLDLOSS). ADL was supported by the Academy of Finland and the Centre of Excellence in Quark Matter (project 346324).

# A   Some useful formulas and calculation techniques

## A.1   Wilson lines: A few properties

Wilson lines naturally appear when we take the high-energy limit of the scattering operator applied to a partonic state, see Sec. 2.3.2, in particular Eq. (90). We show here some of their properties useful for our purposes. A more complete discussion can be found e.g. in Ref. [113, 114].

Let us consider a slightly more general Wilson line than the one introduced in the body of the paper, restricting the integration path to the light-like line segment connecting $(x_i^+, x^\perp)$ to $(x_f^+, x^\perp)$:

$$W^R_{[x_f^+, x_i^+]}(x^\perp) \equiv P \exp\left(-ig T_R^a \int_{x_i^+}^{x_f^+} dx^+ \mathcal{A}_a^-(x^+, 0, x^\perp)\right) \tag{A.1}$$

(We could of course consider an even more general path, but it would be of no use to our discussion). Matrix elements of the path-ordered exponential are defined as[20]

$$\left[W^R_{[x_f^+, x_i^+]}(x^\perp)\right]_{i'i} \equiv \sum_{n=0}^{\infty} (-ig)^n T_{R\,i'i_{n-1}}^{a_n} T_{R\,i_{n-1}i_{n-2}}^{a_{n-1}} \cdots T_{R\,i_1 i}^{a_1} \int_{x_i^+}^{x_f^+} dx_1^+ \mathcal{A}_{a_1}^-(x_1^+, 0, x^\perp)$$

$$\times \int_{x_1^+}^{x_f^+} dx_2^+ \mathcal{A}_{a_2}^-(x_2^+, 0, x^\perp) \cdots \int_{x_{n-1}^+}^{x_f^+} dx_n^+ \mathcal{A}_{a_n}^-(x_n^+, 0, x^\perp). \tag{A.2}$$

In each term, the potentials are ordered in light-cone time in such a way that the one evaluated at the earliest time, $x_1^+$, is the first one to act when the Wilson line operator is applied to a ket in the representation $R$; see Fig. 23 for an illustration. The Wilson line used in the body of the text, Eq. (92), is just the limit

$$W^R(x^\perp) = \lim_{x_f^+ \to +\infty} \lim_{x_i^+ \to -\infty} W^R_{[x_f^+, x_i^+]}(x^\perp). \tag{A.3}$$

It will be instrumental for our discussion that Wilson lines solve first-order ordinary differential equations. To establish one of them, we think of $W^R_{[x_f^+, x_i^+]}(x^\perp)$ as a function of $x_i^+$ alone, and take a derivative of Eq. (A.2) with respect to this variable:[21]

$$\frac{d}{dx_i^+} W^R_{[x_f^+, x_i^+]}(x^\perp) = W^R_{[x_f^+, x_i^+]}(x^\perp)\left[ig T_R^a \mathcal{A}_a^-(x_i^+, 0, x^\perp)\right], \quad \text{with} \quad W^R_{[x_f^+, x_i^+ = x_f^+]}(x^\perp) = \mathbb{I}. \tag{A.4}$$

---

[20]The reader will notice an apparent difference between (A.2) and the definition of a time-ordered exponential given in Eq. (9). These are simply alternative ways of writing multiple integrations: there is no difficulty to go from the latter to the former. The one we chose here is more convenient for the discussion we present in this section.

[21]One may write the solution to Eq. (A.4) with the help of the exponential of a vector of the Lie algebra of the representation $R$: this is known as the Magnus expansion [115]. Most of the properties that we shall prove are straightforward starting from such a representation of the solution. The drawback is that the argument of the exponential is a complicated series of nested Lie brackets.

$$\sum_n \int_{x_i^+}^{x_f^+} dx_1^+ \cdots \int_{x_{n-1}^+}^{x_f^+} dx_n^+ \left( \begin{array}{c} \text{diagram} \end{array} \right)$$

Figure 23: Graphical representation of the matrix element $\left[ W^F_{[x_f^+,x_i^+]}(x^\perp) \right]_{i'i}$ of the fundamental Wilson line, see Eq. (A.2).

### A.1.1 Fundamental Wilson lines are SU($N$) matrices

The unitarity of the Wilson lines may be established in an elementary way. From Eq. (A.4), we easily check that

$$\frac{d}{dx_i^+} \left( W^R_{[x_f^+,x_i^+]}(x^\perp) W^{R\dagger}_{[x_f^+,x_i^+]}(x^\perp) \right) = 0. \tag{A.5}$$

Integrating this trivial differential equation with the boundary condition $W^R_{[x_f^+,x_f^+]} W^{R\dagger}_{[x_f^+,x_f^+]} = \mathbb{I}$ yields

$$W^R_{[x_f^+,x_i^+]}(x^\perp) W^{R\dagger}_{[x_f^+,x_i^+]}(x^\perp) = \mathbb{I}, \tag{A.6}$$

namely that $W^R_{[x_f^+,x_i^+]}(x^\perp)$ is a unitary matrix. A posteriori, this is not very surprising, since the differential equation solved by $W^{R\dagger}$ is nothing but a Schrödinger equation with the Hermitian "Hamiltonian" $g T_R^a \mathcal{A}_a^-$, the solution of which is a unitary evolution operator.

We can also prove that the Wilson lines are of unit determinant: $\det W^R = 1$. This is equivalent to showing that $\ln \det W^R = 0$, namely that $\mathrm{Tr} \ln W^R = 0$. We start by computing the derivative of the latter quantity, using again Eq. (A.4):

$$\begin{aligned} \frac{d}{dx_i^+} \mathrm{Tr} \ln W^R_{[x_f^+,x_i^+]}(x^\perp) &= \mathrm{Tr} \left\{ \left[ W^R_{[x_f^+,x_i^+]}(x^\perp) \right]^{-1} \frac{d}{dx_i^+} W^R_{[x_f^+,x_i^+]}(x^\perp) \right\} \\ &= \mathrm{Tr} \left\{ \left[ W^R_{[x_f^+,x_i^+]}(x^\perp) \right]^{-1} W^R_{[x_f^+,x_i^+]}(x^\perp) \left[ i g T_R^a \mathcal{A}_a^-(x_i^+,0,x^\perp) \right] \right\}, \end{aligned} \tag{A.7}$$

which is obviously null, since the infinitesimal generators $T_R^a$ are traceless. Integrating this equation with the initial condition $\mathrm{Tr} \ln W^R_{[x_f^+,x_i^+=x_f^+]}(x^\perp) = 0$, one gets that the trace of the logarithm of these Wilson lines is null, and thus that, indeed, $\det W^R_{[x_f^+,x_i^+]}(x^\perp) = 1$.

The properties we have just proven are true for any $x_i^+$ and $x_f^+$. There is no difficulty in taking the limits $x_i^+ \to -\infty$, $x_f^+ \to +\infty$. Hence the fundamental Wilson lines $W^F$, $W^{\bar{F}}$ belong to the SU($N$) group. As for the adjoint Wilson line $W^A$, since furthermore it is a real matrix, it belongs to the SO($N^2-1$) group.

### A.1.2 Relation between adjoint and fundamental Wilson lines

We now show the instrumental relation (174) between adjoint and fundamental Wilson lines. Again, we first address the Wilson lines restricted to an interval (A.1). We consider the two

sets of $N^2 - 1$ matrices

$$W^F_{[x^+_f, x^+_i]} t^a W^{F\dagger}_{[x^+_f, x^+_i]}, \qquad \text{and} \qquad t^b \left( W^A_{[x^+_f, x^+_i]} \right)^{ba} \tag{A.8}$$

(We dropped the $x^\perp$-dependence of the Wilson lines to simplify the notations). Using Eq. (A.4), we can establish differential equations that they obey. On the one hand,

$$\frac{d}{dx^+_i} \left( W^F_{[x^+_f, x^+_i]} t^a W^{F\dagger}_{[x^+_f, x^+_i]} \right) = \left( W^F_{[x^+_f, x^+_i]} [t^c, t^a] W^{F\dagger}_{[x^+_f, x^+_i]} \right) ig \mathcal{A}^-_c(x^+_i, 0, x^\perp)$$
$$= \left( W^F_{[x^+_f, x^+_i]} t^d W^{F\dagger}_{[x^+_f, x^+_i]} \right) \left[ ig \left( T^c_A \right)^{da} \mathcal{A}^-_c(x^+_i, 0, x^\perp) \right]. \tag{A.9}$$

We have just used the $\mathfrak{su}(N)$ algebra [see Eq. (43)] to go from the first to the second line. On the other hand,

$$\frac{d}{dx^+_i} \left[ t^b \left( W^A_{[x^+_f, x^+_i]} \right)^{ba} \right] = \left[ t^b \left( W^A_{[x^+_f, x^+_i]} \right)^{bd} \right] \left[ ig \left( T^c_A \right)^{da} \mathcal{A}^-_c(x^+_i, 0, x^\perp) \right]. \tag{A.10}$$

We see that the two sets of matrices (A.8) solve exactly the same system of first-order differential equations. It is easy to check that the initial conditions are also the same. Hence these matrices are identical. The relation (174) follows, after the limits $x_i \to -\infty$, $x_f \to +\infty$ have been taken.

### A.1.3 Relation between fundamental and anti-fundamental Wilson lines

According to Eq. (A.2) with $(T^a_R)_{ij}$ substituted by $(-t^a_{ji})$, the matrix elements of the anti-fundamental Wilson line read

$$\left[ W^{\bar{F}}_{[x^+_f, x^+_i]}(x^\perp) \right]_{i'i} = \sum_{n=0}^\infty (ig)^n t^{a_n}_{i_{n-1} i'} t^{a_{n-1}}_{i_{n-2} i_{n-1}} \cdots t^{a_1}_{i i_1} \times \int_{x^+_i}^{x^+_f} dx^+_1 \mathcal{A}^-_{a_1}(x^+_1, 0, x^\perp)$$
$$\times \int_{x^+_1}^{x^+_f} dx^+_2 \mathcal{A}^-_{a_2}(x^+_2, 0, x^\perp) \cdots \int_{x^+_{n-1}}^{x^+_f} dx^+_n \mathcal{A}^-_{a_n}(x^+_n, 0, x^\perp). \tag{A.11}$$

In each term, the product of the matrix elements of the $t^a$'s summed over the intermediate indices is actually the matrix element $(i, i')$ of the product, $t^{a_1} \cdots t^{a_n}$, which is also the Hermitian conjugate of $t^{a_n} \cdots t^{a_1}$ since the $t^a$'s are Hermitian. Since, furthermore, the field $\mathcal{A}$ is real, it is clear from this expression that

$$\left[ W^{\bar{F}}_{[x^+_f, x^+_i]}(x^\perp) \right]_{i'i} = \left[ W^{F\dagger}_{[x^+_f, x^+_i]}(x^\perp) \right]_{ii'}. \tag{A.12}$$

Moreover, we recognize that the insertions of the potential factors are now anti-path-ordered (i.e. the potential evaluated at the earliest time is now on the left). Therefore,

$$\left[ W^F_{[x^+_f, x^+_i]}(x^\perp) \right]^\dagger = \bar{P} \exp \left( +ig t^a \int_{x^+_i}^{x^+_f} dx^+ \mathcal{A}^-_a(x^+, 0, x^\perp) \right) = W^F_{[x^+_i, x^+_f]}(x^\perp), \tag{A.13}$$

where $\bar{P}$ is a notation for anti-ordering.

## A.2 Spinor bilinears

We provide here a few formulas that are useful for the calculation of light cone perturbation theory graphs; see Tab. 1. They can be obtained purely algebraically, using the expressions of $u$, $v$ as a function of $w$, Eq. (114), and the defining properties of the latter, Eq. (60). Let us sketch their derivation, starting by specifying our choice of the spinors $w$'s.

We consider the spin matrix $\frac{i}{2}\gamma^1\gamma^2$ corresponding to a spin measurement along the $z$-axis. It has two eigenvalues (or helicities) $\sigma = \pm\frac{1}{2}$, each associated to a two-dimensional eigensubspace. We denote their orthonormal basis by $\{v_\sigma^{(k)}, \ k = 1, 2\}$:[22]

$$\frac{i}{2}\gamma^1\gamma^2 v_\sigma^{(k)} = \sigma v_\sigma^{(k)}. \tag{A.14}$$

Without loss of generality, one can assign $w_{(\sigma)} \equiv v_\sigma^{(1)}$ for each helicity $\sigma$. The two remaining vectors $v^{(2)}_{\sigma=\pm\frac{1}{2}}$ then form a basis of $\text{Im}(\Lambda_-)$, which is the orthogonal complement of $\text{Im}(\Lambda_+)$.

The outline for the derivation of some relevant Dirac matrix elements presented in Tab. 1 is as follows. To compute $\bar{u}_{(\sigma_2)}(p_2)\gamma^+ u_{(\sigma_1)}(p_1)$, one first notices that it is identical to

$$\sqrt{2}\, u^\dagger_{(\sigma_2)}(p_2)\Lambda_+ u_{(\sigma_1)}(p_1). \tag{A.15}$$

One subsequently uses the fact that $\Lambda_+$ is a Hermitian projector, and one works out the identity

$$\Lambda_+ u_{(\sigma)} = 2^{1/4}\sqrt{p^+}\, w_{(\sigma)}. \tag{A.16}$$

The first entry in Tab. 1 then stems from the orthogonality of the $w$'s, see Eq. (60). The second entry of that Table is obtained from a similar calculation.

As for the next identities, one needs to work out the action of $\gamma^1$ and $\gamma^2$ on the $w$'s. One multiplies the spinor $\gamma^1 w_{(\sigma)}$ left by the spin matrix:

$$\frac{i}{2}\gamma^1\gamma^2\left(\gamma^1 w_{(\sigma)}\right) = \frac{i}{2}\gamma^2 w_{(\sigma)} = -\gamma^1\left(\frac{i}{2}\gamma^1\gamma^2\right)w_{(\sigma)} = -\sigma\left(\gamma^1 w_{(\sigma)}\right). \tag{A.17}$$

This shows that $\gamma^1 w_{(\sigma)}$ is an eigenvector of the spin operator with eigenvalue $-\sigma$. Since it is invariant under the action of the projector $\Lambda_+$, it should be proportional to $w_{(-\sigma)}$,

$$\gamma^1 w_{(\sigma)} = a_{-\sigma} w_{(-\sigma)}. \tag{A.18}$$

One checks easily that the complex coefficients $a$'s satisfy the following properties:

$$|a_\sigma|^2 = 1, \qquad a_\sigma a_{-\sigma} = -1. \tag{A.19}$$

The action of $\gamma^2$ on $w_{(\sigma)}$ can then be determined also from Eq. (A.17). One eventually gets[23]

$$\gamma^k w_{(\sigma)} = a_{-\sigma}\left(\delta^{k1} + 2i\sigma\delta^{k2}\right)w_{(-\sigma)}. \tag{A.20}$$

We see that the action of the transverse gamma matrices $\gamma^i$ on the spinors $w$'s is to flip the helicity of the latter. With the help of this equation, a straightforward calculation leads to the third, fourth, fifth and sixth lines in Tab. 1. The $a$'s eventually disappear from the expression of physical observables. Therefore, in the body of the text, for definiteness, we will set $a_\sigma = -2\sigma$.

---

[22]We could choose the basis of spinors in the set of the eigenstates of another Hermitian operator: it is enough that the latter commutes with $\Lambda_+$.

[23]In, for e.g., Refs. [5, 30, 31, 37, 38], the authors picked specific forms of the spinors $w$'s, leading to specific values of the coefficients $a$'s. For example, $a_\sigma = -2\sigma$ in Refs. [5, 30, 31].

Table 1: Some useful Dirac matrix elements for $u$- and $v$-spinors. We set $a_{-\sigma} = 2\sigma$ in the body of the text. (See also Refs. [5,30], taking however into account the different conventions).

| Matrix element | Expression |
|---|---|
| $\bar{u}_{(\sigma_2)}(p_2)\gamma^+ u_{(\sigma_1)}(p_1)$ | $2\sqrt{p_1^+ p_2^+}\,\delta_{\sigma_1\sigma_2}$ |
| $\bar{v}_{(\sigma_2)}(p_2)\gamma^+ u_{(\sigma_1)}(p_1)$ | $2\sqrt{p_1^+ p_2^+}\,\delta_{\sigma_1,-\sigma_2}$ |
| $\bar{u}_{(\sigma_2)}(p_2)\gamma^- u_{(\sigma_1)}(p_1)$ | $\delta_{\sigma_1\sigma_2}\dfrac{1}{\sqrt{p_1^+ p_2^+}}\left(p_1^i p_2^i - 2i\sigma_1 \epsilon^{ij}p_1^i p_2^j + m^2\right)$ $-\delta_{\sigma_1,-\sigma_2}a_{-\sigma_1}\dfrac{m}{\sqrt{p_1^+ p_2^+}}(p_2^i - p_1^i)(\delta^{i1} + 2i\sigma_1\delta^{i2})$ |
| $\bar{v}_{(\sigma_2)}(p_2)\gamma^- u_{(\sigma_1)}(p_1)$ | $\delta_{\sigma_1,-\sigma_2}\dfrac{1}{\sqrt{p_1^+ p_2^+}}\left(p_1^i p_2^i - 2i\sigma_1 \epsilon^{ij}p_1^i p_2^j - m^2\right)$ $-\delta_{\sigma_1\sigma_2}a_{-\sigma_1}\dfrac{m}{\sqrt{p_1^+ p_2^+}}(p_1^i + p_2^i)(\delta^{i1} + 2i\sigma_1\delta^{i2})$ |
| $\dfrac{\bar{u}_{(\sigma_2)}(p_2)\gamma^i u_{(\sigma_1)}(p_1)}{\sqrt{p_1^+ p_2^+}}$ | $\delta_{\sigma_1\sigma_2}\left(\dfrac{p_1^i + 2i\sigma_1\epsilon^{ij}p_1^j}{p_1^+} + \dfrac{p_2^i - 2i\sigma_1\epsilon^{ij}p_2^j}{p_2^+}\right)$ $-\delta_{\sigma_1,-\sigma_2}a_{-\sigma_1}m\left(\dfrac{p_2^+ - p_1^+}{p_1^+ p_2^+}\right)(\delta^{i1} + 2i\sigma_1\delta^{i2})$ |
| $\dfrac{\bar{v}_{(\sigma_2)}(p_2)\gamma^i u_{(\sigma_1)}(p_1)}{\sqrt{p_1^+ p_2^+}}$ | $\delta_{\sigma_1,-\sigma_2}\left(\dfrac{p_1^i + 2i\sigma_1\epsilon^{ij}p_1^j}{p_1^+} + \dfrac{p_2^i - 2i\sigma_1\epsilon^{ij}p_2^j}{p_2^+}\right)$ $-\delta_{\sigma_1\sigma_2}a_{-\sigma_1}m\left(\dfrac{p_2^+ + p_1^+}{p_1^+ p_2^+}\right)(\delta^{i1} + 2i\sigma_1\delta^{i2})$ |
| $\bar{v}_{(\sigma_2)}(p_2)\gamma^\mu v_{(\sigma_1)}(p_1)$ | $\bar{u}_{(\sigma_1)}(p_1)\gamma^\mu u_{(\sigma_2)}(p_2)$ |
| $\bar{u}_{(\sigma_1)}(p_1)\gamma^\mu v_{(\sigma_2)}(p_2)$ | $\left[\bar{v}_{(\sigma_2)}(p_2)\gamma^\mu u_{(\sigma_1)}(p_1)\right]^*$ |

Once we have arrived at this point, the next identity in Tab. 1 is easily shown by noticing that the relation between $v_{(\sigma)}$ and $w_{(-\sigma)}$ is the same as the relation between $u_{(\sigma)}$ and $w_{(\sigma)}$, up to the change of sign $m \to -m$; see Eq. (114). One may then deduce explicit expressions for $\bar{v}_{(\sigma_1)}(p_1)\gamma^\mu v_{(\sigma_2)}(p_2)$ from the expressions of $\bar{u}_{(\sigma_2)}(p_2)\gamma^\mu u_{(\sigma_1)}(p_1)$ found in Tab. 1 through the substitutions $m \to -m$, $p_1 \leftrightarrow p_2$ and $\sigma_1 \leftrightarrow -\sigma_2$, but the latter only in the coefficients of the Kronecker $\delta$'s. We check that the relevant formulas in the second column of the table are invariant under these simultaneous substitutions.

The last identity in the Table is trivial.

## A.3 Graphical method for handling the color algebra

In the body of the text, we simplify SU($N$) color factors using graphical techniques. In this context, Feynman graphs are called "birdtracks" [46]. In this Appendix, we provide the basic rules of the correspondence between group-theoretical objects and graphs, as well as some of the mathematical background. We refer the reader to the original texts [46,116] and/or to the excellent recent lectures [117,118] for more details and other, more sophisticated, applications of birdtrack techniques.

To Feynman graphs in a gauge theory are associated color factors, which, mathematically speaking, consist in linear maps between representation spaces of the gauge group SU($N$) to which the states of the initial and final particles belong. If a graph possesses $n_i$ (resp. $n_f$) incoming (resp. outgoing) quarks, $\bar{n}_i$ (resp. $n_f$) incoming (resp. outgoing) antiquarks and $m_i$ (resp. $m_f$) incoming (resp. outgoing) gluons, it can be thought of as a particular linear map between the following tensor product spaces that represent the Hilbert spaces restricted to the color of the initial state (left) and final state (right) of a reaction:

$$F^{\otimes n_i} \otimes \bar{F}^{\otimes \bar{n}_i} \otimes A^{\otimes m_i} \to F^{\otimes n_f} \otimes \bar{F}^{\otimes \bar{n}_f} \otimes A^{\otimes m_f} \,. \tag{A.21}$$

$F$ and $\bar{F}$ stand for the modules of the fundamental and anti-fundamental representations, which are $N$-dimensional complex vector spaces. $A$ stands for the module of the adjoint representation, a $N^2-1$-dimensional real vector space. Alternatively, the same graph, after rotation of all external legs to the initial state, can be interpreted as an invariant tensor under SU($N$) transformations that belongs to the representation space

$$F^{\otimes(n_i+\bar{n}_f)} \otimes \bar{F}^{\otimes(\bar{n}_i+n_f)} \otimes A^{\otimes(m_i+m_f)} \,. \tag{A.22}$$

Tensors are fully characterized by their components on a basis. In the body of the paper, we have not assigned a particular meaning to the position (up or down) of these indices. In this Appendix, distinguishing the components on $F$ and on $\bar{F}$ will add value. Therefore, we will denote by an upper index each component relative to the basis of $F$ in the final state or of $\bar{F}$ in the initial state, and by a lower index each component relative to the basis of $\bar{F}$ in the final state or of $F$ in the initial state. Since the adjoint representation is real, the components relative to the vector space $A$ will be placed up or down, indifferently.

An expression for the components of the tensor associated to a particular graph on the canonical basis of the corresponding tensor product space can be obtained in the following way:

- Assign to incoming and outgoing lines and to both ends of each propagator color indices.

- Assign to each three-particle vertex the invariant tensors

$$i \searrow \underset{a}{\diagdown} \swarrow j = (t^a)^j_{\ i} \,, \qquad c \,\diagdown \underset{a}{\diagup} b = -i f^{abc} \,. \tag{A.23}$$

As for the first vertex, whether the gluon hooks from above or below does not make a difference. As for the second one, the adjoint color indices need to be ordered counterclockwise.

- Replace the propagators by the identity tensors

$$i \blacktriangleright j = \delta^j_{\ i} \,, \qquad a \,\sim\, b = \delta_{ba} \,, \tag{A.24}$$

and sum over the indices that are repeated twice. This amounts to contracting the indices at both ends of the same propagators.

The left-over uncontracted indices are those labeling the components of the tensor or the elements of the linear map represented by the diagram. There are $n_i + \bar{n}_f$ such indices upstairs, $\bar{n}_i + n_f$ downstairs, and $m_i + m_f$ adjoint indices (upstairs or downstairs). In general, the order in which the indices corresponding to a given representation space $(F, \bar{F}, A)$ appear matters.

Traces of tensors, which are scalars, are represented by graphs with no external leg. Useful elementary examples are the following:

$$\bigcirc = \delta^i_{\ i} = N\,, \qquad \text{🌀} = \delta_{aa} = N^2 - 1\,. \tag{A.25}$$

As a simple illustration, here are the components of the tensor associated to the "self-energy" graph, obtained from the rules just stated:

$$i \longrightarrow \underset{i' \quad j'}{\overset{a \quad b}{\bullet \longrightarrow \bullet}} \longrightarrow j \; = (t^b)^j_{\ j'} \delta^{j'}_{\ i'} (t^a)^{i'}_{\ i} \delta_{ba} = (t^a t^a)^j_{\ i}\,. \tag{A.26}$$

One may reduce a tensor through algebraic manipulations, using basically the Lie bracket and the Fierz identity. But one may perform these manipulations graphically. The pictorial representation of the Lie algebra relation is given in Eq. (172), and that of the Fierz identity (178) in Eq. (179). [We have also introduced a notation for the insertion of Wilson lines, and the fundamental relation (174) between adjoint and fundamental Wilson lines is represented by the birdtrack (175).]

Let us evaluate the color factor (A.26), which is just the quadratic Casimir operator of the fundamental representation; see Eq. (145). We can make use of the Fierz identity (178) in order to express the contraction of the matrix elements of the infinitesimal generators $t^a$ with the help of components of the identity tensor of the fundamental representation only. This can of course be performed algebraically, but let us do it purely pictorially. It amounts to getting rid of the gluon line:

$$i \longrightarrow \hspace{-0.5em} \underset{\text{🌀}}{\bullet \longrightarrow \bullet} \hspace{-0.5em} \longrightarrow j \; = \frac{1}{2} \left( i \longrightarrow \hspace{-0.5em} \bigcirc \hspace{-0.5em} \longrightarrow j - \frac{1}{N}\, i \longrightarrow \hspace{-0.2em} \bullet \longrightarrow \bullet \longrightarrow j \right)\,. \tag{A.27}$$

We then factorize the quark line connecting the indices $i$ and $j$, and eventually replace the quark loop in the first term in the right-hand side by a factor $N$:

$$i \longrightarrow \hspace{-0.5em} \underset{\text{🌀}}{\bullet \longrightarrow \bullet} \hspace{-0.5em} \longrightarrow j \; = \frac{1}{2} \left( \bigcirc - \frac{1}{N} \right) \times \; i \longrightarrow j = \frac{N^2 - 1}{2N}\delta^j_{\ i} = C_F \delta^j_{\ i}\,. \tag{A.28}$$

We have indeed recovered Eq. (145).

Note that, thanks to the adopted conventions, this diagram in which the gluon couples twice to a quark line has exactly the same expression as the following one, in which the gluon couples to a pair of incoming quark-antiquark:

$$\begin{matrix} i \longrightarrow \bullet \\ \quad \text{🌀} \\ j \longleftarrow \bullet \end{matrix}\,. \tag{A.29}$$

Only the interpretation differs: while the initial graph represents a map $F \to F$, the rotated one represents an invariant of the tensor product space $F \otimes \bar{F}$.

# B   Virtual photon wave function in a quark-antiquark pair

In this appendix, we calculate the wave function and the probability density to find the initial virtual photon in the state of a quark-antiquark pair of flavor $q$ at lowest order in the electromagnetic coupling constant. The quark has mass $m_q$ and charge $\mathcal{Q}_q e$, where $e$ is the absolute value of the charge of the electron. The pair constituents have definite "+" momenta and are separated by a given transverse distance. This is a classical result, first obtained in Ref. [38], and which was rederived and used in many subsequent papers (see e.g. Ref. [87, 119]).

We denote the four-momentum of the photon by $q$, and choose the frame such that $q^\perp = 0$. We define the photon virtuality as $Q^2 \equiv -q^2$. Hence the components of $q$ read

$$q^\mu = \left( q^+, q^- = -\frac{Q^2}{2q^+}, q^\perp = 0 \right). \tag{B.1}$$

Since the photon is virtual, it may come in three different polarization states: two transverse ones (helicities $\lambda = \pm 1$), and a scalar one (helicity $\lambda = 0$). We choose the polarization four-vectors $\varepsilon_{(\lambda)}$ as follows:

$$\text{transverse: } \varepsilon^\mu_{(\lambda=\pm1)} = -\frac{1}{\sqrt{2}}(0,0,\lambda,i), \qquad \text{longitudinal: } \varepsilon^\mu_{(\lambda=0)} = \left( \frac{q^+}{Q}, \frac{Q}{2q^+}, 0, 0 \right). \tag{B.2}$$

We want to eventually arrive at an expression for the modulus squared of the wave function $\widetilde{\varphi}$ defined in Eqs. (346),(344) (see Fig. 21 for the corresponding diagram), summed over the helicities of the quark and of the antiquark, that enters the DIS cross section (350).

**Wave function in momentum space.**   We apply the light cone perturbation theory formula (17) to lowest order in the coupling constant:

$$\psi^{\sigma_0;\sigma_1;\lambda}_{\bar{q}q,\mathrm{i}(\gamma^*)}(\vec{k}_0,\vec{k}_1;\vec{q}) = \frac{\left(\langle \bar{q};\vec{k}_0,\sigma_0| \otimes \langle q;\vec{k}_1,\sigma_1|\right) H^{\mathrm{QED}}_1 |\gamma^*;\vec{q},\lambda\rangle}{q^- - k_0^- - k_1^-}. \tag{B.3}$$

We use the QCD matrix element (122), with the appropriate substitution to convert it to the corresponding QED matrix element describing the branching of a photon into a fermion-antifermion pair. The energies in the denominator are evaluated using the mass-shell relation (32). We get

$$\psi^{\sigma_0;\sigma_1;\lambda}_{\bar{q}q,\mathrm{i}(\gamma^*)}(\vec{k}_0,\vec{k}_1;\vec{q}) = \left[ (2\pi)^3 2q^+ \delta^3(\vec{k}_0+\vec{k}_1-\vec{q}) \right] \frac{e\mathcal{Q}_q \left[ \bar{u}_{(\sigma_1)}(k_1) \slashed{\varepsilon}_{(\lambda)}(q) v_{(\sigma_0)}(k_0) \right]}{Q^2 + \left( k_1^{\perp 2} + m_q^2 \right) q^{+2}/[k_1^+(q^+ - k_1^+)]}. \tag{B.4}$$

Comparing Eq. (B.4) to Eq. (344), we easily identify $\varphi_q$ as the second factor in the former. We then replace the polarization vector by Eq. (B.2), and we use the matrix elements of Appendix A.2. After some algebra, we find the following expression for the wave function of the transversely-polarized photon, for which $\slashed{\varepsilon}_{(\lambda)}(q) = \frac{1}{\sqrt{2}}(\lambda\gamma^1 + i\gamma^2)$:

$$\varphi^{\sigma_0;\sigma_1;\lambda=\pm1}_q(k_1^\perp,z) = -e\mathcal{Q}_q \sqrt{z(1-z)}$$
$$\times \frac{\delta_{\sigma_1,-\sigma_0}\varepsilon^\perp_{(\lambda)} \cdot k_1^\perp(1-2z-2\sigma_1\lambda) + \delta_{\sigma_1,\sigma_0} m_q(1+2\sigma_1\lambda)/\sqrt{2}}{Q^2 z(1-z) + k_1^{\perp 2} + m_q^2}. \tag{B.5}$$

In the same way, we find that the wave function of the longitudinally-polarized photon reads

$$\varphi^{\sigma_0;\sigma_1;\lambda=0}_q(k_1^\perp,z) = e\mathcal{Q}_q \delta_{\sigma_1,-\sigma_0} \frac{\sqrt{z(1-z)}}{Q} \frac{Q^2 z(1-z) - k_1^{\perp 2} - m_q^2}{Q^2 z(1-z) + k_1^{\perp 2} + m_q^2}. \tag{B.6}$$

**Wave function and probability density in coordinate space.** We perform the Fourier transform of Eqs. (B.5), (B.6) with respect to the transverse variable, as defined in Eq. (110) (see Fig. 21 for the notations). We need the following formulas:

$$\int \frac{d^2 k^\perp}{2\pi} \frac{e^{ik^\perp \cdot x^\perp}}{k^{\perp 2} + M^2} = K_0(M|x^\perp|), \qquad \int \frac{d^2 k^\perp}{2\pi} e^{ik^\perp \cdot x^\perp} \frac{k^\perp}{k^{\perp 2} + M^2} = i\frac{M x^\perp}{|x^\perp|} K_1(M|x^\perp|), \quad \text{(B.7)}$$

where $K_0$, $K_1$ are modified Bessel functions of the second kind.

As for the transversely-polarized photon, we find

$$\varphi_q^{\sigma_0;\sigma_1;\lambda=\pm 1}(x_{01}^\perp, z) = -\frac{e\mathcal{Q}_q}{2\pi}\sqrt{z(1-z)}\Bigg[\delta_{\sigma_1,-\sigma_0}(1-2z-2\lambda\sigma_1)iM_q \frac{\varepsilon_{(\lambda)}^\perp \cdot x_{01}^\perp}{x_{01}} K_1(M_q x_{01})$$
$$+ \delta_{\sigma_1,\sigma_0}\frac{m_q(1+2\lambda\sigma_1)}{\sqrt{2}}K_0(M_q x_{01})\Bigg], \qquad \text{(B.8)}$$

with $M_q^2 \equiv m_q^2 + Q^2 z(1-z)$. We take the modulus squared and sum over the helicities of the quarks. We also average over the two transverse polarization states of the initial photon which are usually not measured

$$\frac{1}{2}\sum_{\lambda=\pm 1}\sum_{\sigma_0,\sigma_1=\pm 1/2}\left|\varphi_q^{\sigma_0;\sigma_1;\lambda}(x_{01}^\perp, z)\right|^2$$
$$= \frac{\alpha_{\text{em}}\mathcal{Q}_q^2}{\pi}2z(1-z)\left\{[z^2+(1-z)^2][M_q K_1(M_q x_{01})]^2 + [m_q K_0(M_q x_{01})]^2\right\}, \qquad \text{(B.9)}$$

where $\alpha_{\text{em}} \equiv e^2/(4\pi)$ is the fine structure constant.

As for the longitudinal polarization, we rewrite the wave function (B.6) as

$$\varphi_q^{\sigma_0;\sigma_1;\lambda=0}(k_1^\perp, z) = e\mathcal{Q}_q \delta_{\sigma_1,-\sigma_0}\frac{\sqrt{z(1-z)}}{Q}\left(\frac{2Q^2 z(1-z)}{Q^2 z(1-z) + k_1^{\perp 2} + m_q^2} - 1\right). \qquad \text{(B.10)}$$

The Fourier transform of the second term is proportional to $\delta^2(x_{01}^\perp)$. Since the scattering amplitude of a zero-size dipole is null, it can be discarded. The remaining term reads, after Fourier transformation,

$$\varphi_q^{\sigma_0;\sigma_1;\lambda=0}(x_{01}^\perp, z) = \delta_{\sigma_1,-\sigma_0}\frac{e\mathcal{Q}_q}{\pi}Q z^{3/2}(1-z)^{3/2}K_0(M_q x_{01}). \qquad \text{(B.11)}$$

Hence the probability density reads

$$\sum_{\sigma_0,\sigma_1}\left|\varphi_q^{\sigma_0;\sigma_1;\lambda=0}(x_{01}^\perp, z)\right|^2 = \frac{\alpha_{\text{em}}\mathcal{Q}_q^2}{\pi}8Q^2 z^3(1-z)^3\left[K_0(M_q x_{01})\right]^2. \qquad \text{(B.12)}$$

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
