# Peer review of "Scattering from an external field in quantum chromodynamics at high energies: from foundations to interdisciplinary connections"

_SciPost Physics Lecture Notes, doi:SciPost Phys. Lect. Notes 92 (2025)_

## Round 2 · Referee Report · Anonymous (Referee 1) · 2024-12-10

Strengths
Weaknesses
Report
Before the paper can be published I have few minor points to be addressed.
1 Below formula (54) the Authors describe the inverse derivative. Their discuss in words the meaning of this operator. I suggest to write explicitly the formula i.e. as integral operator acting on test function
2 While the equation equation is not closed there exist closed equation equivalent to the Balitsky hierarchy i.e. JIMWLK equation. I suggest that the Authors mention it here
3 After the equation (245) the Authors discuss the quantity q_n(t+\Delta t). The transition from the "Random walk" section is not clear. Furthermore the authors arrive at eqn 247 and 249. As the lecture notes are on QCD and statistical physics it would be good to mention that the eq (249) appeared in the context of Mueller model A. H. Mueller, “Unitarity and the bfkl pomeron,” Nuclear Physics B 437 no. 1, (1995) and eq (247) in Lublinsky, Levin Nucl.Phys.A 730 (2004) 191-211
Recommendation
Publish (easily meets expectations and criteria for this Journal; among top 50%)

---

## Round 2 · Referee Report · Anonymous (Referee 2) · 2024-12-12

Report
Requested changes
1) In the introduction of eikonal scattering (the paragraph above Eq.(33)) the authors state that the "the external potential is unaffected, technically..." This is true, technically, but that does not explain what is going physicswise. The physics point of the matter is that the projectile is moving fast against the field. We do not boost the whole system -- rather we accelerate the probe relative to the target. One meets boost very differently as a student, as a means to discuss inertial frames, but that is precisely not what is going on here and a good fraction of your readers will likely be confused, my own students have been often encough. I would urge the authors to add a paragraph to clarify the issue.
2) Before and in Eq. (81) fundamental and defining appear as synomymous, without you linking them. Your intended audience likely needs either one of the two be used exlusively or the fact that they are synonymous spelled out.
3) I personally find the exposition of branching random walks sufficintly clear and the notation in Eq. (236) self explanatory. However: Why $2\mu+r \le 1$ is imposed should be explained. Likewise the notation used in Eq. (240) with the zeroes below the "location bucket graphs" fail to be fully self explanatory and does need explanations.
Recommendation
Ask for minor revision

---

## Round 3 · Referee Report · Anonymous (Referee 1) · 2025-1-29

Report

The Authors addressed my comments. I think that the lectures are very valuable and I suggest to accept them for publication.

Recommendation

Publish (easily meets expectations and criteria for this Journal; among top 50%)

---

## Round 3 · Referee Report · Anonymous (Referee 2) · 2025-1-31

Strengths

The paper is a valuable review of relation of high energy QCD evolution equations to statistical physics. It present the material in great detail.

Weaknesses

I do not see weeknesses of the paper

Report

The paper is a very valuable write up of lecture on relation of QCD at high energy to statistical physics. While the paper is focused on theory the equations that it discusses are of phenomenological interest. In particular tt discusses BFKL and BK equations and their relation to the FKKP equation. In particular I appreciate very much rederivation of the BK equation.

Requested changes

I find all requested changes to be implemented in a satisfactory manner

Recommendation

Publish (surpasses expectations and criteria for this Journal; among top 10%)

---

## Round 3 · Author Response

We thank the Referees for their positive feedback and valuable suggestions.

AUTHORS' RESPONSE TO REPORT #2

1) We are happy to take the Referee's suggestion into account by adding a few sentences to the paragraph 'Eikonal scattering'.

2) The term 'fundamental representation' is used in the physics literature with various meanings, not always aligning with the mathematical definition. Probably in most cases, it is used as a synonym for the 'defining' or 'standard' representation. For clarity, we will adopt this terminology in the revised version, allowing us to omit the term 'defining representation' (except for a new footnote), and refer solely to the 'fundamental representation.' We will use the term 'anti-fundamental representation' to denote the complex conjugate of the defining representation.

3) We will refine the definition of the branching random walk model and, in particular, clarify that the constraint on the parameters is necessary to ensure that the elementary probabilities are well-defined. Regarding the notation in Eq. (240), the zeros below the 'location bucket graphs' were intended to indicate the lattice site labeled '0,' where the particle starts. Since the transition probabilities in Eq. (236) are constant and independent of the lattice site, these labels are actually unnecessary. Therefore, we have decided to remove them from all equations where they appeared, namely Eqs. (240) and (253).

AUTHORS' RESPONSE TO REPORT #1

1) It will indeed be better to provide a formula for the 'inverse derivative' operator, especially since its definition requires an explicit choice of boundary conditions.

2) Equation (232) is just the first in an infinite hierarchy of equations. However, as the Referee points out, alternative formulations for the rapidity evolution of dipole-nucleus S-matrix elements (and even for any correlator of Wilson lines at a finite number of colors) do exist, and they are expressed in terms of closed equations. We agree that it will be worth mentioning this in Sec. 4.3.3. (NB: The complexity of these formulations lies in the fact that they appear either as functional equations or, equivalently, as stochastic differential equations.)

3) The initial idea was to discuss random walks and Brownian motion separately from pure branching (i.e., zero-dimensional) models in the first stage, before combining the two processes to address branching Brownian motion. Exact solutions can indeed be derived for the relevant observables in the former processes, and we found it useful to work through these.

Zero-dimensional branching models in general, and Eqs. (247) and (249) in particular, were discussed as toy models for QCD evolution. In the revised version, we emphasize this fact and include the references suggested by the Referee.

---

## Round 3 · List of Changes

• In Sec. 2.2 (paragraph "Eikonal scattering"), we added a physical explanation of why the boost does not affect the external potential.

  • In Sec. 2.3.1, we included a paragraph to clarify the group theory terminology used throughout our discussion.

  • Further down in Sec. 2.3.1, we provided the full definition of the inverse derivative operator and its "square." We slightly modified a statement made in Sec. 2.3.3, right after the definition of the 4-polarization vector (and Dirac spinors), in order to refer to this newly-introduced definition of the inverse derivative operator.

  • In Sec. 4.3.3, we added a paragraph introducing the JIMWLK equation.

  • In Sec. 5.1.1, we clarified the definition of the branching random walk model we consider.

  • Further down in Sec. 5.1.1, just before the paragraph on "Branching Brownian motion," we included a paragraph referencing relevant works on zero-dimensional models.

  • Various inconsequential phrasing improvements have been made, along with corrections to a few typographical errors.

---

## Editorial Decision

published